# OmniEVA: Embodied Versatile PlAnner via Task-Adaptive 3D-Grounded and Embodiment-aware Reasoning

Yuecheng Liu*, Dafeng Chi*, Shiguang Wu*, Zhanguang Zhang*, Yuzheng Zhuang†,
Bowen Yang, He Zhu, Lingfeng Zhang, Pengwei Xie, David Gamaliel Arcos Bravo,
Yingxue Zhang, Jianye Hao, Xingyue Quan

Huawei Noah's Ark Lab

## Abstract

Recent advances in multimodal large language models (MLLMs) have opened new opportunities for embodied intelligence, enabling multimodal understanding, reasoning, and interaction, as well as continuous spatial decision-making. Nevertheless, current MLLM-based embodied systems face two critical limitations. First, *Geometric Adaptability Gap:* models trained solely on 2D inputs or with hard-coded 3D geometry injection suffer from either insufficient spatial information or restricted 2D generalization, leading to poor adaptability across tasks with diverse spatial demands. Second, *Embodiment Constraint Gap:* prior work often neglects the physical constraints of real robots, resulting in task plans that are theoretically valid but practically infeasible.To address these gaps, we introduce **OmniEVA** – an embodied versatile planner that enables advanced embodied reasoning and task planning through two pivotal innovations: (1) a *Task-Adaptive 3D Grounding* mechanism, which uses a gated router to dynamically inject 3D features based on task context, enabling selective geometric reasoning. (2) an *Embodiment-Aware Reasoning* framework that incorporates task goals and physical constraints into the reasoning loop, ensuring executable plans. Extensive experiments show that OmniEVA achieves state-of-the-art performance on 7 of 8 embodied reasoning benchmarks and excels in downstream tasks such as object navigation and mobile manipulation. Evaluations on proposed benchmarks confirm its robust and versatile planning capabilities. Project page.

## 1 Introduction

The rapid progress in multimodal large language models (MLLMs) has significantly enhanced AI's ability to interpret and reason across text, images, and video. This shift has opened new avenues to embodied intelligence (Reed et al., 2022; Ahn et al., 2022; Driess et al., 2023), capable of perceiving, reasoning, and acting in physical environments. Spatial reasoning serves as a core bridge between perception and action, transforming sensory inputs into structured representations that support long-horizon planning and rational decision-making.

Early vision–language models addressed spatial reasoning primarily in two dimensions. SpatialVLM (Chen et al., 2024a) introduced large-scale synthetic VQA grounded in real imagery, while RoboPoint (Yuan et al., 2024a), RoboSpatial (Song et al., 2025), and RoboRefer (Zhou et al., 2025) incorporated fine-grained spatial grounding by predicting coordinates or bounding boxes from language prompts. RoboBrain (Ji et al., 2025; Team et al., 2025a) further unified high-level planning with low-level spatial pointing, outperforming general MLLMs on embodied benchmarks. More recently, 3D LLMs have extended reasoning beyond 2D perception by incorporating point clouds, voxel grids, or 3D position embeddings into MLLMs (Huang et al., 2023c; Zhu et al., 2024a; Hong et al., 2023; Zheng et al., 2025; Huang et al., 2025).

*: Equal contribution. {liuyuecheng1, chidafeng1, wushiguang, zhanguang.zhang}@huawei.com; †: Corresponding author. zhuangyuzheng@huawei.com

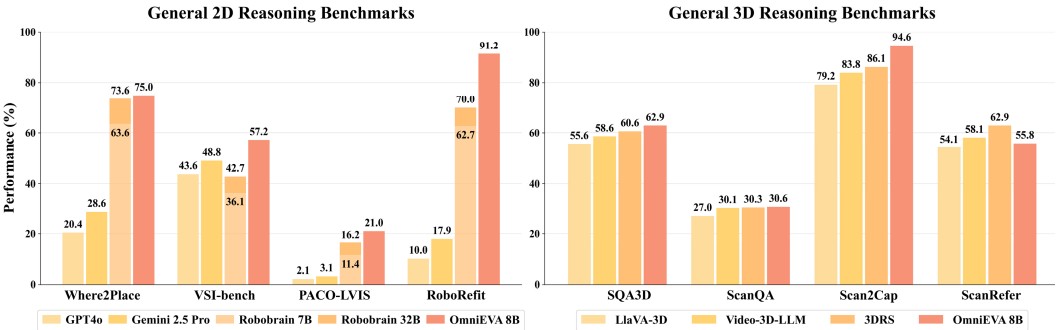

Figure 1: Performance Comparison across 2D and 3D Embodied Reasoning Benchmarks.

Despite recent progress, two core challenges remain. **First, the geometric adaptability gap**: models trained solely on 2D inputs struggle with tasks that require strong spatial reasoning, such as object stacking, occlusion handling, or navigation in cluttered 3D scenes. This limitation arises from the absence of explicit 3D structural encoding, which restricts generalization in geometry-rich environments. Existing 3D-LLM approaches (Zhu et al., 2024a; Zheng et al., 2025; Huang et al., 2025) often depend on hard-coded 3D injection strategies that ignore task relevance, resulting in unnecessary computation and noisy embeddings when 3D inputs are incomplete or nonessential. **Second, the embodiment constraint gap**: current methods often rely on labeled web-scale image or video datasets, or on rule-based synthetic simulations. Models trained on such data frequently overlook the constraints and capabilities of real robots, producing plans that may appear valid in theory but are infeasible in practice. In particular, neglecting object affordances, workspace limitations, and kinematic feasibility leads to action sequences that cannot be executed on physical platforms. Furthermore, the absence of embodied long-horizon planning benchmarks that explicitly incorporate embodiment constraints makes it difficult to systematically evaluate the unique challenges they pose. To address these limitations, we introduce **OmniEVA** (Embodied Versatile Planner), a novel architecture that pioneers **Task-Adaptive 3D Grounding** and **Embodiment-aware Reasoning**. OmniEVA is the first framework to dynamically integrate 2D and 3D inputs via task-conditioned feature selection, enabling versatile and executable embodied reasoning through two key innovations:

- **Task-Adaptive 3D Grounding**: We introduce a gated routing mechanism that dynamically modulates the infusion of 3D features into the visual-language backbone based on contextual task requirements. This allows for explicit, selective geometric grounding only when spatially essential, avoiding the drawbacks of static 3D fusion and enabling robust performance across both 2D and 3D reasoning tasks.

- **Embodiment-Aware Reasoning**: Moving beyond passive scene understanding, OmniEVA jointly incorporates task goals, environmental context, and physical constraints into its reasoning process. Through post-training with our proposed Task- and Embodiment-aware GRPO (TE-GRPO) algorithm, the model learns to generate plans that respect object affordances, workspace boundaries, and kinematic limits, significantly improving executability on real robots.

We evaluate OmniEVA on 8 public benchmarks spanning 2D, 3D, and video-based reasoning. As shown in Figure 1, OmniEVA achieves SOTA performance on 7 benchmarks and tops leaderboards in object navigation (HM3D, MP3D). To further probe embodiment awareness, we contribute four primitive benchmarks—Where2Go, Where2Grasp, Where2Approach, and Where2Fit—each targeting a core skill for long-horizon planning. OmniEVA outperforms existing models across all primitives, demonstrating strong generalization to downstream tasks like mobile manipulation.

## 2 RELATED WORK

**MLLMs for Embodied Reasoning**   Multimodal large language models (MLLMs) have recently improved spatial reasoning via synthetic datasets and spatially grounded visual question answering (VQA). SpatialVLM (Chen et al., 2024a) introduced large-scale spatial QA on real-world images. Building on this, models such as RoboPoint (Yuan et al., 2024a), Robospatial (Song et al., 2025)

and RoboRefer (Zhou et al., 2025) introduced fine-grained spatial outputs, including coordinate prediction and bounding box localization. RoboBrain (Ji et al., 2025; Team et al., 2025a) further advanced this line by integrating high-level planning with low-level spatial pointing, outperforming general-purpose MLLMs on embodied reasoning benchmarks. To assess reasoning and planning in dynamic or large-scale environments, several video-based benchmarks have also emerged, such as VSI-Bench (Yang et al., 2025b) and EgoPlan (Chen et al., 2023). However, despite these developments, most embodied reasoning models remain limited by their reliance on 2D inputs, lacking the capacity to fully interpret environments with complex 3D geometric structures.

**3D Large Language Models** Efforts to extend LLMs to 3D modalities have explored representations such as point clouds (Huang et al., 2023c; Zhu et al., 2024b; Chen et al., 2024d) and voxel grids (Hong et al., 2023; Zhang et al., 2025). More rencent approaches inject 3D positional information into visual tokens, enabling pretrained MLLMs to perform spatial reasoning in three dimensions (Zhu et al., 2024a; Zheng et al., 2025; Huang et al., 2025). While these methods have achieved state-of-the-art results on several 3D benchmarks, the hard-coded 3D injection methods can be problematic when 3D inputs are noisy, incomplete, or irrelevant to the task.

## 3 METHODOLOGY

### 3.1 OVERVIEW

OmniEVA builds on pretrained MLLMs which typically comprises three principal components: 1) A vision transformer encoder $\mathcal{E}_{\text{img}}$ that converts each RGB image into a sequence of discrete visual tokens, 2) a lightweight network that maps visual embeddings into the language model's latent space for seamless cross-modal interaction and 3) an autoregressive text decoder $\mathcal{T}$ that generates output tokens. The model accepts a natural language instruction $T$, a sequence of RGB images or video frames $(I_1, I_2, \ldots, I_N)$, and optionally, depth maps $(D_1, D_2, \ldots, D_N)$ for each view. To support cross-view spatial understanding, the model also ingests camera intrinsic parameters $K$ and extrinsic poses $M_i$ corresponding to each frame.

Conventional MLLMs such as QwenVL and InternVL split each frame into $H_{\text{p}} \times W_{\text{p}}$ patches, augment them with 2D positional encodings, and feed the flattened sequence into $\mathcal{E}_{\text{img}}$. For $N$ frames, the encoder outputs $V^I \in \mathbb{R}^{N \times H_{\text{p}} \times W_{\text{p}} \times d_v}$, where $d_v$ denodes the embedding dimension. While effective for many vision-language tasks, this approach omits direct 3D information—depth values or world coordinates—which is critical for complex geometric reasoning. Recent 3D LLMs, such as 3DRS(Huang et al., 2025), rely on static architecture to integrate 3D features, limiting their flexibility in tasks where such features are unnecessary. OmniEVA introduces a *Task-Adaptive Gated Router* (TAGR) to dynamically fuse 3D features and a two-stage training paradigm to enable *Embodiment-aware Planning*.

### 3.2 TASK-ADAPTIVE GATED ROUTER

The module of *Task-Adaptive Gated Router* (TAGR) is illustrated in Figure 2. TAGR serves as a dynamic mediator between task demands and spatial complexity, selectively regulating the injection of 3D positional encodings. We will introduce the details of the framework in the following sections.

**Patch-Level 3D Positional Encoding** Each depth image $D_i \in \mathbb{R}^{H \times W}$ is projected into a world coordinate matrix $P_i \in \mathbb{R}^{H \times W \times 3}$ using the camera parameters. The 3D coordinates $P_i$ is then partitioned into patches aligned with the patch size of the RGB image processed by the ViT Encoder. For each patch, the 3D coordinates of all pixels are averaged, producing a patched coordinate matrix $P_i' \in \mathbb{R}^{H_{\text{p}} \times W_{\text{p}} \times 3}$. Finally, a sinusoidal encoding is applied to the 3D coordinates of each patch, mapping them into vectors of dimension $d_v$. For $N$ frames, this process yields the 3D positional encoding features denoted as $V^p \in \mathbb{R}^{N \times H_p \times W_p \times d_v}$.

**Dynamic 3D Injection via Gated Routing** Rather than applying 3D positional encoding uniformly for all tasks, we propose a *Task-Adaptive Gated Router* (TAGR) that selectively integrates 3D information based on task-specific requirements. TAGR determines whether to inject 3D positional priors based on two conditioning signals: 1) the *task condition*, reflecting the nature of the

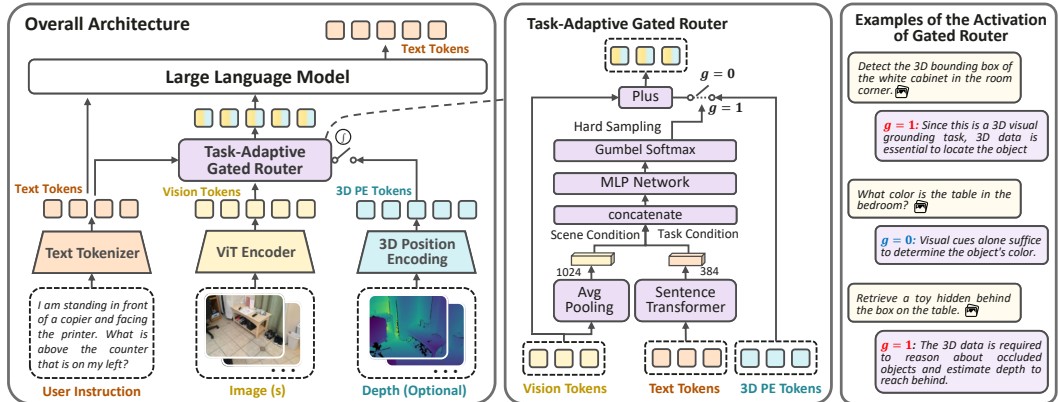

**Figure 2: Model Architecture of OmniEVA**. **Left**: The overall architecture of OmniEVA, featuring a novel task-adaptive gated router that dynamically incorporates 3D positional embeddings. **Middle**: Detailed implementation of the gated router module. **Right**: Illustrative examples of the gated router's activation state across different tasks.

task to be performed, and 2) the *scene condition*, reflecting the structural complexity of the visual input. For task conditioning, a lightweight sentence transformer (Reimers & Gurevych, 2019) encodes the instruction $T$ into a latent vector $V^T \in \mathbb{R}^{d_{st}}$. For scene conditioning, the vision encoder output $V^I \in \mathbb{R}^{N \times H_p \times W_p \times d_v}$ is aggregated via average pooling to obtain a global scene descriptor $V_{avg}^I \in \mathbb{R}^{d_v}$:

$$V^T = \text{SentenceTransformer}(T) \tag{1}$$

$$V_{avg}^I = \text{AvgPooling}(V^I, \dim = 0, 1, 2) \tag{2}$$

The concatenated vector $[V^T, V_{avg}^I]$ is passed through a multi-layer perceptron (MLP) module to produce gate logits $V^g \in \mathbb{R}^2$, which represent the probabilities corresponding to the activation and deactivation of the gate module.

$$V^g = \text{MLP}_\psi(\text{Concatenate}([V^T, V_{avg}^I])) \in \mathbb{R}^2 \tag{3}$$

To preserve the amplitude of sinusoidal encodings $V^p$—whose distortion can lead to significant performance degradation—we avoid soft-weighting and instead employ hard gating via Gumbel-Softmax (Jang et al., 2016) to enable end-to-end gradient propagation, where $\tau$ is the temperature. When $g = 1$ the model augments 3D features with 3D cues; when $g = 0$ the model uses only 2D visual features. This is equivalent to a Mixture-of-Experts (MoE) between pure visual tokens $V^I$ and the fused tokens $(V^I + V^p)$, as shown in Equation 5. The resulting hybrid visual tokens and the text tokens are then passed to the LLM backbone $\mathcal{F}_\theta^{llm}(\cdot)$ to generate the response tokens $o$.

$$g = \text{GumbelSoftmax}(V^g, \tau) \in \{0, 1\} \tag{4}$$

$$V_{hybrid}^I = V^I + g \cdot V^p = (1 - g)V^I + g(V^I + V^p) \tag{5}$$

$$o = \mathcal{F}_\theta^{llm}(T, V_{hybrid}^I) \tag{6}$$

The TAGR module is optimized using cross-entropy loss to align the predicted output $o$ with the ground truth label $o^{label}$. To stabilize and regularize gating behavior, we add a KL divergence term between the predicted gate distribution and a Bernoulli(0.5) prior,

$$\mathcal{L}_{\psi, \theta}^{total} = \mathcal{L}_{\psi, \theta}^{CE}(o^{label}, o) + \alpha \cdot \mathcal{L}_\psi^{KL}(V^g || \mathcal{P}_{prior}) \tag{7}$$

### 3.3 EMBODIMENT-AWARE TRAINING STRATEGY

To unify perception, reasoning, and execution across heterogeneous embodied tasks, we introduce a clear, sequential three-stage training pipeline: (1) Task-Adaptive Gated Router (TAGR) Pretraining (detailed in Section 3.2), (2) Supervised Fine-Tuning (SFT), and (3) Task- and Embodiment-aware Reinforced Finetuning (RFT). This cascaded approach progressively enables the model to evolve from foundational spatial understanding to sophisticated, executable plan generation.

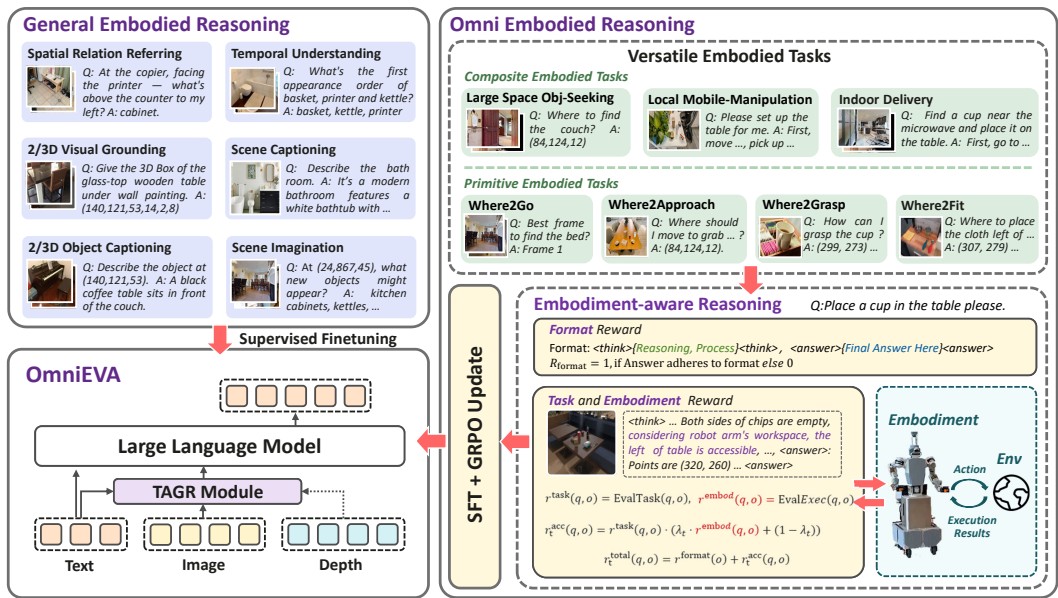

**Figure 3: Training Paradigm of OmniEVA**. The two-stage cascade progressively enhances embodied intelligence: Stage 1 builds a broad reasoning foundation, while Stage 2 grounds it in physical reality—culminating in robust task execution across diverse real-world scenarios.

### 3.3.1 STAGE 1: TASK-ADAPTIVE GATED ROUTER PRETRAINING

In this initial stage, the task-adaptive gated router is pretrained to learn dynamic 3D feature integration gating behavior, utilizing the optimization objective defined in Equation 7. The training incorporates a combination of depth-aware spatial reasoning datasets, including scene data from ScanNet, Matterport3D, 3RScan, and ArkitScenes (detailed in Appendix D). To facilitate effective learning of the TAGR module, we apply a differentiated learning rate strategy: a small rate ($5e^{-7}$) for the LLM backbone to preserve pretrained knowledge, and a larger rate ($1e^{-4}$) for the TAGR parameters to accelerate adaptation. Upon completion of pretraining, the TAGR parameters are frozen for use in subsequent stages, while the fine-tuned LLM backbone parameters are discarded to avoid interference.

### 3.3.2 STAGE 2: SUPERVISED FINE-TUNING FOR GENERAL EMBODIED REASONING

This stage takes the TAGR-pretrained model as its input to establish a robust foundation for general embodied reasoning. We curate a hybrid dataset comprising: (1) General Embodied Reasoning Data: Multimodal sources (2D images, video, 3D) for tasks like spatial relation referring and scene captioning, building omni-dimensional spatial cognition (holistic scene understanding). (2) Custom Embodied Task Data: Navigation and manipulation tasks that foster initial embodiment-aware reasoning (basic physical interaction awareness). The model resulting from the SFT stage is designated as **OmniEVA-Base**. Details of the dataset usage in this stage are provided in Appendix D.

### 3.3.3 STAGE 3: TASK- AND EMBODIMENT-AWARE REINFORCED FINETUNING

This final stage uses **OmniEVA-Base** as its starting point to ground reasoning in physical reality. While prior methods have largely focused on improving semantic fidelity, they often neglect the physical feasibility of generated plans. TE-GRPO seeks to bridge this gap by promoting outputs that are not only semantically aligned with task objectives but also executable within the constraints of robotic embodiment. Building on GRPO (Shao et al., 2024), we keep the format reward $r^{\text{format}}$ to encourage the "think-answer" reasoning structure. To further guide the model toward generating both task-directed and physically feasible plans, we introduce two additional reward components:

$$r_i^{\text{task}}(q, o_i) = \text{EvalTask}(q, o_i) \in [0, 1], \quad r_i^{\text{embod}}(q, o_i) = \text{EvalExec}(q, o_i) \in \{0, 1\} \quad (8)$$

where $q$ is the user prompt and $o_i$ the $i-$th model response. EvalTask($\cdot$) measures semantic task satisfaction independent of embodiment (e.g., fraction of points placed inside a target region for a pointing task). In contrast, EvalExec($\cdot$) verifies embodiment feasibility by checking kinematics, reachability, and environment constraints in simulation. These two reward components reflect distinct optimization objectives: $r_i^{\text{task}}$ emphasizes performance on offline evaluation benchmarks, while $r_i^{\text{embod}}$ targets end-to-end execution success in real-world robotic deployments.

**Progressive Embodiment Curriculum**   To accelerate convergence and promote physically grounded reasoning, we employ a curriculum learning-inspired reward scheduling strategy. This approach gradually transitions the model's optimization focus from semantic correctness to embodiment feasibility. At training step $t$, the composite accuracy reward $r_{i,t}^{\text{acc}}(q, o_i)$ is defined as,

$$r_{i,t}^{\text{acc}}(q, o_i) = r_i^{\text{task}}(q, o_i) \cdot \left( \lambda_t \cdot r_i^{\text{embod}}(q, o_i) + (1 - \lambda_t) \right) \tag{9}$$

where $\lambda_t \in [0, 1]$ is a scheduling coefficient that increases over time, gradually shifting the model's focus from task completion to embodiment feasibility. Early in training, $\lambda_t \approx 0$, allowing the model to receive positive reward even when embodiment constraints are only partially satisfied. As training progresses, $\lambda_t \to 1$, enforcing stricter compliance with physical constraints. The final reward for the $i$-th response is then computed as:

$$r_{i,t}(q, o_i) = r_i^{\text{format}}(o_i) + r_{i,t}^{\text{acc}}(q, o_i) \tag{10}$$

Policy updates follow the original GRPO objective. See Appendix A.3 for detail. The model resulting from this RFT stage is designated as **OmniEVA-ER** (ER for Embodiement-aware Reasoning).

## 4   EXPERIMENTAL RESULTS

This section presents a comprehensive evaluation of OmniEVA to validate its effectiveness and answer two pivotal questions: (1) How does the proposed dynamic 3D-grounding mechanism enhance multimodal reasoning? (2) To what extent does the embodiment-aware reasoning framework improve task success rates under physical constraints?

To ensure clarity, we evaluate two distinct model checkpoints: **OmniEVA-Base** (after Stage 2 SFT), which is equipped with strong general reasoning capabilities, and **OmniEVA-ER** (after Stage 3 RFT), which is optimized for physical feasibility. OmniEVA-Base's results are in Tables 2-4; OmniEVA-ER's are in Figure 5. Implementation details are provided in Appendix A.

### 4.1   BENCHMARKS FOR EVALUATION

**Embodied Reasoning Benchmarks with 2D Inputs**   To assess the model's embodied reasoning capabilities across visual modalities, we employ four established benchmarks: **Where2Place** (Yuan et al., 2024a), **VSI-bench** (Yang et al., 2025b), **PACO-LVIS** (Ramanathan et al., 2023), and **RoboRefit** (Lu et al., 2023). [1] These datasets span both static images and dynamic video inputs, enabling comprehensive evaluation of spatial and temporal understanding and multimodal reasoning. Furthermore, we introduce four specialized benchmarks targeting fundamental embodied reasoning skills: **Where2Go** for next-best view selection under partial observability, **Where2Fit** for free-space prediction with collision constraints, **Where2Approach** for occlusion-aware navigation, and **Where2Grasp** for object-centric recognition. These primitive benchmarks form the foundation for complex downstream tasks like mobile manipulation. Detailed descriptions are provided in Appendix E.

**Embodied Reasoning Benchmarks with 3D Inputs**   To evaluate 3D spatial reasoning we use **SQA3D** (Ma et al., 2022), **ScanQA** (Azuma et al., 2022), **Scan2Cap** (Chen et al., 2021), and **ScanRefer** (Chen et al., 2020). These benchmarks probe open-ended QA, captioning, and 3D visual grounding in richly structured environments, stressing depth and geometry beyond planar views.

---

[1]Since the annotations in RoboRefit and PACO-LVIS lack VQA pairs, we built a VLM evaluation set based on image distribution, object categories, and part categories. The evaluation code aligns with Where2Place.

**End-to-End Online Evaluation within Simulators** To bridge the gap between high-level planning and low-level robotic execution, we run end-to-end experiments in simulators on composite tasks: Large-Space Object Seeking and Mobile Manipulation. These tasks combine the primitive skills above to measure holistic embodied reasoning and execution.

## 4.2 Task-Adaptive 3D-Grounding: Validation across Multimodal Benchmarks

**Effectiveness of the Task-Adaptive Gated Router** To rigorously evaluate the efficacy of our approach, we compared it against several baselines: (1) **Hard-coded 3D integration**, which statically fuses 3D features into visual tokens across all tasks—a common strategy in prior 3D LLMs; (2) **Without 3D integration**, effectively reducing the model to a traditional 2D MLLM; and (3) **Cross-attention-based fusion**, implemented using either separate or interleaved visual and 3D tokens (see Appendix and Figure 10 for detail), to assess implicit feature weighting. As summarized in Table 1, the proposed dynamic 3D integration approach not only surpasses cross-attention variants but also outperforms state-of-the-art 2D/3D fusion baselines, achieving an average performance gain of 1.2%. Notably, cross-attention methods result in a substantial performance drop—approximately 6 points on SQA3D and a striking 50-point decrease on Scan2Cap. This pronounced degradation highlights the limitations of naive cross-attention for 3D fusion, which doubles the input sequence length and compels the model to learn cross-modal interactions from scratch, thereby compromising both efficiency and generalization. We further analyzed a soft-gating alternative that injects 3D positions via continuous sigmoid weighting. This method consistently underperformed on all benchmarks, as it compromises the numerical stability of the positional encodings crucial for spatial reasoning. The inferior performance of soft-gating validates the superiority of our strategy, which offers a more stable and adaptive solution for geometric integration.

**Table 1: Results of Different 3D-Integration Methods.** To ensure a fair comparison and isolate the impact of 3D integration, models were trained exclusively on the training splits of the SQA3D, ScanQA, Scan2Cap, and ScanRefer datasets. This experimental setup is consistent with prior work like Video-3D-LLM and 3DRS.

| Methods | Benchmark Results | | | | |
|---|---|---|---|---|---|
| | SQA3D | ScanQA | Scan2Cap | ScanRefer | Average |
| **Indivisual 3D Input** | | | | | |
| Cross-Attention (seperate vision & 3d tokens) | 55.1 | 27.5 | 43.3 | 4.5 | 32.6 |
| Cross-Attention (interleaved vision & 3d tokens) | 55.8 | 27.5 | 42.0 | 3.6 | 32.2 |
| **Direct 2D/3D Fusion Methods** | | | | | |
| Hard-coded 3D Integration | 61.2 | 31.5 | 95.5 | 41.2 | 57.3 |
| Without 3D Integration | 61.2 | 30.7 | 75.5 | 4.3 | 42.9 |
| Dynamic 3D Integration, Soft Gating | 60.6 | 30.7 | 85.6 | 26.9 | 51.0 |
| Dynamic 3D Integration, Hard Gating (Ours) | 62.6 | 30.8 | 97.9 | 43.1 | 58.7 |

**When Is the TAGR Module Activated?** We analyze the activation behavior of the task-adaptive gated router (TAGR) by clustering prompts based on semantic categories. As shown in Figure 4, prompts are embedded using a lightweight sentence transformer and grouped by semantic similarity. The activation probabilities vary considerably across categories: *shape-related* prompts trigger the highest activation (76.9%), followed by *action/activity* (50.9%) and *visibility occlusion* (33.0%), indicating that 3D reasoning is strongly invoked by geometric, dynamic, and occlusion-related cues. In contrast, categories such as *state condition* (6.3%), *counting* (9.0%), and *material texture* (19.4%) result in much lower activation, suggesting limited reliance on 3D features for descriptive or numerical tasks. A detailed case study is included in Appendix B.1.

**Comparison Between OmniEVA and State-of-the-Art Models on 2D/3D Benchmarks** Table 2 reports OmniEVA-Base's results on four 2D embodied reasoning benchmarks (**Where2Place**, **VSI-Bench**, **PACO-LVIS**, **RoboRefit**) spanning images and video. Despite its compact 8B parameter size, OmniEVA-Base attains state-of-the-art performance across all benchmarks, surpassing much larger models and yielding an average gain of +10.45 over the previous SOTA (Robobrain2.0-32B (Team et al., 2025a)). Furthermore, OmniEVA-Base demonstrates robust generalization across four additional primitive benchmarks (**Where2Go, Where2Fit, Where2Approach, Where2Grasp**), showing its versatility and superiority in diverse embodied tasks. See Appendix F for qualitative examples.

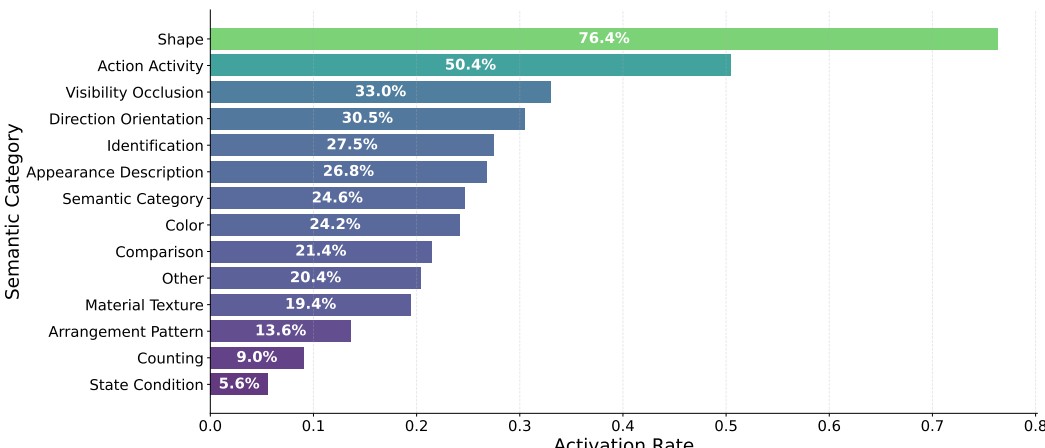

**Figure 4: 3D Activation Analysis by Prompt Clustering**: Prompts are embedded using a lightweight sentence transformer and clustered into semantic groups. The chart shows the 3D activation probability per category.

**Table 2:** 2D General Reasoning Benchmarks and In-house Benchmarks. [1] Hurst et al. (2024),[2] Team et al. (2025b),[3] Zhang et al. (2024b),[4] Li et al. (2024),[5] Zhu et al. (2025),[6] Bai et al. (2025),[7] Yuan et al. (2024a),[8] Azzolini et al. (2025),[9] Luo et al. (2025),[10] Yang et al. (2025a),[11] Team et al. (2025a)

| Models / Benchmarks | Public Embodied Benchmarks | | | | In house Embodied Benchmarks | | | |
|---|---|---|---|---|---|---|---|---|
| | Where2Place | VSI-bench | PACO-LVIS | RoboRefit | Where2Go | Where2Fit | Where2Approach | Where2Grasp |
| **General Models** | | | | | | | | |
| GPT-4o [1] | 20.41 | 43.60 | 2.09 | 9.96 | 50.72 | 37.15 | 0.17 | 6.38 |
| Gemini-2.5-Pro [2] | 28.60 | 48.83 | 3.14 | 17.91 | 55.07 | 41.82 | 3.50 | 27.00 |
| Llava-Next-Video 7B [3] | 4.76 | 35.62 | 1.44 | 1.18 | 31.88 | 61.34 | 0.10 | 0.89 |
| Llava-OneVision 7B [4] | 5.87 | 32.57 | 2.18 | 9.48 | 0.00 | 63.32 | 1.98 | 6.87 |
| InternVL3-8B [5] | 12.68 | 42.89 | 4.57 | 13.76 | 41.06 | 33.07 | 2.08 | 8.63 |
| InternVL3-78B [5] | 21.74 | 48.48 | 3.49 | 21.48 | 51.69 | 41.16 | 1.04 | 11.80 |
| Qwen2.5-VL-7B [6] | 10.99 | 37.51 | 3.21 | 1.21 | 38.16 | 38.59 | 1.50 | 12.75 |
| Qwen2.5-VL-72B [6] | 39.92 | 39.41 | 4.06 | 32.58 | 49.76 | 41.49 | 0.00 | 30.50 |
| **Embodied Models** | | | | | | | | |
| RoboPoint [7] | 46.80 | - | 9.21 | 47.83 | - | 56.64 | 2.46 | 35.97 |
| Cosmos-Reason1-7B [8] | 5.51 | 25.64 | 2.58 | 14.42 | 40.10 | 38.86 | 0.00 | 6.70 |
| VeBrain-8B [9] | 11.34 | 26.30 | 0.89 | 4.00 | 28.98 | 28.47 | 0.00 | 0.00 |
| Magma-8B [10] | 10.89 | 12.65 | 3.23 | 4.95 | 0.00 | 28.45 | 0.00 | 13.50 |
| RoboBrain2.0-7B [11] | 63.59 | 36.10 | 11.38 | 62.74 | 38.64 | 32.99 | 2.85 | 63.24 |
| RoboBrain-2.0-32B [11] | 73.59 | 42.69 | 16.23 | 69.98 | 41.06 | 59.23 | 4.35 | 67.60 |
| **OmniEVA-Base** | **74.95** | **57.17** | **21.01** | **91.19** | **86.96** | **78.14** | **7.37** | **73.91** |

Extending to **3D embodied reasoning**, we evaluated OmniEVA-Base on **SQA3D**, **ScanQA**, **Scan2Cap**, and **ScanRefer**, which encompass 3D question answering, captioning, and 3D visual grounding tasks. As shown in Table 3, OmniEVA-Base leads on three out of four benchmarks, outperforming *state-of-the-art* specialized 3D LLMs such as Video-3D-LLM (Zheng et al., 2025) and 3DRS (Huang et al., 2025) with notable improvements of +2.3, +0.3, and +8.5, respectively. While it slightly trails in 3D visual grounding (ScanRefer), OmniEVA-Base attains 55.8 accuracy (IoU@0.25) using only *text I/O*, *without external detectors or task-specific heads*, substantially exceeding the previous best of 44.4 and showcasing strong end-to-end 3D grounding ability.

In addition to general embodied reasoning, OmniEVA-Base demonstrates strong performance in downstream tasks such as **Object Navigation**, evaluated on the HM3D (Ramakrishnan et al., 2021) and MP3D (Chang et al., 2017) datasets. Here, the model is tasked to predict a 3D subgoal location to guide exploration toward a target object. As shown in Table 4, OmniEVA-Base outperforms the state-of-the-art navigation model **UniNavid** (Zhang et al., 2024a) in both success rate (SR) and path efficiency (SPL), achieving a notable +5.4 improvement in SPL. Qualitative examples of exploration trajectories are provided in Appendix F.2.

**Table 3:** 3D Reasoning Benchmarks. [1] Hong et al. (2023),[2] Zhu et al. (2024b),[3] Huang et al. (2023c),[4] Chen et al. (2024d),[5] Zhang et al. (2025),[6] Wang et al. (2025),[7] Huang et al. (2023b),[8] Huang et al. (2023a),[9] Chen et al. (2024c),[10] Zhu et al. (2024a),[11] Yu et al. (2025),[12] Deng et al. (2025),[13] Zheng et al. (2025),[14] Huang et al. (2025)

| Models | SQA3D | ScanQA | Scan2Cap | ScanRefer | |
|---|---|---|---|---|---|
| | EM | EM | CIDEr | w.a. | w/o.a. |
| **Baseline Models** | | | | | |
| 3D-LLM(Flam) [1] | – | 20.3 | – | – | 21.2 |
| 3D-LLM(blip2) [1] | – | 20.5 | – | – | 30.3 |
| PQ3D [2] | 47.1 | – | – | 57.0 | – |
| LEO [3] | 50.0 | 21.5 | 72.4 | – | – |
| G-3D-LLM [4] | – | – | 70.6 | 47.9 | – |
| SceneLLM [5] | 53.6 | 27.2 | – | – | – |
| S-3D-LLM [6] | 46.2 | – | 72.2 | – | 44.3 |
| ChatScene [7] | 54.6 | 21.6 | 77.1 | 55.5 | – |
| Chat-3D v2 [8] | 54.7 | – | 63.9 | 42.5 | – |
| LL3DA [9] | – | – | 62.9 | – | – |
| LLaVA-3D [10] | 55.6 | 27.0 | 79.2 | 54.1 | – |
| Inst3D-LMM [11] | – | 24.6 | 79.7 | 57.8 | – |
| 3D-LLaVA [12] | 54.5 | – | 62.9 | – | – |
| V-3D LLM[13] | 58.6 | 30.1 | 83.8 | 58.1 | – |
| 3DRS [14] | 60.6 | 30.3 | 86.1 | 62.9 | – |
| **OmniEVA-Base** | 62.9 | 30.6 | 94.6 | – | 55.8 |

**Table 4:** ObjNav Benchmarks [1] Wijmans et al. (2019),[2] Zhou et al. (2023),[3] Wu et al. (2024),[4] Yokoyama et al. (2024),[5] Huang et al. (2024),[6] Yin et al. (2024),[7] Yu et al. (2023),[8] Yin et al. (2025),[9] Ramrakhya et al. (2022),[10] Long et al. (2024),[11] Yadav et al. (2023b),[12] Yadav et al. (2023a),[13] Shah et al. (2023),[14] Ramrakhya et al. (2023),[15] Zhang et al. (2024a),

| Methods | HM3D | | MP3D | |
|---|---|---|---|---|
| | SR | SPL | SR | SPL |
| **Baseline Methods** | | | | |
| DD-PPO [1] | 27.9 | 14.2 | – | – |
| ESC [2] | 39.2 | 22.3 | 28.7 | 14.2 |
| VoroNav [3] | 42.0 | 26.0 | – | – |
| VLFM [4] | 52.5 | 30.4 | 36.4 | 17.5 |
| GAMap [5] | 53.1 | 26.0 | – | – |
| SG-Nav [6] | 54.0 | 24.9 | 40.2 | 16.0 |
| L3MVN [7] | 54.2 | 25.5 | – | – |
| UniGoal [8] | 54.5 | 25.1 | 41.0 | 16.4 |
| Habitat-Web [9] | 57.6 | 23.8 | 31.6 | 8.5 |
| InstructNav [10] | 58.0 | 20.9 | – | – |
| OVRL [11] | 62.0 | 26.8 | 28.6 | 7.4 |
| OVRL-v2 [12] | 62.8 | 28.1 | – | – |
| LFG [13] | 68.9 | 36.0 | – | – |
| PIRLNav [14] | 70.4 | 34.1 | – | – |
| UniNavid [15] | 73.7 | 37.1 | – | – |
| **OmniEVA-Base** | 74.2 | 42.5 | 59.1 | 26.2 |

### 4.3 EMBODIMENT-AWARE REASONING: PERFORMANCE UNDER PHYSICAL CONSTRAINTS

**Effectiveness Across Diverse Embodied Tasks** We evaluate the impact of task- and embodiment-aware reasoning by comparing models trained with and without $r^{\text{task}}$ and $r^{\text{embod}}$. Experiments span primitive skill benchmarks (Where2Approach, Where2Fit, Where2Grasp) and downstream tasks requiring physical execution (Mobile Placement and Mobile Pickup).

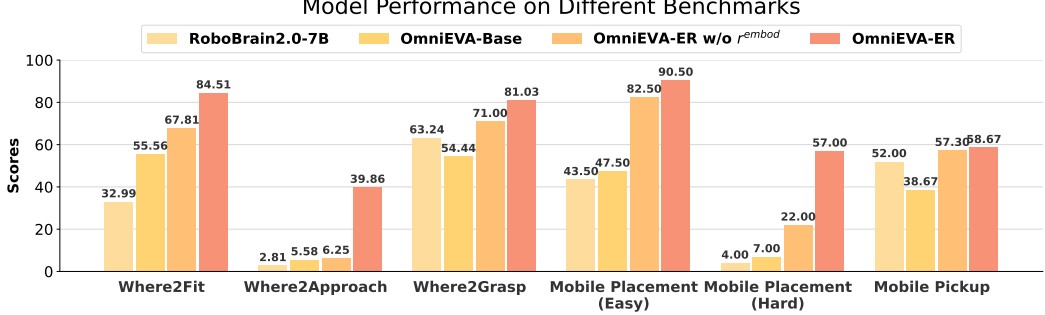

**Figure 5:** Ablation Results of the proposed TE-GRPO Method on Local Mobile-Manipulation Tasks

As shown in Figure 5, OmniEVA-ER—jointly optimized with $r^{\text{task}}$ and $r^{\text{embod}}$—demonstrates substantial performance gains over OmniEVA-Base and naive RL training across all evaluated tasks. Where2Approach and Where2Fit exhibit accuracy surges of 28.95% and 34.28%, respectively, while Mobile Placement achieves notable success rate improvements of 43% (Easy) and 50% (Hard). Although each reward component provides individual benefits, their synergistic combination yields optimal results. These consistent advancements across primitive and composite tasks confirm that embodiment-aware reasoning significantly strengthens performance robustness in diverse embodied scenarios.

**Generalization Across Diverse Embodiments** To assess the adaptability of embodiment-aware reasoning, we conduct experiments varying a key robot parameter—arm length—as a representative embodiment factor. While other dynamic parameters warrant further exploration, arm length directly impacts workspace constraints and thus serves as a suitable initial testbed for evaluating physical generalization. We generate different embodiments by scaling the same base robot arm model, producing lengths from 72cm to 110cm. Models are trained on arm lengths of 75cm, 88cm, and 110cm, and tested on both seen and unseen configurations (72cm, 80cm, 84cm, 92cm, 96cm,

**Table 5: Results of Different Embodiment Execution Success Rate.**

| Models / Embodiments | Average (SR) | Seen Arm Length (cm) | | | | Unseen Arm Length (cm) | | | | | | | |
|---|---|---|---|---|---|---|---|---|---|---|---|---|---|
| | | 75 | 88 | 110 | Seen Avg | 72 | 80 | 84 | 92 | 96 | 100 | 105 | Unseen Avg |
| RoboBrain2.0-7B | 18.37 | 7.69 | 12.82 | 38.88 | 19.29 | 0 | 7.69 | 12.82 | 17.94 | 20.51 | 33.33 | 38.88 | 17.97 |
| **OmniEVA-Base** | 43.30 | 51.28 | 41.02 | 44.44 | 45.61 | 35.89 | 48.71 | 43.58 | 41.02 | 51.28 | 36.11 | 38.88 | 42.32 |
| **OmniEVA-ER w/o $r_{embod}$** | 65.35 | 69.23 | 71.79 | 69.44 | 70.17 | 41.02 | 64.10 | 66.66 | 61.53 | 69.23 | 72.22 | 69.44 | 63.29 |
| **OmniEVA-ER** | **81.89** | 87.17 | 82.05 | 86.11 | **85.08** | 56.41 | 87.17 | 84.61 | 79.48 | 82.05 | 88.88 | 86.11 | **80.52** |

105cm). The model is prompted with embodiment specifications (e.g., "The robot arm length is 72cm") to enable dynamic adaptation.

OmniEVA-ER consistently outperforms OmniEVA-Base and the ablated variant (without $r^{embod}$), achieving an average improvement of 15% on seen arm lengths. Moreover, it generalizes robustly to unseen lengths, attaining an 80.5% success rate—a 17% gain over the baseline. These results confirm that embodiment-aware reasoning facilitates not only task-level adaptation but also effective generalization across previously unseen physical configurations, highlighting its utility for real-world deployment in diverse robotic systems.

**Real World Deployment** To emphasize the generalizability of our approach, we conduct real-world experiments on two off-the-shelf wheeled mobile robots with inherently different dual 6-DoF arm designs: one platform has a 75 cm arm span, while the other (Galaxea R1) has a more constrained 70 cm arm span. As shown in Table 6, OmniEVA-ER demonstrates consistent performance improvements across both embodiments, particularly excelling in manipulation tasks. It achieves near-perfect performance (9/10) in Cluttered Place tasks and maintains robust navigation capabilities. The results confirm that embodiment-aware reasoning enhances real-world robustness, with OmniEVA-ER showing better adaptation to the more constrained 70cm arm span configuration.

**Table 6: Real World Performance on Different Tasks and Embodiments.** Both embodiments are wheeled mobile robots with dual 6-DoF arms. Embodiment 1 has 75cm arm span, Embodiment 2 has 70cm arm span.

| Models / Tasks | Cluttered Pick | | Cluttered Place | | Constrained Navigation | |
|---|---|---|---|---|---|---|
| | Embodiment 1 | Embodiment 2 | Embodiment 1 | Embodiment 2 | Embodiment 1 | Embodiment 2 |
| OmniEVA-Base | 6 / 10 | 5 / 10 | 6 / 10 | 5 / 10 | 9 / 10 | 8 / 10 |
| OmniEVA-ER | 8 / 10 | 7 / 10 | 9 / 10 | 9 / 10 | 8 / 10 | 8 / 10 |

## 5 LIMITATION AND FUTURE WORK

While OmniEVA demonstrates strong performance across various benchmarks, we acknowledge two key limitations. First, the scene-level gating mechanism in TAGR may lead to suboptimal 3D feature integration in heterogeneous environments, as evidenced by unexpected low activation for tasks involving spatial relationships in our activation analysis. Second, embodiment-aware planning may face challenges with unmodeled physical constraints, such as arm degrees of freedom or installation height, which could impact robustness in dynamic scenarios. For future work, we will explore patch-level gating strategies to enable finer-grained 3D adaptation and integrate broader physical parameters to enhance overall adaptability.

## 6 CONCLUSION

This work presents OmniEVA, a versatile embodied planner that delivers robust cross-dimensional reasoning across a wide range of tasks. By introducing a dynamic routing mechanism for 3D grounding, OmniEVA adapts effectively to varying spatial demands and improves reasoning performance. Its embodiment-aware fine-tuning strategy closes the gap between high-level reasoning and low-level robotic execution, enabling plans that are both logically sound and physically executable. Taken together, OmniEVA takes a strong step toward building general-purpose embodied agents that can reason, plan, and act in complex environments.

REPRODUCIBILITY STATEMENT

We provide implementation details in Appendix A, including model architecture and training configurations. Source code and benchmark datasets will be released upon acceptance.

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

APPENDIX

# A   IMPLEMENTATION DETAILS

## A.1   MODEL ARCHITECTURE AND TRAINING CONFIGURATIONS

Our experiments are built upon the pretrained InternVL3-8B model (Zhu et al., 2025), which serves as the foundational backbone for our multimodal large language models (MLLMs). To encode user instructions within the task-adaptive gated routing (TAGR) module, we utilize the all-MiniLM-L6-v2 (Reimers & Face, 2021), chosen for its efficiency and semantic fidelity. Instruction embeddings are further processed through a lightweight two-layer multilayer perceptron (MLP) with a hidden dimension of 256, enabling compact yet expressive representation learning.

For video-based inputs, we uniformly sample 16 frames during training and 32 frames during inference, striking a balance between temporal granularity and computational efficiency. To handle 3D spatial information, we voxelize both point clouds for positioning and 3D bounding boxes using a fixed voxel size of 0.1 meters. This discretization facilitates consistent spatial reasoning across diverse environments and tasks. Detailed hyper-parameters as given in Table 7

**TAGR Pretraining**   During TAGR pretraining, we freeze the sentence transformer and train the MLP encoder with a learning rate of $1e-4$. The ViT encoder is frozen, while the LLM backbone is updated with a reduced learning rate of $5e-7$ to prioritize learning within the TAGR module. We apply cosine decay for learning rate scheduling and use exponential decay to control the Gumbel-softmax temperature $\tau$, defined as $\tau_{\text{init}} \cdot \exp(\frac{-4.5 \cdot \text{steps}}{\text{max\_steps}})$.

**Supervised Finetuning**   For For supervised finetuning, the ViT encoder remains frozen. We finetune the LLM parameters using a learning rate of $1e-5$ with cosine decay.

**Reinforced Finetuning**   In the reinforcement stage, we continue finetuning the LLM with a learning rate of $1e-5$. Each training group generates 8 candidate outputs.

Complete hyperparameter settings for all training stages are provided in Table 7.

**Table 7:** OmniEVA Training Hyper-parameter Configuration

| Hyper-parameters | TAGR Pretraining | Supervised Finetuning | Reinforced Finetuning |
|---|---|---|---|
| epochs | 1 | 1 | 1 |
| batch size | 256 | 256 | 128 |
| learning rate (LLM) | 5e-7 | 1e-5 | 1e-5 |
| learning rate (TAGR) | 1e-4 | - | - |
| learning rate (ViTs) | - | - | - |
| lr schedular | cosine | cosine | - |
| $\tau_{\text{init}}$ (gumbel softmax) | 1.0 | 1e-6 | 1e-6 |
| $\tau_{\text{min}}$ (gumbel softmax) | 0.05 | 1e-6 | 1e-6 |
| $\tau$ schedular | $\tau_{\text{init}} \cdot \exp(\frac{-4.5 \cdot \text{steps}}{\text{max\_steps}})$ | - | - |
| weight decay | 0.1 | 0.1 | 0.1 |
| gradient clipping | 1.0 | 1.0 | 1.0 |
| use bf16 | true | true | true |
| use fp16 | false | false | false |
| warmup ratio | 1e-3 | 1e-3 | 0.0 |
| optimizer | AdamW | AdamW | AdamW |
| image resolution | 448×448 | 448×448 | 448×448 |
| video frames (training) | 16 | 16 | 16 |
| video frames (inference) | 32 | 32 | 32 |
| 3D voxel size | 0.1m | 0.1m | 0.1m |

## A.2   INPUT MODALITIES AND OUTPUT REPRESENTATIONS

OmniEVA is designed to accommodate a wide range of input modalities and output formats, enabling versatile interaction across visual and textual domains. Below, we detail the supported configurations.

### A.2.1 VISUAL INPUT MODALITIES

**Single Image**  Ideal for static or minimally dynamic environments, single-frame inputs support 2D spatial reasoning tasks such as object recognition, scene description, and basic grounding. This modality is particularly effective when temporal context is unnecessary and spatial relationships are confined to a single viewpoint.

**Multi-View Images or Video**  By aggregating information across multiple viewpoints or temporal frames, this modality facilitates both spatial and temporal reasoning. It is well-suited for dynamic or large-scale environments where understanding motion, continuity, or cross-frame object relationships is essential—such as in navigation, tracking, or multi-step manipulation tasks.

**RGB-D Video**  This modality integrates RGB visual data with depth information to reconstruct full 3D scene geometry. It is indispensable for tasks requiring occlusion-aware reasoning, volumetric understanding, or precise spatial manipulation. To enable accurate position embedding in world coordinates, users must provide the intrinsic camera matrix and corresponding extrinsic poses for each frame. These parameters allow the model to transform depth maps into structured 3D representations, forming the foundation for geometry-aware decision-making.

### A.2.2 TEXTUAL AND COORDINATE-BASED OUTPUTS

OmniEVA accommodates a range of textual and spatial formats for both input queries and output responses, enabling flexible interaction across semantic and geometric dimensions.

**Natural Language Queries and Responses**  Natural language serves as the primary interface for user interaction, supporting expressive queries and interpretable model responses. This format aligns with standard benchmarks in VQA and facilitates rich semantic engagement, allowing users to specify tasks in intuitive, human-readable form.

**2D Spatial Annotations**  For tasks such as 2D visual grounding and image captioning, inputs and outputs can be expressed using normalized pixel coordinates within the range [0, 1000]. This format enables precise object localization and descriptive annotation within a single image frame. Examples:

- **Question**: *Describe the object located at <point>(24, 312)</point>.*
  **Answer**: *It is a brown book next to a pencil.*

- **Question**: *Locate the apple on the left side of the book.*
  **Answer**: *<point>(122, 213)</point>.*

**3D Spatial Annotations**  For tasks involving 3D spatial reasoning—such as object captioning, grounding, and navigation—users may specify coordinates manually or allow the model to infer them from RGB-D inputs. Coordinates are discretized using a 0.1-meter grid to ensure consistency and precision across scenes.

- **Question**: *What is the object located at <3dbox>(61,217,26,5,7,3)</3dbox>?*
  **Answer**: *It is a brown wooden chair located at the center of the room.*

- **Question**: *Locate the second chair next to the table.*
  **Answer**: *<3dbox>(74,213,123,10,8,8)</3dbox>.*

This format empowers OmniEVA to reason about partially occluded objects and those outside the current frame, enabling robust interpretation and interaction within complex 3D environments.huozh

### A.3 IMPLEMENTATION DETAIL OF EMBODIMENT-AWARE REASONING

Given a reward for the $i$-th response:

$$r_{i,t}(q, o_i) = r_i^{\text{format}}(o_i) + r_{i,t}^{\text{acc}}(q, o_i) \tag{11}$$

For a group of responses with group size $G$, the normalized advantages of the $i$-th response at training step $t$ is calculated as:

$$A_{i,t} = \frac{r_{i,t} - \text{mean}(\{r_{0,t}, r_{1,t}, \cdots, r_{G,t}\})}{\text{std}(\{r_{0,t}, r_{1,t}, \cdots, r_{G,t}\})} \tag{12}$$

The final policy update objective is,

$$\mathcal{J}_t(\boldsymbol{\theta}) = \mathbb{E}\left[\frac{1}{G}\sum_{i=1}^{G}\left(\min\left(\frac{\pi_{\boldsymbol{\theta}}(o_i|q)}{\pi_{\theta_{\text{old}}}(o_i|q)}A_{i,t}, \text{clip}\left(\frac{\pi_{\boldsymbol{\theta}}(o_i|q)}{\pi_{\theta_{\text{old}}}(o_i|q)}, 1-\epsilon, 1+\epsilon\right)A_{i,t}\right) - \beta\mathbb{D}_{\text{KL}}(\pi_{\boldsymbol{\theta}}|\pi_{\theta_{\text{old}}})\right] \tag{13}$$

where $\beta$ is a regularization coefficient that restricts the deviation degree between the current policy $\pi_{\boldsymbol{\theta}}$ and the reference policy $\pi_{\theta_{\text{old}}}$ during optimization; $\epsilon$ is a positive coefficient that limits the magnitude of policy updates, preventing training instability caused by excessive updates. Through this embodiment-aware training pipeline, OmniEVA evolves from perceptual understanding to physically grounded execution, enabling generalizable planning and reliable performance across diverse real-world scenarios.

## A.4 MULTI-DIMENSIONAL DATA MIXED TRAINING

To enable mixed training of text-only, image-text, and RGB-D data within mini-batches, we preprocess the data to ensure uniform tensor dimensions (as described in Listing 1. For image-only samples, we add a 3D data placeholder. A binary depth flag associated with each sample indicates the presence of valid 3D data; during training, this flag acts as a zero mask to ignore the 3D component for samples that lack it. See Listing 2 for detail.

```python
def preprocess_sample(sample: dict) -> dict:
    # 1) Load and transform frames -> pixel_values: (T, C, H, W)
    frames = load_frames(sample)
    pixel_values = stack([transform(img) for img in frames])
    pixel_values = pixel_values.to(bfloat16)
    T, C, H, W = pixel_values.shape

    # 2) Process depth data based on sample type
    if sample.get("video-rgbd"):  # RGB-D video
        # Obtain 3D world coordinates from preprocessor
        world_coords = preprocessor.get_world_coords(sample)
        depth_flags = tensor([True] * T, dtype=bool)
    else:  # Pure RGB (image or video without depth)
        # Fill with zeros and mark as no-depth
        world_coords = zeros((T, H, W, 3), dtype=bfloat16)
        depth_flags = tensor([False] * T, dtype=bool)

    # 3) Return dictionary for model input
    return {
        "pixel_values": pixel_values,
        "world_coords": world_coords,
        "depth_flags": depth_flags,
        ...
    }
```

**Listing 1:** RGB-D and RGB Data Processing

```python
class OmniEVAModel(nn.Module):
    def __init__(self):
        super().__init__()
        self.vision_encoder = ...
        self.llm = ...
        self.mlp_proj = ...
        # the following modules are introduced by OmniEVA
        self.position_3d_encoder = ...
        self.lang_encoder = SentenceTransformer(...)
```

```python
        self.gate_module = ...

    def forward(self, pixel_values, input_ids, world_coords,
                depth_flags, lang_instr, **kwargs):
        # 1) Extract vision features and project to LLM space
        vit_feats = self.vision_encoder(pixel_values)  # (B*T, N, D_vit)
        vit_embeds = self.mlp_proj(vit_feats)          # (B*T, N, D_llm)

        # 2) Compute 3D positional encoding from world coordinates
        if world_coords is not None:
            pe3d = self.position_3d_encoder(world_coords)
        else:
            pe3d = 0

        # 3) Hard gating: pool vision features for gate control
        h = w = int(vit_embeds.shape[1] ** 0.5)
        # Reshape: (BT, N, D) -> (B, T, h, w, D)
        vit_pooled = vit_embeds.detach().view(B, T, h, w, -1)
        # Flatten spatial-temporal: (B, T, h, w, D) -> (B, T*h*w, D)
        vit_pooled = vit_pooled.flatten(start_dim=1, end_dim=3)
        # Average pool: (B, T*h*w, D) -> (B, D)
        vit_pooled = vit_pooled.mean(dim=1, keepdim=False)

        # Compute gate mask from language and pooled vision features
        lang_embeds = self.lang_encoder.encode(lang_instr)  # (B, D_lang)
        gate_logits = self.gate_module(lang_embeds, vit_pooled)  # (B, 2)
        gate_mask = gumbel_softmax(gate_logits, tau=tau, hard=True)[:, 1]
        gate_mask = gate_mask.view(B, 1, 1).expand(B, T, 1).view(BT, 1,
    ↪ 1)

        # 4) Apply depth flag mask
        depth_mask = depth_flags.view(BT, 1, 1)
        vit_embeds = vit_embeds + pe3d * gate_mask * depth_mask

        # 5) Merge vision tokens into language embeddings
        input_embeds = self.llm.get_input_embeddings()(input_ids)
        img_token_mask = (input_ids == self.img_context_token_id)
        input_embeds[img_token_mask] = vit_embeds.flatten(0, 1)

        # 6) Forward through LLM
        outputs = self.llm(inputs_embeds=input_embeds, **kwargs)

        # 7) Compute auxiliary gating loss (KL divergence with soft
    ↪ labels)
        if self.training:
            soft_labels = get_soft_labels()
            aux_loss = kl_div(log_softmax(gate_logits), soft_labels)
            return outputs, aux_loss

        # 8) Replace placeholders in input_tokens with visual tokens.
        ...
```

Listing 2: Forward Pass of OmniEVA

## A.5 IMPLEMENTATION DETAIL OF EMBODIMENT-AWARE REINFORCED FINETUNING

# B CASE STUDY

## B.1 CASE STUDY OF THE ACTIVATION STATE OF THE TAGR MODULE

We illustrate this behavior through qualitative case studies (Figure 6). In the first two examples, querying the shape of a table and a desk activates the 3D gate with differing probabilities: 0.73 for

---

**Algorithm 1** Embodied Task Learning with Embodiment Constraints

---

1: **Input:** Task parameters $\theta$, Embodiment constraints $\mathcal{E}$
2: **Output:** Optimized policy parameters $\theta^*$
3: Initialize policy network $\pi_\theta$
4: Initialize task reward model $R_{\text{task}}$
5: Initialize embodiment constraint handler $R_{\text{embod}}$
6: **for** episode = 1 to $N$ **do**
7:     Reset environment and get initial observation $o_0$
8:     Initialize episode trajectory $\tau \leftarrow \emptyset$
9:     **for** $t = 0$ to $T$ **do**
10:         Sample action $a_t \sim \pi_\theta(\cdot|o_t)$
11:         Execute action and get next observation $o_{t+1}$
12:         **Task Reward Computation:**
13:         Given instruction $q$ and observation $o_t$
14:         $r_{\text{task}} \leftarrow \text{CVModel}(q, o_t) + \text{RuleBasedMethod}(q, o_t)$
15:         Get corresponding mask $M_t$ for RGB image $o_t$
16:         $r_{\text{task}} \leftarrow \mathbb{I}[\text{MaskIndicatesSuccess}(M_t)]$ {1 if success, 0 otherwise}
17:         **Embodiment Constraint Reward:**
18:         Inflate objects in $r_{\text{task}}^i$: $M_{\text{inflated}} \leftarrow \text{InflateObjects}(M_t)$
19:         Compute intersection $r_{\text{inter}}^i$: $M_{\text{intersect}} \leftarrow M_t \cap M_{\text{inflated}}$
20:         Random sample points from $M_{\text{intersect}}$: $P \leftarrow \text{RandomSample}(M_{\text{intersect}})$
21:         Initialize $r_{\text{embod}}^i \leftarrow r_{\text{inter}}^i$ {Start with original task reward}
22:         **for** each point $p_j \in P$ **do**
23:             Execute MoP at $p_j$
24:             **if** execution failed **then**
25:                 $r_{\text{embod}}^i \leftarrow 0$ {Set to 0 if any point fails}
26:                 **break**
27:             **end if**
28:         **end for**
29:         **Accuracy Reward:**
30:         $r_{\text{acc}} \leftarrow r_{\text{task}} \cdot (\lambda_t \cdot r_{\text{embod}} + (1 - \lambda_t))$
31:         Store transition $(o_t, a_t, r_{\text{acc}}, o_{t+1})$ in $\tau$
32:     **end for**
33:     **Policy Optimization:**
34:     Update $\theta$ using GRPO with trajectory $\tau$
35: **end for**
36: **return** Optimized parameters $\theta^*$

---

the rectangular table, indicating ambiguity between "*square*" and "*rectangular*" and thus a reliance on 3D cues; and 0.52 for the round table, suggesting sufficient 2D visual information. In contrast, object counting and color identification in the two right-hand examples leave the 3D gate inactive, demonstrating the TAGR module's ability to omit 3D features when spatial reasoning is unnecessary.

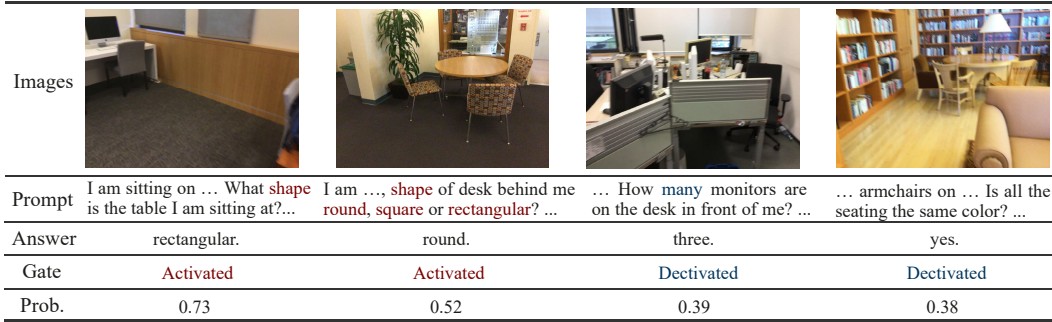

| | | | | |
|---|---|---|---|---|
| Images | | | | |
| Prompt | I am sitting on … What shape is the table I am sitting at?... | I am …, shape of desk behind me round, square or rectangular? ... | … How many monitors are on the desk in front of me? ... | … armchairs on … Is all the seating the same color? ... |
| Answer | rectangular. | round. | three. | yes. |
| Gate | Activated | Activated | Dectivated | Dectivated |
| Prob. | 0.73 | 0.52 | 0.39 | 0.38 |

**Figure 6: Case Study of Gate Activation State**. Selected examples from the validation dataset illustrate the most prominently activated and deactivated words within the input prompts, highlighting the model's sensitivity to specific language cues.

## B.2    CASE STUDY OF EMBODIMENT-AWARE REASONING

We compare outputs from models trained with and without it for the same task, as shown in Figure 7. Results show that both models effectively identify unoccupied areas on the tabletop, indicating satisfactory task-level performance. However, the placement locations proposed by the model without TE-GRPO training frequently fall outside the operational range of the robotic arm or exhibit suboptimal execution efficiency. In contrast, OmniEVA trained with TE-GRPO consistently identifies vacant areas that are both feasible and executable, demonstrating enhanced alignment with physical and task constraints.

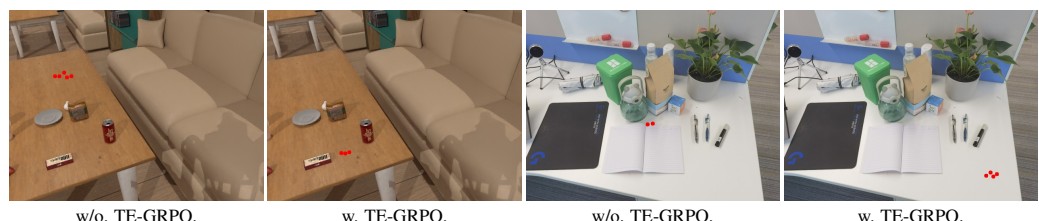

w/o. TE-GRPO.          w. TE-GRPO.          w/o. TE-GRPO.          w. TE-GRPO.

**Figure 7:** Comparison of responses w/o and w embodiment-aware reasoning.

## B.3    ROBUSTNESS IN DYNAMIC ENVIRONMENTS

While compatible with existing temporal embedding techniques like MRoPE used in Qwen2.5VL, OmniEVA's architecture exhibits strong emergent robustness in dynamic environments, despite not being specifically designed for them. This capability is evaluated through scenarios involving moving pedestrians, as shown in Figure 8. In the first experiment, the model is tasked with locating a table opposite to the fridge that is initially occluded by a pedestrian. After a change in the agent's viewpoint causes the pedestrian to move away, OmniEVA successfully identifies the newly revealed table. In a more challenging second experiment, the model accurately detects a microwave despite a pedestrian continuously moving within the field of view. These results demonstrate that OmniEVA's design inherently handles the visual uncertainty posed by both transient and persistent dynamic obstructions.

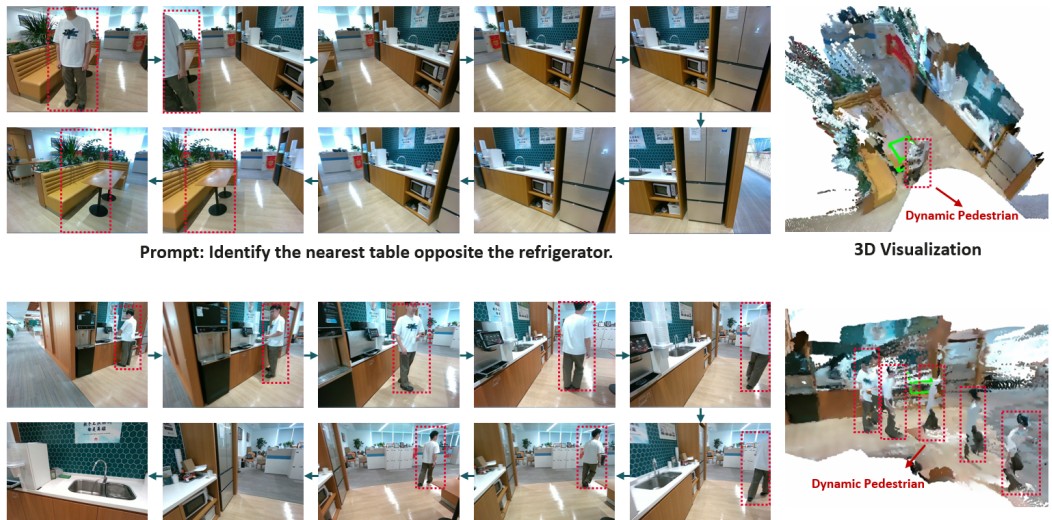

**Figure 8: OmniEVA's Performance with Dynamic Obstructions.** (Top) Sequence showing successful detection of a table after an occluding pedestrian moves away. (Bottom) Accurate detection of a microwave despite continuous pedestrian motion.

### B.4 HOW DOES OMNIEVA ADAPT TO PHYSICAL AND EMBODIMENT CONSTRAINTS?

To further understand the model's embodiment awareness, we analyze the output of OmniEVA-ER trained with TE-GRPO. As illustrated in Figure 9, OmniEVA-ER initiates reasoning with task-level analysis, including scenario interpretation, commonsense spatial analysis, and textual localization of vacant areas. It then incorporates physical constraints—such as reachability and workspace boundaries—to refine its predictions. Through such embodiment-aware slow thinking processes and reward reinforcement, the final output is a placement region that satisfies all constraints. A comparative study (Appendix B.2) highlights the impact of embodiment-aware reasoning. For real-world validation, OmniEVA is deployed on a wheeled dual-arm robot, demonstrating reliable performance converting user instructions into executable plans across diverse physical settings (Appendix G).

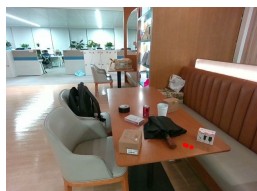

**Prompt**: *Locate some free space for me, please.*
**Response**: *<think>1. Describing the Image: a table with various objects, including a cup ... 2. Common-Sense Considerations: Free space refers to areas on the table that are not occupied ... 3. Relevant Factors: Object Dimensions: ... Table Dimensions: ... 4. Embodiment Awareness: For optimal performance, we should place the bottom of the table near the bottom of the table to ensure it can access its entire intended work area. 5. Answering the Question: free space on the right side of the table near the edge... </think><answer>Locations are <point>(450, 346), (455, 348), (467, 352) </point>.</answer>*

**Figure 9:** Case study illustrating OmniEVA's reasoning process under embodiment-aware constraints.

## C ABLATION STUDY IMPLEMENTATION DETAILS

### C.1 IMPLEMENTATION OF CROSS-ATTENTION BASED 3D FUSION

To rigorously evaluate the necessity of our proposed gated router, we implemented two variants of cross-attention-based 3D fusion as strong baselines. The objective was to assess whether standard attention mechanisms could achieve adaptive feature fusion without an explicit gating module. The architectural details are illustrated in Figure 10.

- **Separate Tokens Arrangement:** In this variant, the sequences of visual tokens ($V^I$) and 3D positional encoding tokens ($V^P$) are kept entirely separate. They are concatenated

along the sequence dimension, effectively doubling the input length presented to the transformer backbone. A standard cross-attention layer is then employed to enable interaction between these two modalities.

- **Interleaved Tokens Arrangement:** In this variant, tokens are grouped and sequenced by the source image. Specifically, for a batch of $N$ input images, the token sequence is structured as: $[V_1^P, V_1^I, V_2^P, V_2^I, ..., V_N^P, V_N^I]$. That is, for each image, all its 3D tokens are placed first, followed by all its visual tokens, and this pattern repeats for the next image. This arrangement maintains the total sequence length but structures it to preserve intra-image modality groupings, relying on the self-attention mechanism to learn the cross-modal dependencies.

As discussed in Section 4.2, both cross-attention variants led to significant performance drops compared to our gated fusion. We attribute this to the inefficient learning burden placed on the attention mechanism—it must either handle a doubled sequence length (Separate) or learn to associate interleaved modality-specific groups (Interleaved) from scratch. In contrast, our TAGR module performs explicit, lightweight routing before the backbone, leading to more stable and efficient integration.

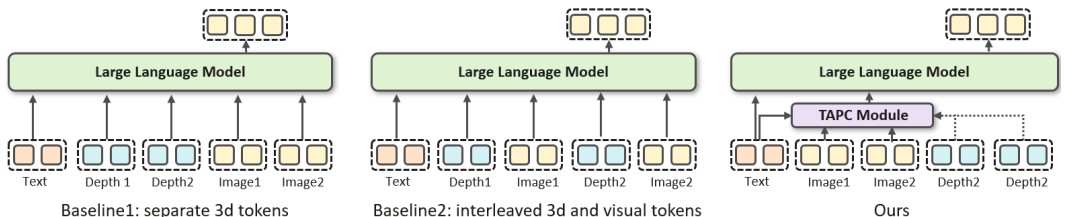

**Figure 10:** Architectural diagrams of the cross-attention-based 3D fusion baselines. (a) **Separate Tokens**: Visual and 3D tokens are processed as separate sequences. (b) **Interleaved Tokens**: Tokens are sequenced by grouping all 3D tokens and all visual tokens for each image consecutively. Both approaches introduce learning inefficiencies absent in our gated fusion.

## C.2 Gate Activation Analysis by Semantic Clustering

We introduce a more robust analysis based on semantic clustering of the entire prompt. This method leverages a language model encoder (e.g., a lightweight Sentence Transformer) to capture the overall meaning of a prompt before categorizing it. The detailed procedure is outlined in Algorithm 2. The example prompts used for clustering are listed in Table 8.

## D Training Dataset

### D.1 Dataset Overview

Figure 11 presents a comprehensive breakdown of the training dataset utilized by OmniEVA. Designed to support omni-multimodal and cross-dimensional reasoning, the dataset integrates three major categories: general data, image-based reasoning data, and 3D reasoning data. This diverse composition enables OmniEVA to develop robust capabilities across a wide spectrum of tasks, from basic object recognition to complex spatial and semantic understanding.

In total, the dataset comprises approximately 5.2 million samples. Detailed descriptions and distributions of each data category are provided in the subsequent sections.

### D.2 Embodied Reasoning Data

**General Visual Question Answering** To maintain the foundational visual reasoning capabilities and generalization strength of the vision-language model (VLM), we integrated a diverse set of general visual question answering (VQA) datasets. Notably, **LLaVA-665K** (Liu et al., 2023) contributes a broad spectrum of tasks, including VQA, optical character recognition (OCR), region localization, and instruction-following. To further enrich the dataset, we incorporated academic datasets such as GQA (Hudson & Manning, 2019), OKVQA (Marino et al., 2019), and A-OKVQA (Schwenk et al.,

---

**Algorithm 2** Semantic-based Gate Activation Analysis

---

**Require:** Prompts $P = \{p_1, p_2, ..., p_n\}$, Gate logits $G = \{g_1, g_2, ..., g_n\}$
**Require:** Semantic templates $T = \{T_1, T_2, ..., T_k\}$ for $k$ categories
**Require:** Similarity thresholds $\tau_{high} = 0.4, \tau_{low} = 0.25$
 1: **// Step 1: Encode templates**
 2: **for** each category $c \in \{1, ..., k\}$ **do**
 3:    $E_c \leftarrow$ SentenceTransformer$(T_c)$ {Encode templates}
 4: **end for**
 5: **// Step 2: Batch encode all prompts**
 6: $E_P \leftarrow$ SentenceTransformer$(P)$ {Shape: $n \times d$}
 7: **// Step 3: Vectorized similarity computation**
 8: **for** each category $c \in \{1, ..., k\}$ **do**
 9:    $S_c \leftarrow$ CosineSimilarity$(E_P, E_c)$ {Shape: $n \times |T_c|$}
10:    $M_c \leftarrow \max(S_c, \text{axis} = 1)$ {Max similarity per prompt}
11: **end for**
12: $M \leftarrow [M_1, M_2, ..., M_k]$ {Similarity matrix: $n \times k$}
13: **// Step 4: Assign categories**
14: **for** each prompt $i \in \{1, ..., n\}$ **do**
15:    $C_i \leftarrow \{c \mid M[i, c] > \tau_{high}\}$ {High confidence match}
16:    **if** $C_i = \emptyset$ **then**
17:      $c^* \leftarrow \arg\max_c M[i, c]$
18:      $C_i \leftarrow \{c^*\}$ **if** $M[i, c^*] > \tau_{low}$ **else** {other}
19:    **end if**
20: **end for**
21: **// Step 5: Compute activation statistics**
22: **for** each prompt $i$ with categories $C_i$ **do**
23:    is_activated$_i \leftarrow (g_i[1] > g_i[0])$ {Binary gate decision}
24:    **for** each category $c \in C_i$ **do**
25:      **if** is_activated$_i$ **then**
26:        $\text{Count}_{\text{active}}[c] \leftarrow \text{Count}_{\text{active}}[c] + 1$
27:      **else**
28:        $\text{Count}_{\text{inactive}}[c] \leftarrow \text{Count}_{\text{inactive}}[c] + 1$
29:      **end if**
30:    **end for**
31: **end for**
32: **// Step 6: Compute activation rates**
33: **for** each category $c$ **do**
34:    $\text{Rate}_c \leftarrow \frac{\text{Count}_{\text{active}}[c]}{\text{Count}_{\text{active}}[c] + \text{Count}_{\text{inactive}}[c]}$
35: **end for**
36: **return** $\{\text{Rate}_c\}_{c=1}^k$

---

2022), which emphasize compositional reasoning and external knowledge grounding. OCR-focused datasets like OCR-VQA (Mishra et al., 2019), TextVQA (Singh et al., 2019), and ScienceQA (Saikh et al., 2022) enhance the model's ability to interpret embedded textual content. Additionally, region-level vision-language understanding is supported through RefCOCO (Yu et al., 2016) and Visual Genome (Krishna et al., 2017), which provide fine-grained spatial and semantic annotations. To bolster language fluency and multimodal dialogue coherence, we also included approximately 40,000 pure text instruction samples from ShareGPT (Chen et al., 2024b).

**2D Visual Grounding** To endow OmniEVA with robust object detection and geometric localization capabilities, we incorporated the **LVIS** (Gupta et al., 2019) dataset—a comprehensive benchmark for large-vocabulary instance segmentation. LVIS offers approximately 164,000 images annotated with nearly 2 million high-quality segmentation masks, spanning over 1,000 entry-level object categories such as "chair" and "plate." Its rich diversity and precise annotations make it an ideal foundation for training models in spatially grounded object recognition. We utilize 140K samples from LVIS to train the 2D visual grounding module.

**Table 8:** Semantic Category Templates for Gate Activation Analysis

| Category | Template Examples | Category | Template Examples |
|---|---|---|---|
| **Shape** | What shape is it? Square, rectangular, round Cylindrical, spherical Triangular, oval | **Spatial Location** | Where is the object? Position and location Left, right, above, below |
| **State Condition** | Is it open or closed? The current state Empty, full, broken | **Appearance** | What does it look like? Describe appearance Visual characteristics |
| **Counting** | How many objects? Count the number Several items present | **Identification** | What is this object? Identify the item What type of thing |
| **Spatial Relation** | What is next to it? Between which objects Closest to what | **Direction** | Which direction facing? The orientation Pointing towards |
| **Material Texture** | What is it made of? Wood, metal, plastic Smooth, rough surface | **Visibility** | Can you see it clearly? Visible or hidden? Partially blocked |
| **Color** | What color is it? The color of object Red, blue, green | **Existence** | Is there any object? Does it exist? Something present or not |
| **Semantic Cat.** | What category? Type of furniture Classification of objects | **Distance Depth** | How far is it? Distance from camera Near or far away |
| **Action Activity** | What is it doing? The action or activity Walk, run, throw, placing | **Other** | Uncategorized or low-similarity prompts (sim. $< 0.25$) |

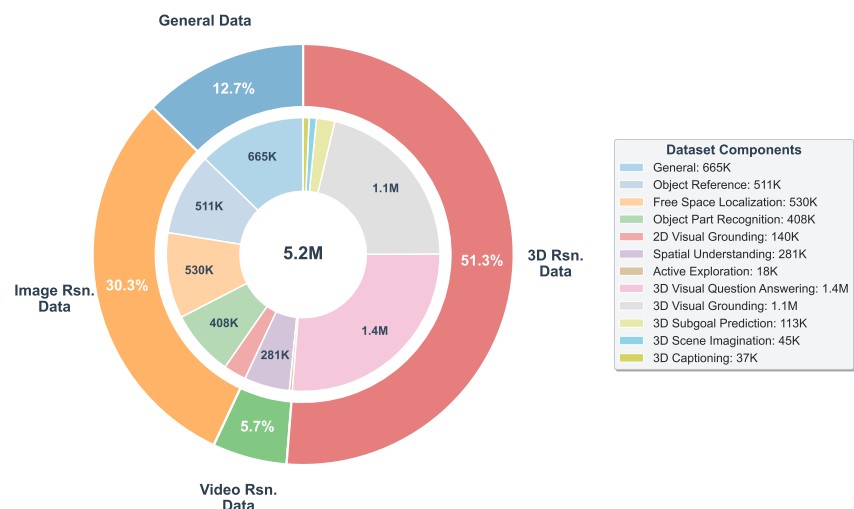

**Figure 11: Overview of the Training Data used by OmniEVA**

**Object Reference**   Object reference data plays a crucial role in enabling OmniEVA to associate linguistic instructions with specific visual regions. We curated three complementary datasets to support this capability. (1) **Osprey-724K** (Yuan et al., 2024b) is a large-scale instruction tuning dataset specifically constructed to achieve pixel-level vision-language alignment. Designed to overcome the limitations of traditional multimodal models—which primarily operate at the image-level or bounding box-level—it incorporates regional masks linked with precise language descriptions to enhance

model performance in fine-grained visual understanding tasks. (2) **Robopoint Object Reference-347K** (Yuan et al., 2024a) involves locating keypoints within a given reference object in an image. For example: "In the image, an object is marked with a red box. Please indicate several points located in the area below this object." The model's response would be in the form of normalized coordinates such as [(0.56, 0.69), ...]. Such data helps the model learn to accurately identify target positions that have spatial relationships with reference objects, making it suitable for applications like robotic grasping or object association scenarios. (3) **RoboRefIt** (Lu et al., 2023) specifically designed for visual grounding tasks in robot interaction. It aims to enhance robots' ability to recognize and locate target objects based on language instructions in real-world scenarios. The dataset comprises 10,872 real RGB-D images collected from cluttered indoor environments in daily life. Each image is annotated with referring expressions (instruction sentences), totaling 50,758 entries, which describe object features or locations in a robot-oriented language style. Approximately half of the images contain similar or distracting objects, increasing recognition difficulty to simulate challenges in real-world grasping scenarios. Together, these datasets provide 511K training samples for object reference grounding.

**Object Part Recognition** Part recognition data is primarily used to visually highlight specific functional parts of objects in images, as referred to by corresponding linguistic instructions. These datasets include: (1) **AGD20K** (Luo et al., 2022) is constructed by collecting and labeling over 20K images from 36 affordance categories, such as sit on, type on, and drink etc. Affordance grounding aims to locate objects' "action possibilities" regions, an essential step toward embodied intelligence. (2) **HANDAL** (Guo et al., 2023) is used for category-level object pose estimation and affordance prediction. Unlike previous datasets, it is focused on robotics-ready manipulable objects that are of the proper size and shape for functional grasping by robot manipulators, such as pliers, utensils, and screwdrivers. The dataset consists of 308k annotated image frames from 2.2k videos of 212 real-world objects in 17 categories. It focus on hardware and kitchen tool objects to facilitate research in practical scenarios in which a robot manipulator needs to interact with the environment beyond simple pushing or indiscriminate grasping. (3) **PACO** (Ramanathan et al., 2023) is a large-scale dataset constructed for fine-grained image understanding tasks, designed to support object- and part-level instance segmentation as well as attribute recognition. It contains 57,643 images with 1,644,461 annotated object instances spanning 270 distinct categories (e.g., "body", "rim", "handle"). The dataset can be used for object detection, semantic segmentation, and instance segmentation tasks, and supports conversion from instance masks to semantic masks or bounding boxes for diverse downstream applications. In total, we utilize 408K samples from these datasets.

**Free Space Location** Free space location data is primarily used to visually mark vacant placement areas in images, as indicated by corresponding linguistic instructions. These datasets include: (1) **Robopoint Free Space Reference-320K** (Yuan et al., 2024a) In this dataset, the language instructions require the model to identify keypoints in free space near a reference object—despite the absence of clear visual cues. For example: "Indicate several points in the empty space to the left of the pizza box." This type of data enables the model to understand "where is a suitable region to perform an action, even if no object is visually present," making it highly applicable to robotic navigation or assisted placement tasks. (2) **RefSpatial 3D Vacant** (Zhou et al., 2025) RefSpatial is a large-scale dataset created to support Visual Language Models (VLMs) in performing 3D multi-step reasoning for spatial referencing tasks. It aims to enhance models' ability to understand complex spatial instructions in real-world environments. The dataset contains approximately 20 million (20M) question-answer (QA) pairs, covering 31 spatial relation categories, and supports multi-step reasoning of up to 5 steps. It includes locate empty space: define a point in an empty area on a surface based on its spatial relationships with surrounding objects, and ask to confirm this empty location (e.g., "Please provide a point in the vacant area on the desktop that simultaneously satisfies the following spatial conditions: ..."). (3) **Open-X-Embodiment** (O'Neill et al., 2024) In current datasets involving actions performed by robots/robotic arms, skills typically include "pick," "pick and place," and more. We extract keyframes and manipulated objects from the corresponding trajectory data to construct object placement data. For example: **RT-1** (Brohan et al., 2022) dataset comprises over 130,000 real-world robotic demonstrations (episodes), covering more than 700 different tasks. These were collected by 13 robots over a period of 17 months. The trained actions include diverse skills such as grasping/placing objects, opening/closing drawers, extracting items, standing objects upright, knocking them over, pulling tissues, and opening jars. **BridgeData**

**V2** (Walke et al., 2023) dataset contains 60,096 trajectories spanning 24 different environments, including toy kitchens, sinks, microwaves, desktops, washing machines, toolboxes, and other diverse settings. It encompasses 13 types of manipulation skills, ranging from basic pick-and-place, pushing/pulling, and sweeping, to more complex tasks such as stacking blocks, folding cloth, and manipulating granular media. Together, these datasets provide 530K training samples.

**Video-based Spatial Reasoning** Despite the emergence of benchmarks such as OpenEQA (Majumdar et al., 2024) and VSI-Bench (Yang et al., 2025b), large-scale training datasets for video-based spatial reasoning remain limited. To bridge this gap, we construct a comprehensive dataset by harnessing high-fidelity indoor scene sources from ScanNet (Dai et al., 2017), Matterport3D (Chang et al., 2017), and 3RScan (Wald et al., 2019). From these sources, we extract egocentric video sequences and generate question–answer pairs aligned with the task taxonomy defined in VSI-Bench (Yang et al., 2025b), encompassing: (1) object count, (2) relative distance, (3) appearance order, (4) relative direction, (5) object size, (6) absolute distance, (7) room size, and (8) route planning. Each QA pair is produced through a hybrid pipeline combining automated template generation with manual verification to ensure spatial coherence and semantic precision. For route planning tasks, we first convert point clouds into x-y navigation mesh maps. Navigable waypoints are selected based on three independently defined anchors: the start object, its orientation, and the end object. Using the A* algorithm, we compute the shortest path while merging trajectory points of adjacent objects belonging to the same entity. Steering directions are then derived from angular changes along the path, enabling fine-grained spatial reasoning. We use 281K samples for training.

**Active Exploration** Prior approaches typically assume fully observable environments, limiting their applicability to real-world scenarios. We propose a novel task that enhances spatial reasoning under partial observability. Given multiple images from an indoor scene, the model must select the most informative view to locate a specified object. For instance: "From the provided visual input, identify the most informative image frame that offers the best chance of locating the bed. Format your response as: Frame ID: [Selected Frame ID]." This task strengthens the model's decision-making in incomplete environments and is crucial for downstream embodied tasks such as object navigation. To support this, we curated 18K training samples from HM3D (Ramakrishnan et al., 2021) and MP3D (Chang et al., 2017), covering 6 and 21 object categories respectively.

**3D Visual Question Answering** To advance spatial reasoning in 3D environments, we integrate three complementary datasets, each contributing unique challenges and perspectives. (1) SQA3D (Azuma et al., 2022): This dataset emphasizes situational awareness, requiring agents to interpret their position, orientation, and context within a 3D scene before answering questions. It simulates real-world embodied cognition, where understanding one's spatial state is a prerequisite for reasoning. We collected approximately 79K samples, covering diverse indoor layouts and object configurations. (2) ScanQA (Ma et al., 2022): Focused on general spatial understanding, ScanQA includes questions about object alignment, relative direction, and localization. It challenges models to parse nuanced spatial relationships from textual queries and visual cues. Our training set includes 23K samples, offering a rich variety of spatial scenarios. (3) MMScan-QA (Lyu et al., 2024): As the largest and most comprehensive resource, MMScan provides over 1.28M QA samples built on ScanNet (Dai et al., 2017), Matterport3D (Chang et al., 2017), 3RScan (Wald et al., 2019), and Arkitscenes (Baruch et al., 2021). It features hierarchical grounded language annotations spanning object-level and region-level semantics, enabling multi-granular reasoning. In total, we utilize approximately 1.4M samples for 3D VQA training.

**3D Captioning** In this task, the model generates descriptive captions for objects given a 3D position or bounding box, detailing attributes such as color, shape, and spatial relations. For instance, *"Question: Describe the object located at (155,72,23,15,13,3). Answer: It is a light brown, wooden chair, located in front of a white table in the room."* This task bridges geometric localization with natural language generation. We train on the Scan2Cap dataset (Chen et al., 2021), which comprises 37K annotated samples across diverse indoor scenes.

**3D Visual Grounding** As the inverse of 3D captioning, this task requires the model to localize objects in 3D space based on natural language descriptions. It poses a significant challenge for MLLMs, which often struggle to generate accurate 3D bounding boxes without priors from off-the-

shelf 2D or 3D detectors. For example, *"Question: Detect the bounding box of a chair in the corner of the room, opposite to a brown desk. Answer: (78, 23, 135, 5, 5, 7)"*. We leverage ScanRefer (Chen et al., 2020) and MMScan-VG (Lyu et al., 2024), totaling 1.1M samples.

**3D Scene Imagination**  To push the boundaries of spatial reasoning, we introduce a task set in partially observable environments—where some objects are occluded or outside the agent's field of view. The model must infer the contents of unobserved regions based on contextual cues and spatial layout, given a 3D location within the scenario. For example, *"Question: Based on the currently observed environment, when the agent walks to position (384, 42, 15), what new objects might become visible? Only consider objects not currently seen. Answer: You may see various cookers, cabinets, kitchen counters, kettles, ..."*. This task probes the model's understanding of object co-occurrence and spatial regularities. We collect 45K samples using the Habitat simulator (Puig et al., 2023), drawing from MP3D (Chang et al., 2017) and HM3D (Ramakrishnan et al., 2021) with a randomized walk policy to ensure diverse and unbiased scene coverage.

**3D Subgoal Prediction**  Existing spatial reasoning methods for large-scale navigation—image-based pointing Yuan et al. (2024a), marker selection, or direct command outputs (e.g., "move forward" or "turn left") —often struggle in complex, occlusion-heavy scenes under partial observability. To overcome these limitations, we introduce a 3D-aware planning framework that ingests sequential RGB-D observations and directly generates subgoals in continuous 3D coordinate space. By formulating intermediate objectives as 3D waypoints, the model leverages the geometric structure of the environment, avoids local optima caused by relying on single image views, and supports explicit long-term trajectory planning rather than making only myopic action predictions. Moreover, it accounts for occluded or unseen regions, enabling the agent to propose subgoals that guide exploration around obstacles and through partially observed areas. Our training set includes approximately 113K samples, supporting robust learning of spatial planning under uncertainty.

## E  BENCHMARKS FOR EVALUATION

### E.1  EMBODIED REASONING BENCHMARKS

**Embodied Reasoning Benchmarks with 2D Inputs**  To assess the model's embodied reasoning capabilities across visual modalities, we employ four established benchmarks: **Where2Place** (Yuan et al., 2024a), **VSI-bench** (Yang et al., 2025b), **PACO-LVIS** (Ramanathan et al., 2023), and **RoboRefit** (Lu et al., 2023). [2] These datasets span both static images and dynamic video inputs, enabling comprehensive evaluation of spatial and temporal understanding and multimodal reasoning.

To further evaluate the model's capacities in versatile embodied tasks with physical constraints, we introduce four benchmarks that connect the primitive embodied capabilities with composite down-streamed tasks: **Where2Go**, **Where2Fit**, and **Where2Approach**, **Where2Grasp**. Compared to simulator-based online evaluation, this VQA-style approach substantially reduces evaluation overhead. Detailed examples and description can be found in Appendix E.3.

- **Where2Go:**  The agent must select the most informative next view from multiple images to locate a target object in *partially observable* environment. The setting closely aligns with the **Large Space Object Seeking** tasks, where agents must infer spatial layouts and make decisions under uncertainty.

- **Where2Fit:**  The agent must identify the free space on the table by predicting a set of 2D points. Physical constraints, including object location, size, collision potential, must be considered, making this task highly relevant to the **Mobile Placement (Easy)** tasks.

- **Where2Approach:** The agent must identify free space on the table that is not obstructed by any chairs. This task demands reasoning under occlusion as well as handling locomotion and manipulation constraints, making it closely aligned with the **Mobile Placement (Hard)** tasks in geometrically challenging scenarios.

---

[2]Since the original data annotations of RoboRefit and PACO-LVIS lack VQA pairs, we constructed a minimal evaluation set suitable for VLM based on image distribution, object category, part category, etc. The evaluation code is consistent with Where2Place.

- **Where2Grasp:** The agent must identify objects based on their color, size, location, and category. This task emphasizes object-centric recognition and directly aligns with the requirements of the **Mobile Pick-up** tasks.

The relationship between primitive embodied capabilities and composite downstream tasks is demonstrated in Section 4.3 .

**Embodied Reasoning Benchmarks with 3D Inputs** To extend evaluation into three-dimensional spatial contexts, we adopt four 3D benchmarks: (Ma et al., 2022), **ScanQA** (Azuma et al., 2022), **Scan2Cap** (Chen et al., 2021) and **ScanRefer** (Chen et al., 2020). These datasets challenge the model's capacity for open-ended question answering, scene captioning, and 3D visual grounding within richly structured 3D environments. By incorporating depth and geometry, they serve as critical tests of the model's ability to reason beyond planar representations.

## E.2 END-TO-END ONLINE EVALUATION WITHIN SIMULATORS

While previous works often evaluate the performance of the MLLMs on offline dataset, we also perform end-to-end evaluation to bridge the gap between planning and robot execution within simulators, on the following 3 introduced benchmarks. The benchmark is built based on a $3000m^2$ office environment containing 8 core operation scenarios and 95 object categories representative of common workplace items. We categorize the benchmark into three progressive evaluation stages:

- **Large-Space Object Seeking**: It is also referred as object navigation in prior work. This task evaluates the agent's capability to locate a given object in large space.

- **Local Mobile Manipulation**: This evaluation set comprises over 30 representative scenarios featuring diverse background configurations, varing initial robot poses, and a range of object types, sizes, and locations. The **Mobile Pick-up** task involves grasphing various objects across diverse scenes and tabletop configurations. The **Moile Placement** is divided into two difficulty tiers based on environment complexity. In the *easy* tier, the robot only needs to consider the immediate table surface condition (e.g., object occlusion) to determine the optimal placement location, as done in **Where2Fit**, before placing the object. For the *hard* tier tasks, the robot must fist determine the optimal chassis poses while accounting for environmental constraints imposed by the spatial arrangements of tabletop objects and surrounding chairs (same setting as **Where2Approach**). The evaluation involves navigating to target poses, followed by assessing trajectory planning for safe mug placement on the table, with success rates calculated based on task completion accuracy. A comprehensive description of scenario design and task categorization is provided in AppendixE.4.

- **End-to-End Delivery**: This task evaluates the integration of embodied skills by requiring the robot to complete end-to-end object-delivery tasks across the entire office environment. We select two metrics, the overall success rates and the average task completion times, to evaluate the effectiveness of the pipeline.

## E.3 EXAMPLES OF THE IN-HOUSE PRIMITIVE EMBODIED BENCHMARKS

### E.3.1 WHERE2GO

The Where2Go benchmark is constructed using the validation splits of the HM3D (Chang et al., 2017) and MP3D (Chang et al., 2017) datasets. Each sample presents a partially observable environment in which the model must select the most informative view to locate a specified target object. A frame is designated as the ground truth if it contains a visible segment of the shortest navigable path from the agent's current position to the target object. In total, the benchmark comprises 207 samples, forming a diverse and challenging validation set for evaluating view selection under uncertainty.

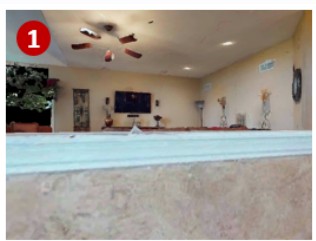 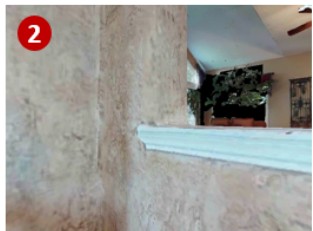 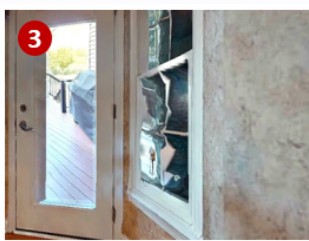

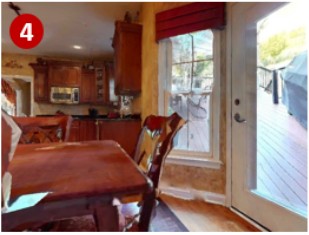 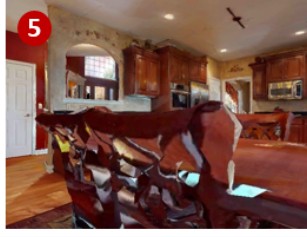 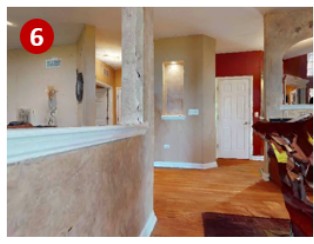

**Prompt**: *From the provided visual input, identify the most informative image frame (with IDs starting from 1) that offers the best chance of locating the sofa. Format your response as: Frame ID: [Selected Frame ID]*
**Ground Truth**: *Frame ID: 5, 6*

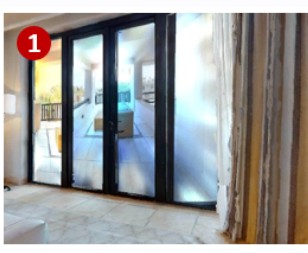 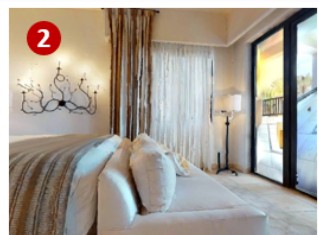 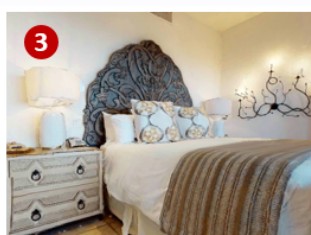

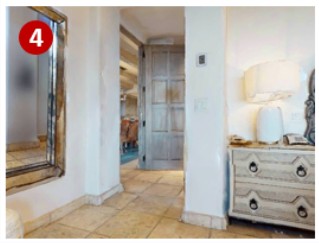 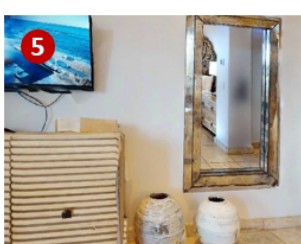 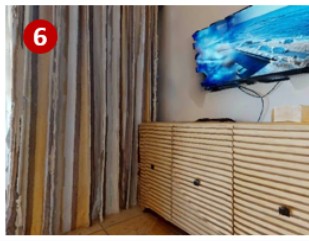

**Prompt**: *From the provided visual input, identify the most informative image frame (with IDs starting from 1) that offers the best chance of locating the tv monitor. Format your response as: Frame ID: [Selected Frame ID]*
**Ground Truth**: *Frame ID: 4*

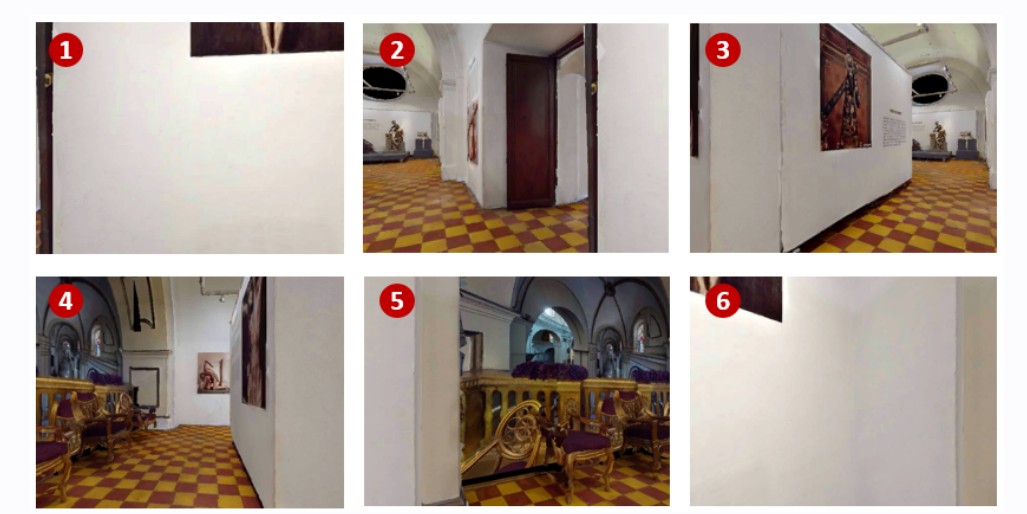

**Prompt**: *From the provided visual input, identify the most informative image frame (with IDs starting from 1) that offers the best chance of locating the plant. Format your response as: Frame ID: [Selected Frame ID]*
**Ground Truth**: *Frame ID: 1, 2, 4*

### E.3.2  WHERE2FIT

The Where2Fit Benchmark addresses the task of identifying free space on tables by predicting a set of 2D points. These tables are drawn from a variety of real-world scenes—such as offices, conference rooms, pantries, and workstations—and exhibit different levels of clutter, ranging from blank and sparse to densely occupied. The benchmark systematically increases the number of objects across these clutter conditions, presenting a progressive challenge. In addition, it incorporates critical physical constraints, including object dimensions, fit within the available space, and collision avoidance with other objects. The entire benchmark consists of 464 samples, including 200 generation tasks that require the model to output corresponding points, and 264 judgment tasks where the model must determine whether a given point would cause a collision.

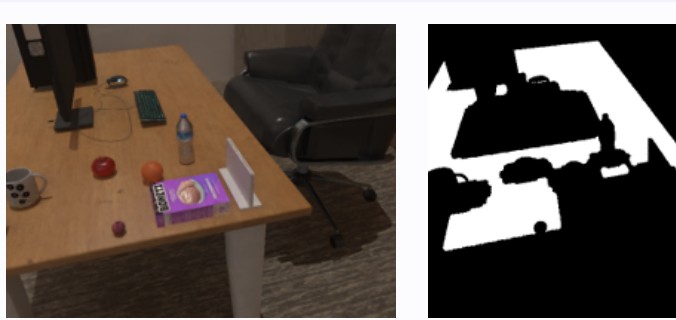

Prompt: Locate some free space for me on the table.

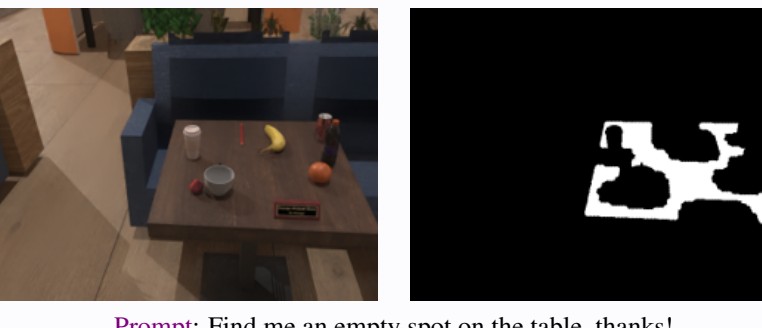

Prompt: Find me an empty spot on the table, thanks!

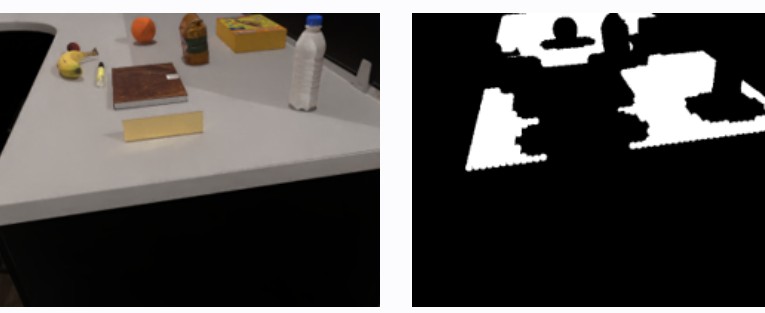

Prompt: Would you be able to place the red plug on the conveyor belt?

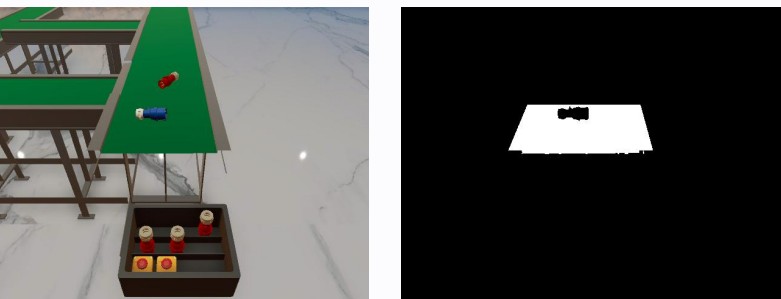

Prompt: Could you help me find a vacant area on the table?

### E.3.3    WHERE2APPROACH

The Where2Approach benchmark is required to identify unobstructed free space on a table while accounting for potential occlusions caused by surrounding chairs. The testbed features a long table cluttered with objects and encircled by randomly arranged chairs, simulating geometrically complex and occlusion-rich environments. This task necessitates advanced spatial reasoning under substantial visual occlusion, as well as the integration of locomotion and manipulation constraints. Specifically, the agent must determine feasible chassis positions that offer sufficient unobstructed area for successful placement operations. These requirements closely align with the challenges posed by **Mobile Placement (Hard)** tasks, which emphasize operation in highly constrained and visually disordered scenarios. The entire test set consists of 200 samples, each covering completely different perspectives, robot positions, tabletop object configurations, and chair arrangements.

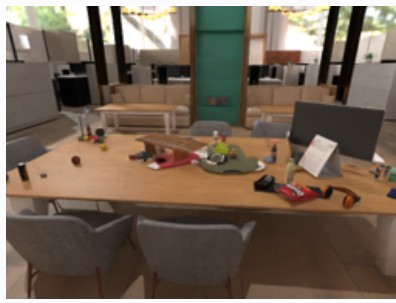 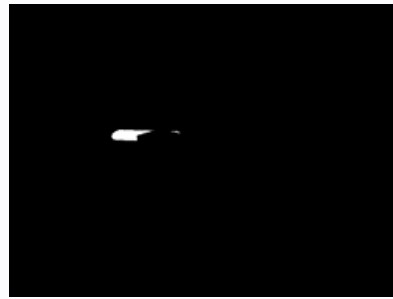

Prompt: Find the nearest free space on the table with no chairs around.

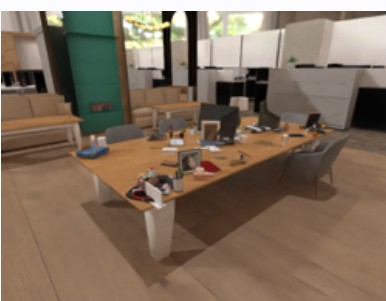 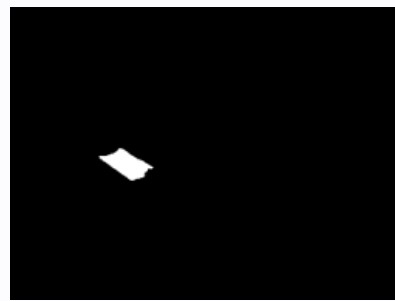

Prompt: Locate the closest empty spot on the table that isn't surrounded by chairs.

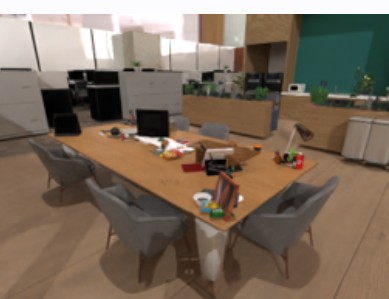 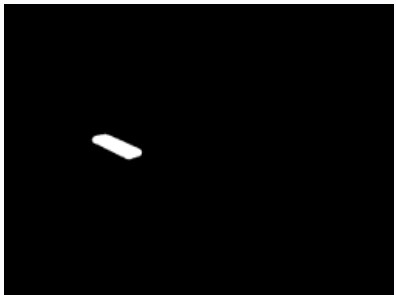

Prompt: Look for the nearest available area on the table where no chairs are placed nearby.

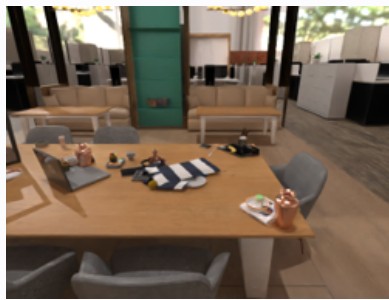 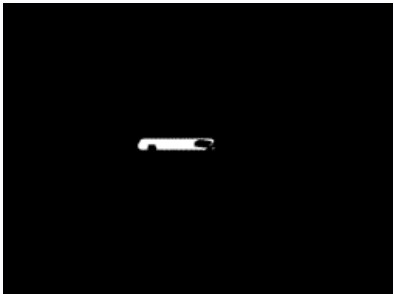

Prompt: Search for a nearby open space on the table that has no chairs in its vicinity.

### E.3.4  WHERE2GRASP

The Where2Grasp benchmark requires the identification of objects based on key attributes including color, size, location, and category. The evaluation set consist 200 samples and encompasses over 40 common object categories sourced from a variety of household and office environments, with diverse backgrounds. This task emphasizes object-centric cognitive capabilities, focusing on the perception and interpretation of object characteristics under real-world conditions.

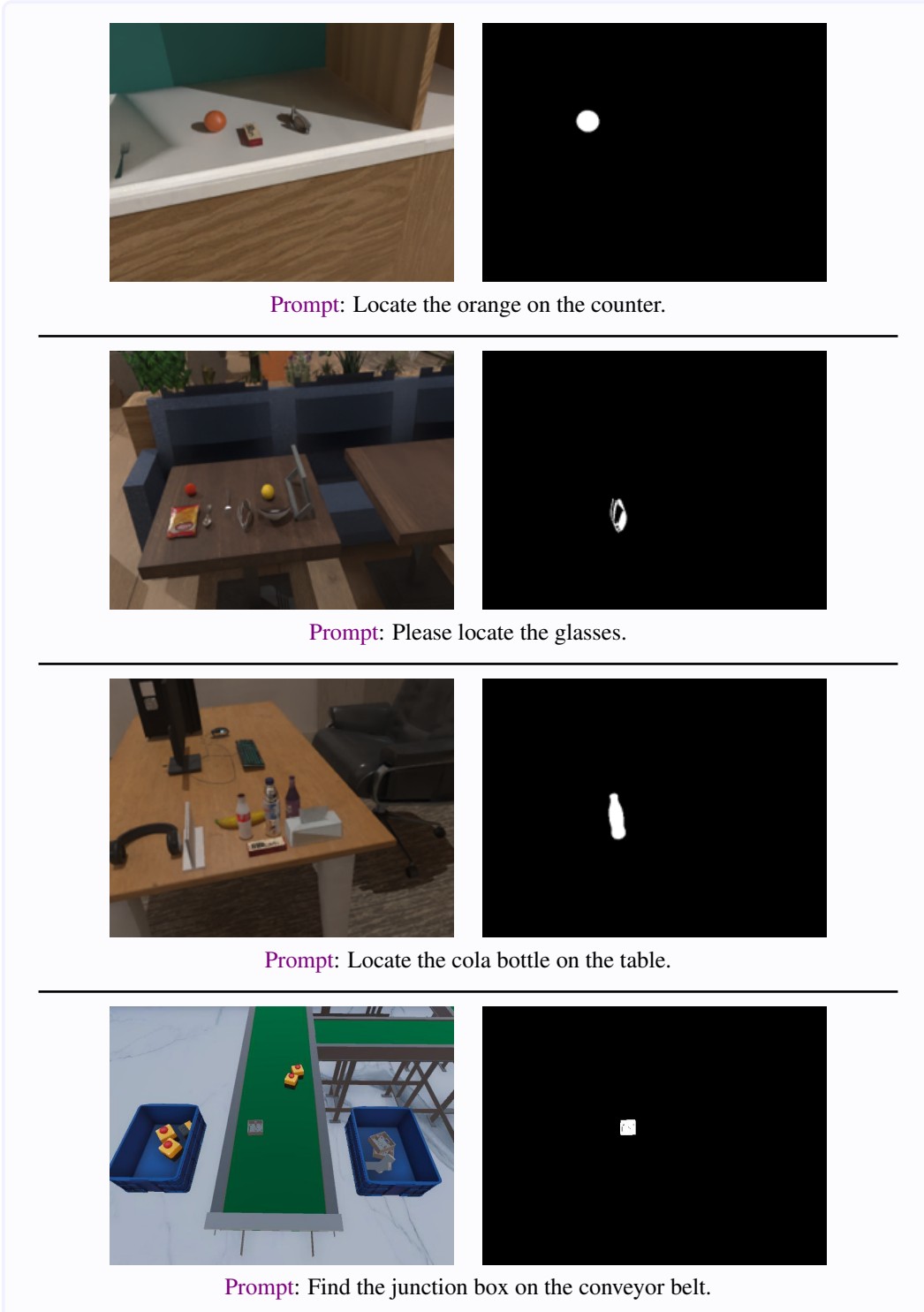

Prompt: Locate the orange on the counter.

Prompt: Please locate the glasses.

Prompt: Locate the cola bottle on the table.

Prompt: Find the junction box on the conveyor belt.

## E.4 DOWNSTREAM TASK DESCRIPTION

### E.4.1 MOBILE PLACEMENT EASY

For the Mobile Placement Easy benchmark, we constructed scenes with 8 tables in an office environment, with various items randomly scattered on the tabletops. There are a total of 40 types of items to enhance the diversity of the scenes. The robot's initial position is 1 to 1.5 meters away from

the edge of the table, with an angular deviation of -15 to 15 degrees, to observe the environment and objects. The scenes are divided into three levels based on the number of randomly scattered items on the tabletop: no-objects, sparse, and dense, with 0, 4, and 8 objects on the tabletop, respectively. The model's performance is evaluated in 200 simulation scenes, with 50 no-object scenes, 50 sparse scenes, and 100 dense scenes. We use the success rate of placing objects as the evaluation metric.

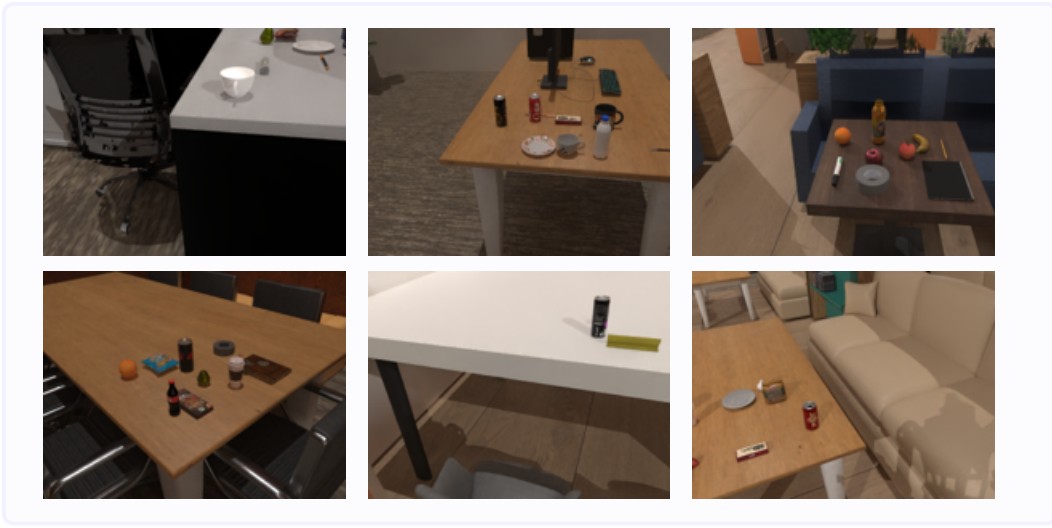

### E.4.2 MOBILE PLACEMENT HARD

In the Mobile Placement Hard benchmark, we construct a long-table environment with chairs positioned around the perimeter and varied objects distributed on the table surface. The robot is required to identify approachable regions unobstructed by chairs or densely placed tabletop items. The table periphery is systematically divided into 12 candidate zones (three per long side, one per short side, and four corners), each classified as either occupied (by chairs or dense objects, excluding corners) or vacant (clear spaces or sparse-object areas permitting approach). We additionally distribute random objects in the central table area, which improves visual realism and increases scene diversity.

Each simulation scenario involves randomized assignment of occupancy states to candidate regions, with objects sampled from a 70-category office inventory (e.g., monitors, books, mugs). The robot is initialized 0.9 to 1.5 meters from the table edge ($\pm15 \deg$ angular deviation) to observe the table and surroundings. The evaluation requires the model to predict feasible placement points, followed by robot execution attempts. Performance is assessed in 100 simulated scenarios, where we measure the success rates of planning placement trajectories.

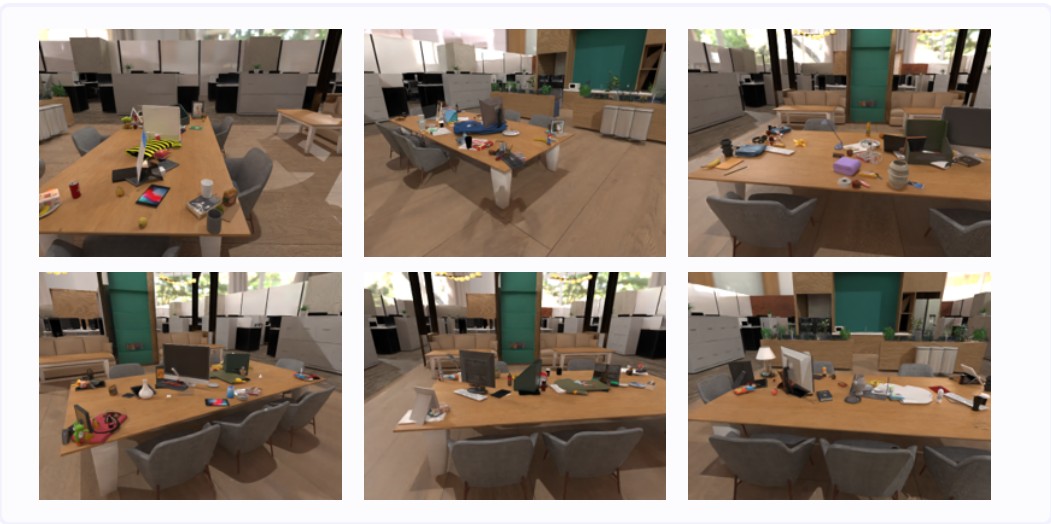

### E.4.3 MOBILE PICKUP

For the Mobile Pick-up benchmark, we constructed scenes with 8 tables in an office environment, with various items randomly scattered on the tabletops. The objects to be picked up include: pen holders, cans, bottles, coffee cups, oranges, pears, apples, and lemons. There are a total of 40 background objects to enhance the diversity of the scenes. The robot's initial position is 1 to 1.5 meters away from the edge of the table, with an angular deviation of -15 to 15 degrees, to observe the environment and objects. The scenes are divided into three levels based on the number of randomly scattered items on the tabletop: easy, medium, and hard. In the easy level, only the objects to be picked up are on the tabletop, while in the medium and hard levels, there are 3 and 7 additional objects besides the ones to be picked up. The model's performance is evaluated in 150 simulation scenes, with 60 easy scenes, 50 medium scenes, and 40 hard scenes. We use the success rate of picking up objects as the evaluation metric.

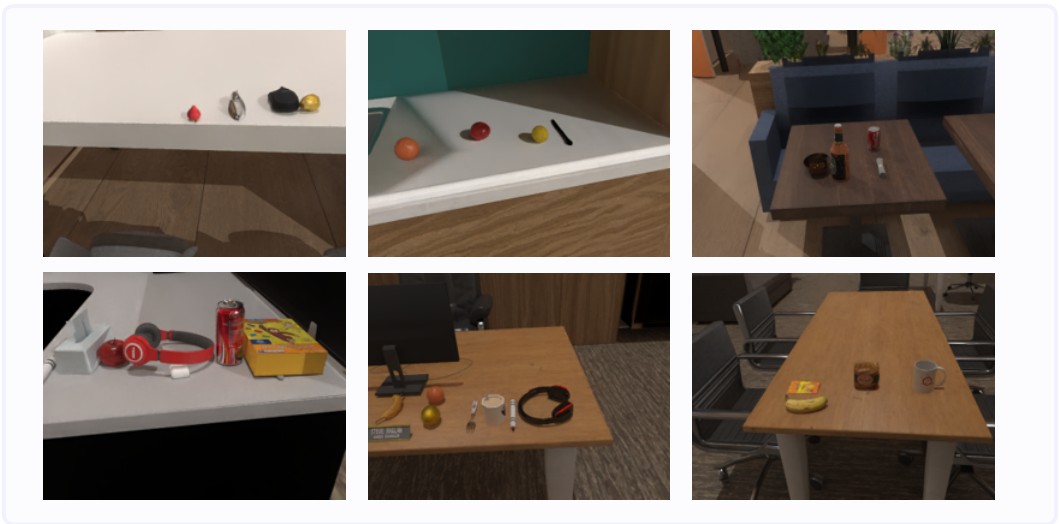

## F QUALITATIVE RESULTS OF OMNIEVA

Here we present sample inference results generated by OmniEVA, including 2D visual grounding, object reference, object part recognition, free space location, activate exploration, 3d questionn answering, 3d captioning, 3d visual grounding and 3d scene imagination. All examples shown were not included in the training dataset.

### F.1 QUALITATIVE RESULTS ON 2D SPATIAL REASONING TASKS

**2D Visual Grounding**

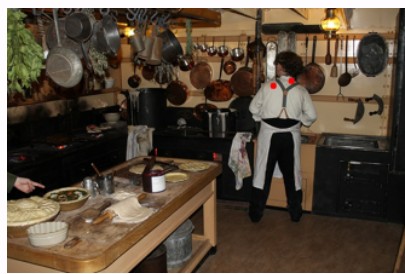

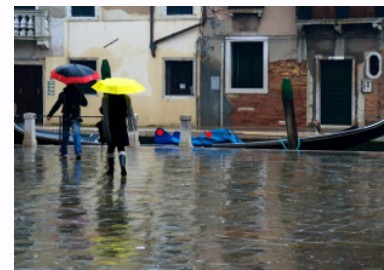

**Prompt**: *In the photo, where can I find the shirt?*
**Response**: *You can find the shirt at <point>(457, 117), (428, 125)</point>.*

**Prompt**: *Highlight the location of the tarp.*
**Response**: *Highlighted region: <point>(331, 219), (248, 215) </point>.*

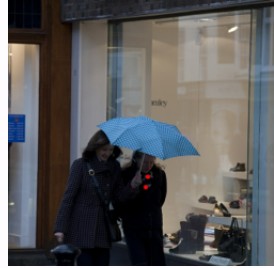

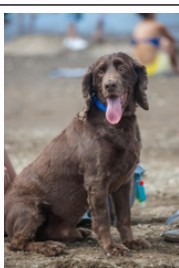

**Prompt**: *Locate the scarf in this image.*
**Response**: *The scarf is found at <point>(336,425),(331,450) </point>.*

**Prompt**: *Please specify the coordinates for the handbag.*
**Response**: *Specifying coordinates: <point>(331, 409), (256, 490) </point>.*

**Object Reference**

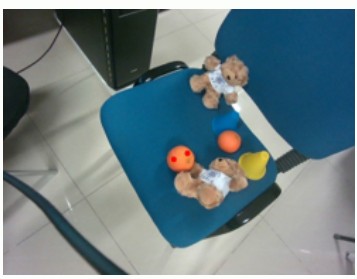

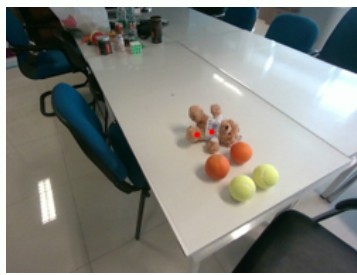

**Prompt**: *Please hand me the orange on the left.*
**Response**: *Points are <point>(302, 271), (343, 260)</point>.*

**Prompt**: *I would like to grab the teddy bear in shirt.*
**Response**: *Center points are <point>(347, 241), (374, 245) </point>.*

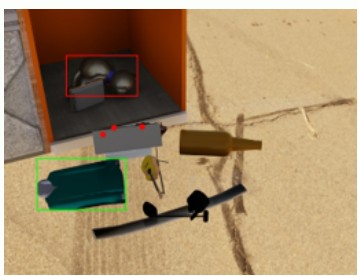

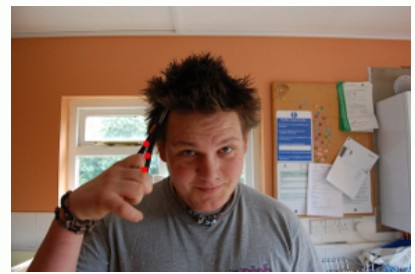

**Prompt**: *The image features an item encased within a red rectilinear shape and another item surrounded by a green rectangular boundary. Pinpoint several spot son any item situated between the objects framed by these rectangles.*
**Response**: *Locations are <point>(250, 212), (199, 214), (179, 226) </point>.*

**Prompt**: *What object is in <point>(213, 261), (218, 239), (216, 220)</point>?*
**Response**: *Scissors.*

## Object Part Recognition

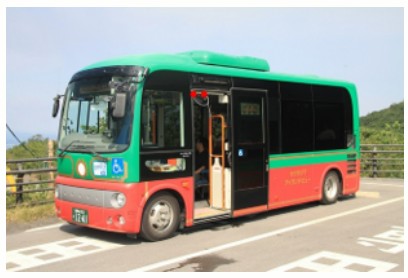

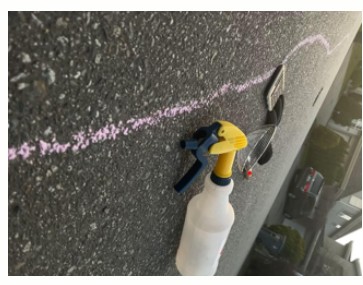

**Prompt**: *Can you locate the frame on this mirror?*
**Response**: *Points are <point>(301, 140), (318, 140)</point>.*

**Prompt**: *How do you interact with the strainers?*
**Response**: *By interacting with the <point>(1310, 880)</point>.*

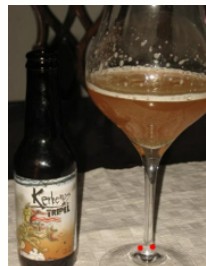

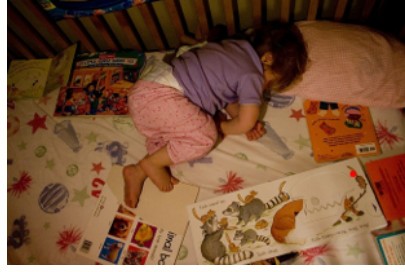

**Prompt**: *If I wanted to interact with the base of this glass, where would I find it?*
**Response**: *You should interact with it on the <point>(249, 458), (272, 456)</point>area.*

**Prompt**: *Could you point out the page of the book?*
**Response**: *<point>(556, 254)</point>.*

**Free Space Location**

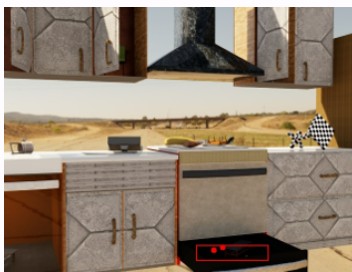

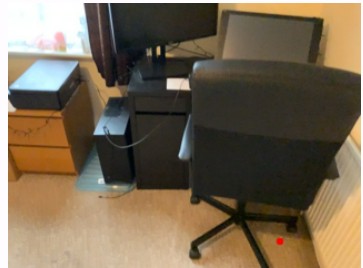

**Prompt**: *In the image, there's a region outlined by a red rectangle. Locate several points within the vacant area on this marked surface.*
**Response**: *Locations are <point>(396, 435), (380, 440)</point>.*

**Prompt**: *Point to a free spot behind white textured radiator at lower right.*
**Response**: *Points are <point>(394, 345)</point>.*

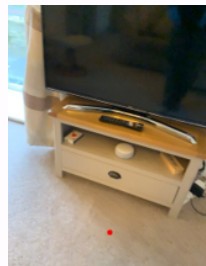

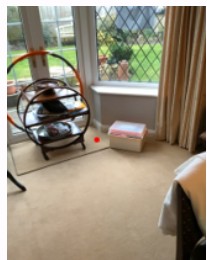

**Prompt**: *Point to a free spot in front of white painted wood chest of drawers at center.*
**Response**: *Points are <point>(196, 439)</point>.*

**Prompt**: *What is the location of a free spot to the right of wooden trolley at left? Please provide its 2D coordinates.*
**Response**: *Points are <point>(173, 258)</point>.*

**Active Exploration**

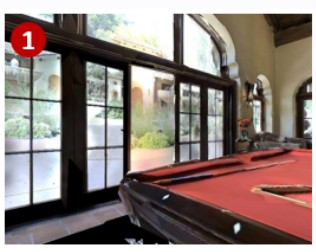
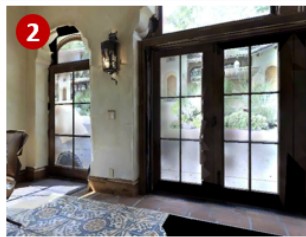
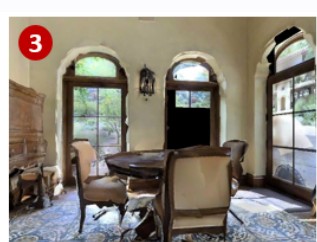
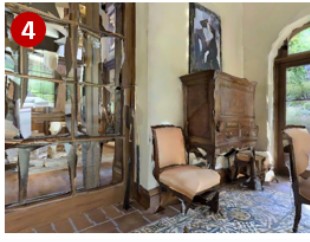
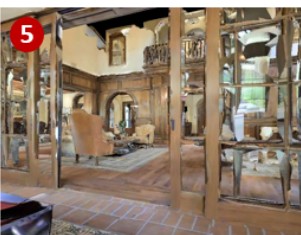
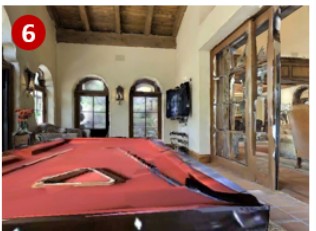

**Prompt**: *From the provided visual input, identify the most informative image frame (with IDs starting from 1) that offers the best chance of locating the fireplace. Format your response as: Frame ID: [Selected Frame ID]*

**Ground Truth**: *Frame ID: 5, 6*
**Prediction**: *Frame ID: 5*

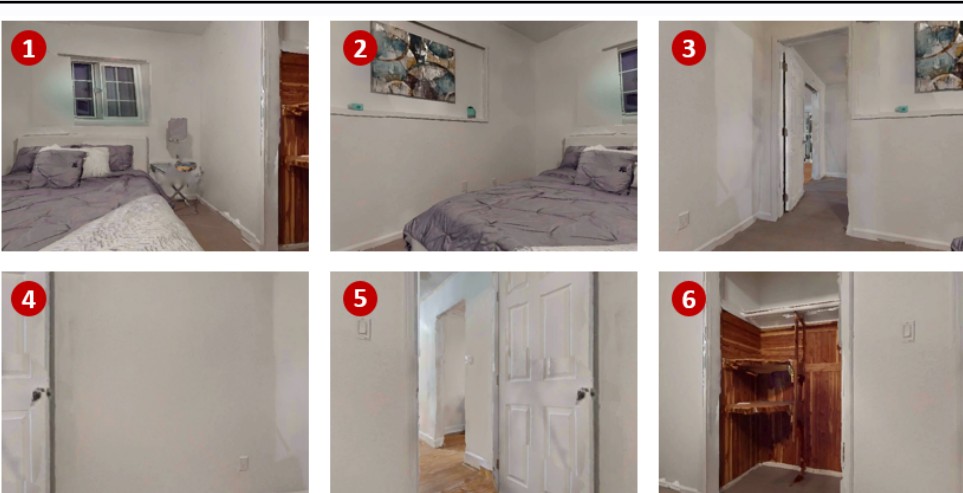

**Prompt**: *From the provided visual input, identify the most informative image frame (with IDs starting from 1) that offers the best chance of locating the sofa. Format your response as: Frame ID: [Selected Frame ID]*
**Ground Truth**: *Frame ID: 3, 5*
**Prediction**: *Frame ID: 5*

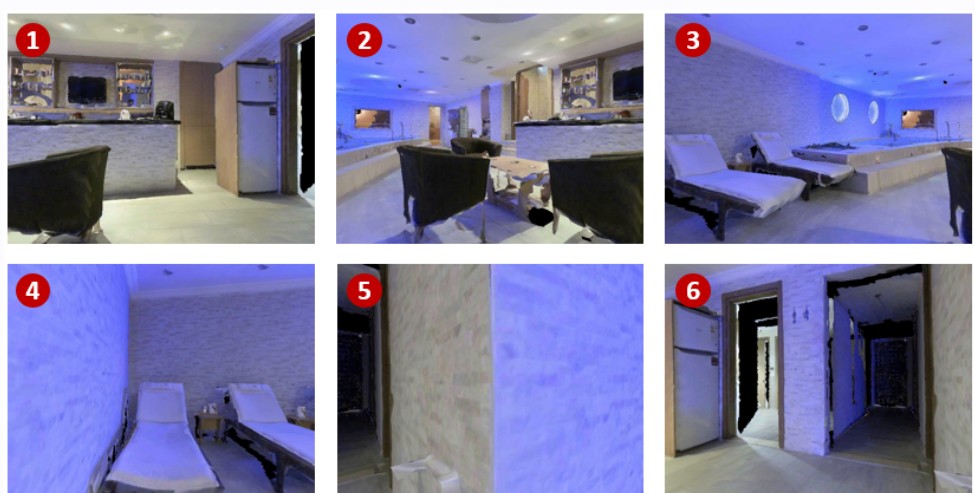

**Prompt**: *From the provided visual input, identify the most informative image frame (with IDs starting from 1) that offers the best chance of locating the sink. Format your response as: Frame ID: [Selected Frame ID]*
**Ground Truth**: *Frame ID: 2, 3*
**Prediction**: *Frame ID: 1*

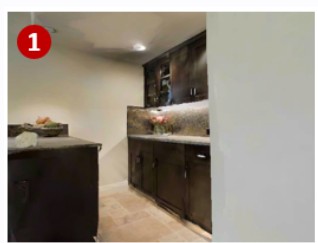 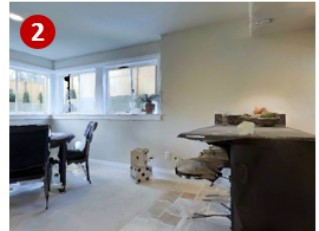 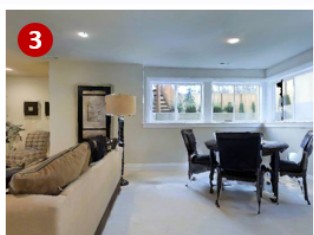
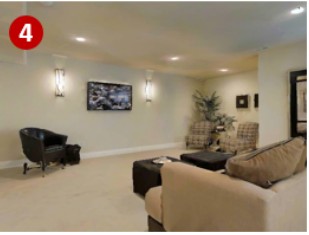 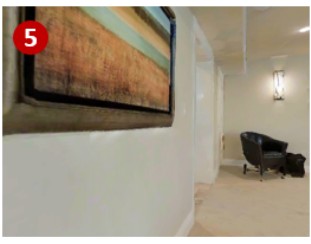 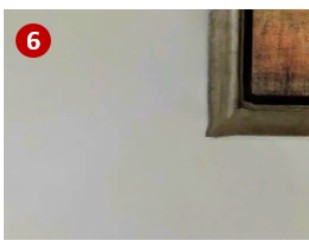

**Prompt**: *From the provided visual input, identify the most informative image frame (with IDs starting from 1) that offers the best chance of locating the* gym equipment. *Format your response as: Frame ID: [Selected Frame ID]*
**Ground Truth**: *Frame ID: 5*
**Prediction**: *Frame ID: 5*

## F.2    QUALITATIVE RESULTS ON 3D SPATIAL REASONING TASKS

**3D Question Answering**

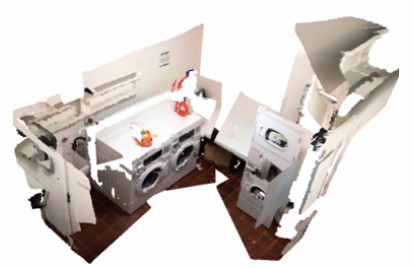 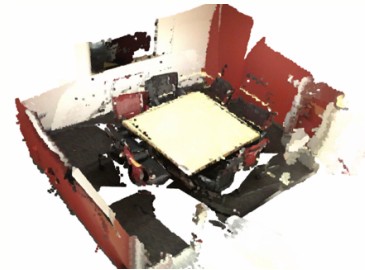

**Prompt**: *I am using the clothes dryer under another clothes dryer. What object is beneath the table to my left? ...*
**Ground Truth**: *box*
**Prediction**: *box*

**Prompt**: *I am sitting on a chair while facing the whiteboard and there is a chair on the same side of the table on my left. What is the shape of the table in front of me? ...*
**Ground Truth**: *rectangular*
**Prediction**: *rectangular*

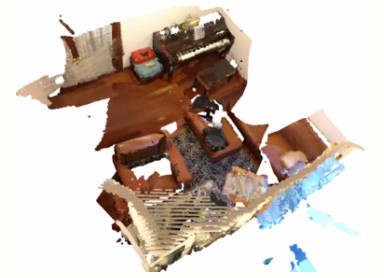

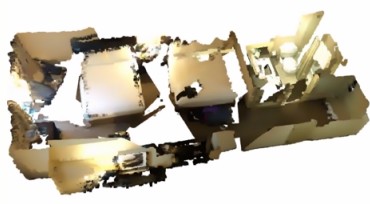

**Prompt**: *What color is the tool box to the left of the piano? Answer the question simply.*
**Ground Truth**: *red green*
**Prediction**: *red*

**Prompt**: *Where is the bathroom vanity located? Answer the question simply.*
**Ground Truth**: *to right of bathroom door below window*
**Prediction**: *left of toilet*

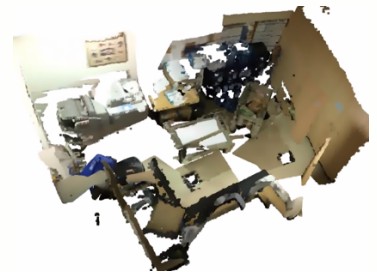

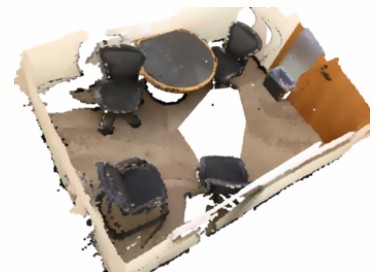

**Prompt**: *I am making some copies and there are some pictures on my left. What is brown and square to my left? ...*
**Ground Truth**: *table*
**Prediction**: *table*

**Prompt**: *I am throwing trash with a chair on my right within reach. What is the color of the chair that is within reach on my right? ...*
**Ground Truth**: *black*
**Prediction**: *black*

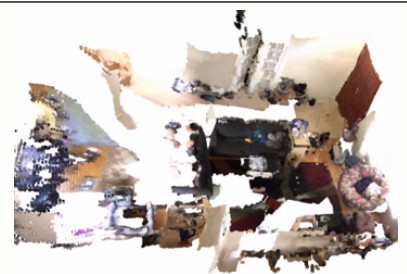

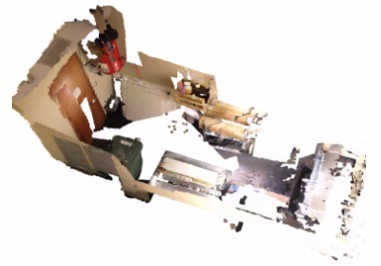

**Prompt**: *I am opening the window with the shoe rack on my right. What is on top of shelf behind me that I can place on my back? ...*
**Ground Truth**: *bag*
**Prediction**: *pillow*

**Prompt**: *I am standing by the poster printer and there is a table behind me. Can I see the box if I turn around? ...*
**Ground Truth**: *no*
**Prediction**: *yes*

**3D Captioning**    (red box: 3D box given in prompt)

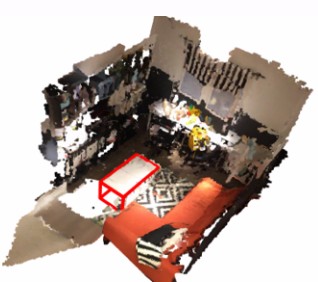 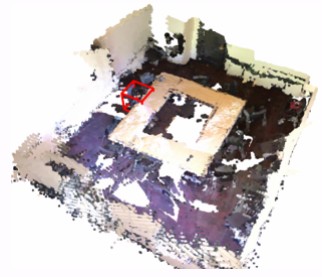

**Prompt**: *Given an object located at <3dbox>(150,155,51,5,10,5)</3dbox>, describe ...*
**Ground Truth**: *The coffee table is white. The coffee table is in front of the orange couch.*
**Prediction**: *The coffee table is in the middle of the room. The coffee table is a white rectangle.*

**Prompt**: *Given an object located at <3dbox>(132,151,54,7,6,7)</3dbox>, describe ...*
**Ground Truth**: *The chair is the color black. It is facing the table along with other chairs. It is spaced beside other chairs.*
**Prediction**: *This is a black chair. It is at the corner of the table.*

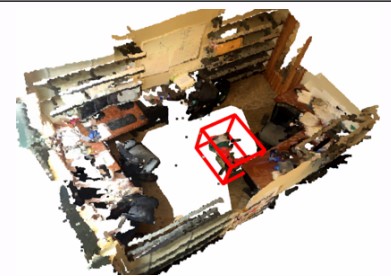 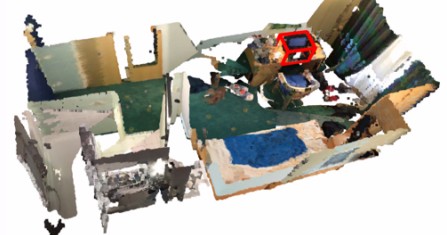

**Prompt**: *Given an object located at <3dbox>(153,152,55,8,7,6)</3dbox>...*
**Ground Truth**: *The chair is red and green. It is to the right of the black chair.*
**Prediction**: *This is a chair with no arms. It is in front of a desk.*

**Prompt**: *Given an object located at <3dbox>(140,166,58,4,5,4)</3dbox>...*
**Ground Truth**: *A black computer with a blue screen. It's located on a wooden desk with a wooden chair in front of it.*
**Prediction**: *This is a black laptop. It is on a desk.*

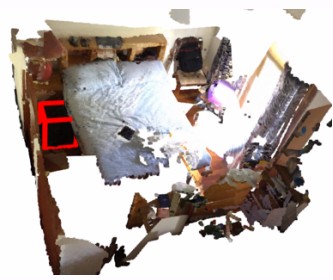 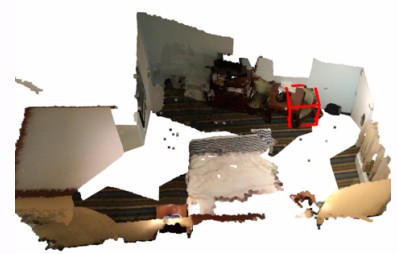

**Prompt**: *Given an object located at <3dbox>(137,158,51,4,4,5)</3dbox>...*
**Ground Truth**: *The black backpack is directly in front of the nightstand. The backpack is also directly to the left of the bed.*
**Prediction**: *The backpack is black. It is on the floor, to the left of the bed.*

**Prompt**: *Given an object located at <3dbox>(159,133,53,6,7,9)</3dbox>...*
**Ground Truth**: *This is a sturdy chair, made of wood and with green cushioning. It is in contact with a wall surface and a brown table. It is oriented about 120 degrees in relation to the viewer.*
**Prediction**: *The chair is the one closest to the door. the chair has a green seat and four legs.*

**3D Visual Grounding** (red Box: ground truth; green box: prediction)

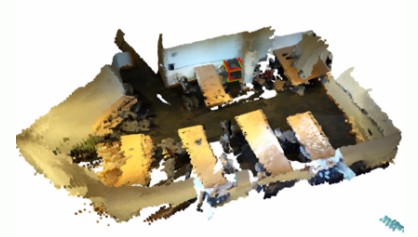 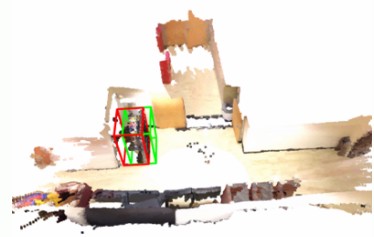

**Prompt**: *... There is a black office chair. placed in the side of the wall.*
**Response**: *<3dbox>(130, 155, 55, 6, 5, 4)</3dbox>.*
**Result**: *Success*

**Prompt**: *... It is next to the tall floor lamp. it is a snack machine and it is full of tasty snacks. it is over by the door ...*
**Response**: *<3dbox>(143, 135, 59, 10, 8, 17)</3dbox>.*
**Result**: *Success*

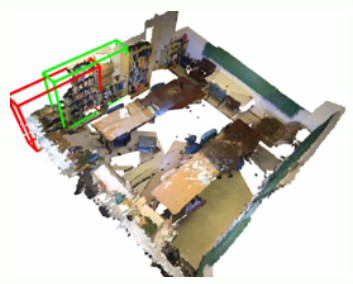

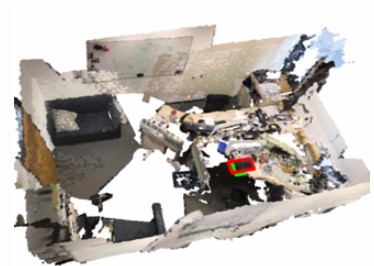

**Prompt**: ... *In the corner of the room are boxes. the first two book shelves in the corner to the right of the boxes are the bookshelves we are looking for.*
**Response**: *<3dbox>(127, 139, 61, 4, 21, 20)</3dbox>*
**Result**: *Success*

**Prompt**: ... *The keyboard is sitting on top of the desk. it is in front of the left-side monitor.*
**Response**: *<3dbox>(141, 125, 59, 9, 1, 20)</3dbox>*
**Result**: *Success*

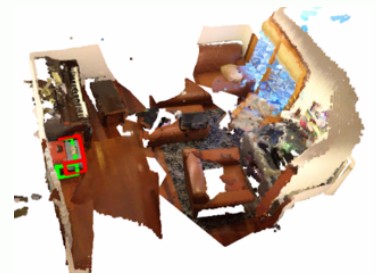

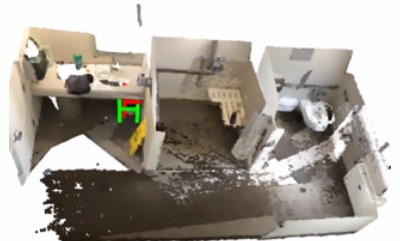

**Prompt**: ... *This is a black trash can. it is under the counter top.*
**Response**: *<3dbox>(159, 143, 51, 3, 4, 4)</3dbox>.*
**Result**: *Success*

**Prompt**: ... *This is a green tool box. . the green tool box is in front of a red tool box on the floor next to a piano.*
**Response**: *<3dbox>(147, 127, 52, 5, 3, 3)</3dbox>.*
**Result**: *Success*

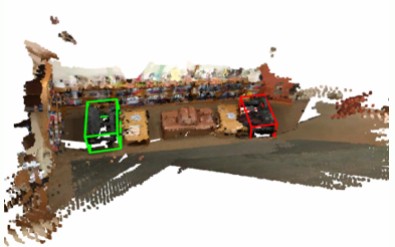

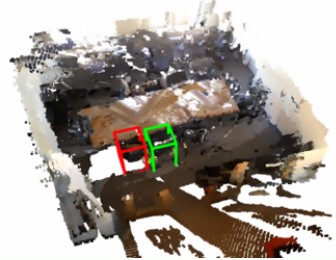

**Prompt**: ... *This is a black leather loveseat in a library. if you were sitting in it, the long, short bookshelf would be on your right and the tall book shelf, opposite you across the room.*
**Response**: *<3dbox>(145, 168, 57, 15, 9, 7)</3dbox>.*
**Result**: *Fail*

**Prompt**: ... *The black chair, in front of the chair with a brown table, on the right has 2 black chairs. on the left is a black chair with the left to right it is the first chair.*
**Response**: *<3dbox>(154, 159, 54, 5, 6, 10)</3dbox>.*
**Result**: *Fail*

**3D Scene Imagination** (from HM3D and MP3D validation set)

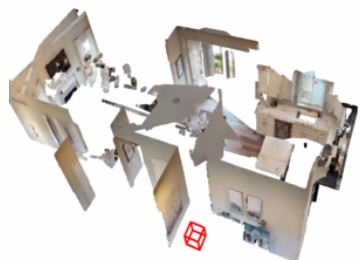

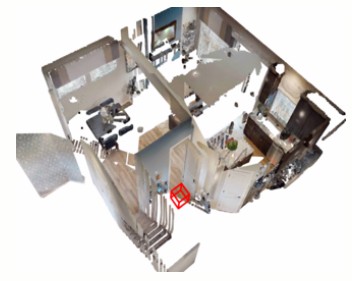

**Prompt**: *..., when the agent walks to position <3dbox>(122,129,21,10,10,10) </3dbox>, what new objects might become visible? ...*
**Ground Truth**: *You may see various pictures, chairs, vases, trays, plates, coffee mugs, washing machines, lamps, curtains, rugs, vents, balustrades, sensors, tables, books, candlesticks, and pillows.*
**Prediction**: *You may see various lamps, ventilations, and cabinets.*

**Prompt**: *..., when the agent walks to position <3dbox>(153,164,26,10,10,10) </3dbox>, what new objects might become visible? ...*
**Ground Truth**: *You may see a few motion detectors, blinds, wall toilet papers, bathroom towels, trashcans, toilets, decorations, decorative plants, and bathroom shelves.*
**Prediction**: *You may see various lamps, pictures, and beds.*

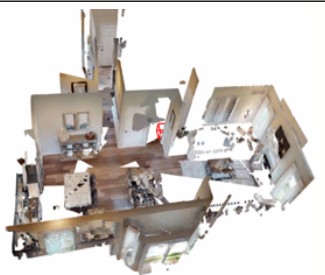

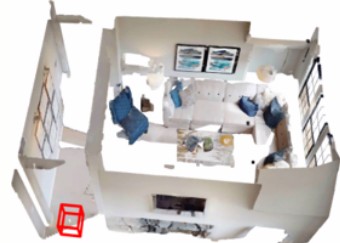

**Prompt**: *..., when the agent walks to position <3dbox>(146,157,21,10,10,10) </3dbox>, what new objects might become visible? ...*
**Ground Truth**: *You may see various bathroom cabinets, pillows, benches, handbags, seats, and pictures.*
**Prediction**: *You may see various lamps, cabinets, pillows, and beds.*

**Prompt**: *..., when the agent walks to position <3dbox>(125,155,49,10,10,10) </3dbox>, what new objects might become visible? ...*
**Ground Truth**: *You may see various boxes, pictures, lamps, ceiling lamps, drawers, stools, appliances, beds, fire alarms, pillows, fireplace walls, racks, and tables.*
**Prediction**: *You may see various pictures, chairs, tables, and plants.*

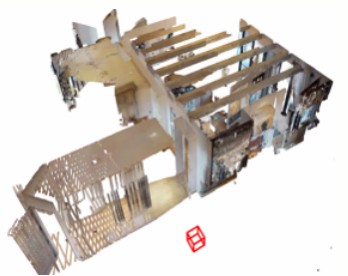 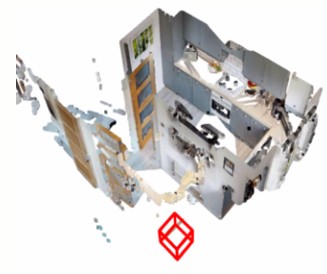

**Prompt**: *..., when the agent walks to position <3dbox>(130,100,34,10,10,10) </3dbox>, what new objects might become visible? ...*

**Ground Truth**: *: You may see various shelves, tables, mirrors, decorations, chairs, sinks, faucets, liquid soaps, door windows, TVs, curtains, carpets, wine racks, closet areas for hanging clothes, clothes, bath-tubs, bath towels, shower soap shelves, soap bottles, shower walls, shower floors, shower glasses, shower mats, towel bars, towels, shower handles, ceiling lamps, and shower doors.*

**Prediction**: *You may see various pictures, tables, chairs, plants, cabinets, counters, sinks, and towels.*

**Prompt**: *..., when the agent walks to position <3dbox>(166,186,24,10,10,10) </3dbox>, what new objects might become visible? ...*

**Ground Truth**: *You may see various ceiling lamps, smoke alarms, doorposts, headboards, wall lamps, nightstands, radios, lounge chairs, pillows, dog beds, window curtains, bathtubs, shower doors, bathroom cabinets, baskets, stair steps, and towels.*

**Prediction**: *You may see various shelves, cabinets, tables, chairs, flower vases, lamps, pictures, curtain rails, curtains, and carpets.*

---

**Object Navigation in Partially Observable Environments** (examples are sampled from HM3D validation set): green box–predicted 3D subgoal at each planning step; red line: past trajectory; red box: target object to be found

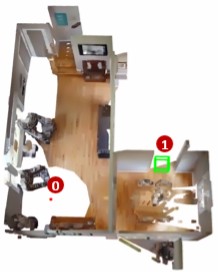 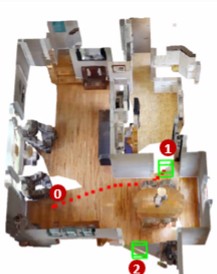 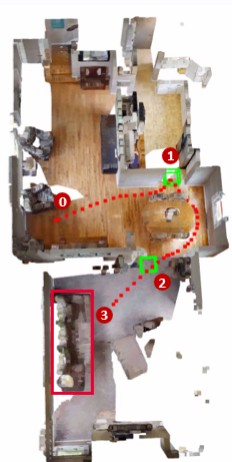

**Prompt**: *Based on the currently observed environment, determine the best position to go to find the toilet.*

**Exploration Trajectory**: *OmniEVA first explored the doorway of the room in the upper right corner. After mistakenly entering the kitchen area, it quickly adjusted its path, selected a new area to explore, and directly located the sofa in the living room.*

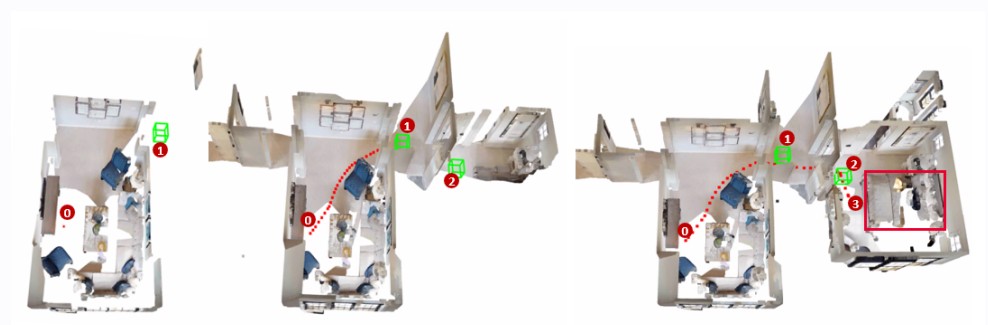

**Prompt**: *Based on the currently observed environment, determine the best position to go to find a bed.*

**Exploration Trajectory**: *The model first moves to the hallway at the intersection of multiple rooms to gather more comprehensive environmental information. It then determines that the room in the lower right corner is more likely to be a bedroom, proceeds to that location, and ultimately finds the bed.*

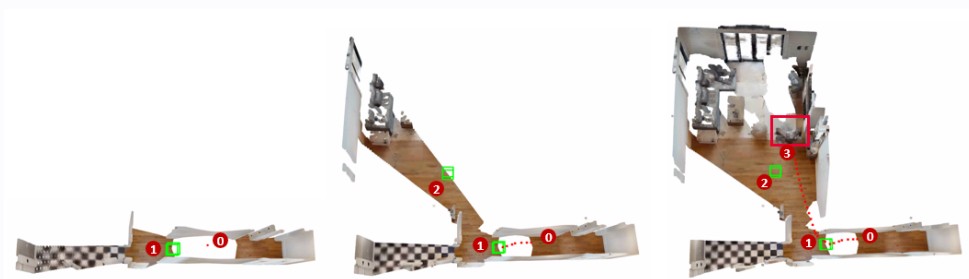

**Prompt**: *Based on the currently observed environment, determine the best position to go to find the chair.*

**Response**: *The model first walks to the center of the corridor, sees the sofa in the upper left, then heads to that room area and finds the chair.*

## F.3 EXAMPLES OF EMBODIMENT-AWARE REASONING

**Embodiment-aware Reasoning**

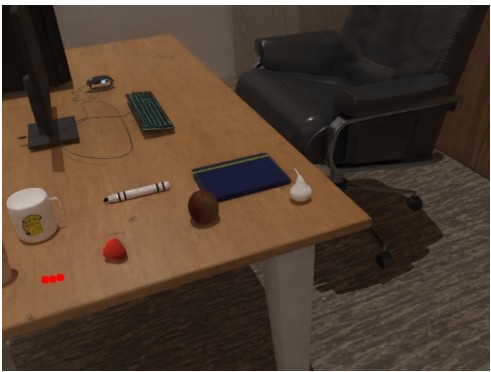

**Prompt**: *Could you help me find a vacant area on the table?*

**Response**: *<think>1. **Describing the Image**: The image shows a wooden table with various objects placed on it, including a mug, a highlighter, a small fruit, a notebook, a white object resembling a bottle, and a chair positioned nearby. The table has some visible*

*empty spaces, but the objects occupy certain areas.2. **Common-Sense Considerations**: - The goal is to identify free space on the table where additional objects can be placed without causing collisions or instability. - Free space should be large enough to accommodate typical objects like the ones present on the table. - The placement should avoid overlapping with existing objects or causing them to tip over.3. **Relevant Factors**: - **Object Dimensions**: The size of the objects already on the table should be considered when determining free space. - **Table Dimensions**: The overall dimensions of the table should be taken into account to ensure that the identified free space is realistic. - **Collision Avoidance**: Any suggested free space should not interfere with the existing objects or lead to potential collisions with the chair or other furniture.4. **Free Space Identification**: - The bottom-left corner of the table appears to have some free space, as there are no objects occupying that area. - The top-right corner also seems to have some free space, though it is farther than the bottom-left. - The middle-right section of the table has limited free space due to the presence of the notebook and the white bottle.5. **Embodiment Awareness**: The operational range of the robot necessitates that it be positioned as close as practicable to the table's bottom edge.6. **Answer Formulation**: - Based on the analysis, the bottom-left corner of the table is the most suitable for placing additional objects, as it offers the largest and most unobstructed free space.</think><answer>Locations are <point>(66, 359), (56, 360), (76, 357)</point>.</answer>*

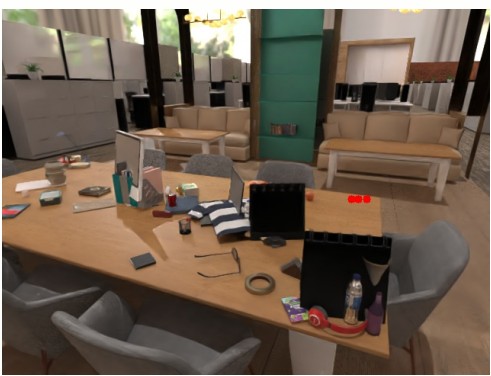

**Prompt**: *Find the nearest free space on the table with no chairs around.*

**Response**: *<think>To answer the question, we need to analyze the image comprehensively:1. **Describing the Image**: The image shows a large wooden table in a room with multiple chairs placed around it. The table is cluttered with various objects, including books, containers, tools, and other miscellaneous items. There are also some empty spaces on the table.2. **Common-Sense Considerations**: - The question asks for "nearest free space" on the table, meaning we need to identify areas that are not occupied by objects and are close to the center of the table. - Free space should be large enough to place something without causing a collision with existing objects. - The table's edges and corners are potential candidates for free space, but they may not be ideal due to limited usable area.3. **Relevant Factors**: - Identify all objects currently occupying the table. - Determine the table's layout and identify any visible gaps or open areas. - Assess the proximity of these gaps to the center of the table. - Ensure the identified space is large enough to accommodate a new object.4. **Analysis**: - The table has several visible gaps, particularly near the center-left and top-right sections of the table. - These gaps appear to have enough space to place small to medium-sized objects without causing a collision with existing items. - The center area of the table is less cluttered than the top-right edges, but is obscured by chairs in front.5. **Embodiment Awareness**: To ensure the robot can cover its entire potential work area, placement near the top-right edge of the table is critical.6. **Conclusion**: - The nearest free space on the table can be identified as the central-left section of the table, specifically around the coordinates (460, 250) to (470, 250), which is a relatively clear area with no objects obstructing it.</think><answer>Points are <point>(467, 249), (458, 249), (478, 249)</point>. </answer>*

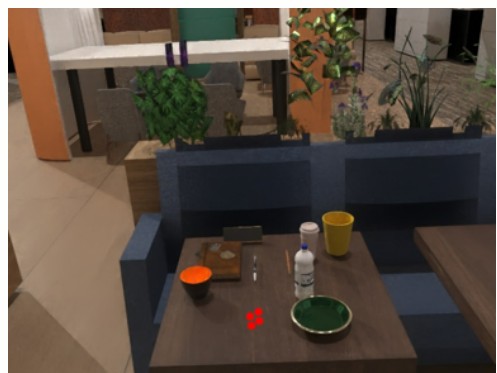

**Prompt:** Could you help me find a vacant area on the table?
**Response:** *<think>* ..., 1. Describing the Image: The image shows a table with ... 2. Common-Sense Considerations: - A vacant area must not overlap any existing objects ... large enough to accommodate ... avoid edges or corners where objects might fall off. 3. Vacant Area Identification: The left side of the table appears to have some empty space, particularly around the yellow cup and the green plate. 4. Embodiment Awareness: Considering the range of the robot's executable area, it should be placed as close to the bottom edge of the table as possible. 5. Output: - Based on the analysis: ... *</think> <answer>* Points are <point>(319, 416), (317, 404), (328, 398), (329, 410)</point>.*</answer>*

## G  REAL WORLD DEPLOYMENT EXAMPLES

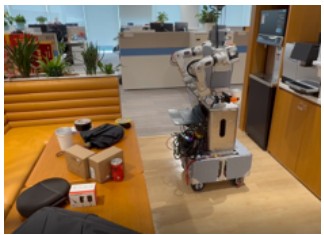 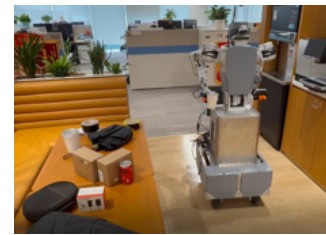 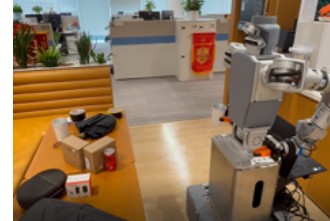

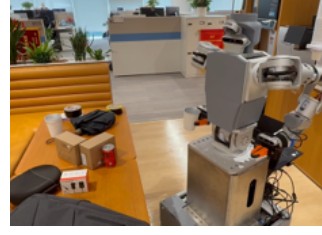 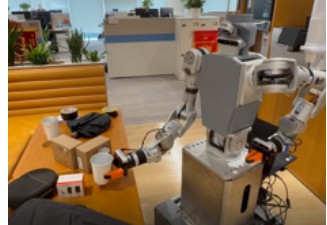 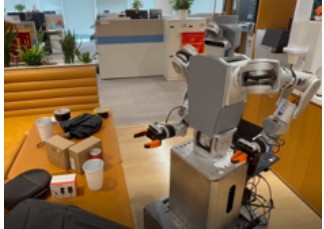

**Figure 12: Example for Deployment of OmniEVA on Real World Robots**. Prompt: Place the paper cup in the empty space on the table at the back right.

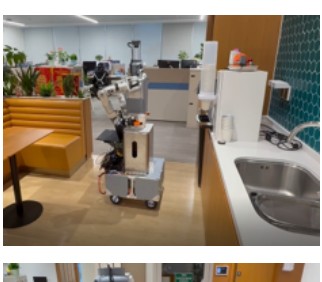 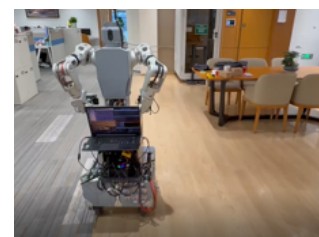 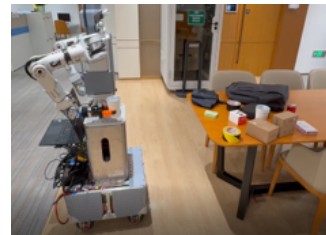

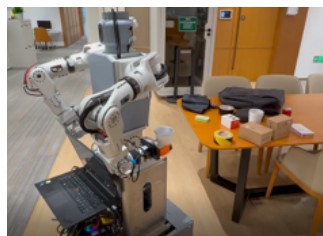 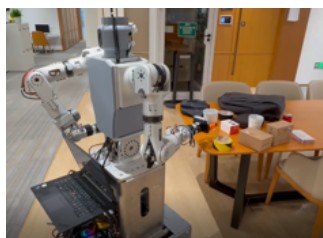 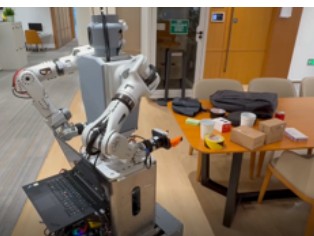

**Figure 13: Example for Deployment of OmniEVA on Real World Robots**. Prompt: Place the cup on the long table next to the meeting room.

# H USE OF LARGE LANGUAGE MODELS

In accordance with the ICLR 2026 policy on LLM usage, we disclose that Large Language Models (specifically, DeepSeek-R1 (Guo et al., 2025)) were employed during the writing process of this paper. Their usage was limited to aiding in the correction of grammatical errors of the manuscript. All scientific ideas, experiments, and analyses were conducted solely by the authors.

