# OpenReview forum: "OmniEVA: Embodied Versatile Planner via Task-Adaptive 3D-Grounded and Embodiment-aware Reasoning"
_ICLR.cc/2026/Conference — ICLR 2026 Poster_

### Official Review · Reviewer_BYjf · 2025-10-30

**Soundness:** 2
**Presentation:** 2
**Contribution:** 2
**Rating:** 4
**Confidence:** 4

**Summary:**

This paper primarily focuses on improving spatial reasoning, task planning, and embodiment feasibility within vision-language models (VLMs). Specifically, it introduces a gating mechanism to selectively fuse token-wise 2D visual features with their corresponding 3D representations, enhancing the model’s ability to absorb and reason over 3D information. Moreover, it incorporates task reward and embodiment feasibility objectives to strengthen the framework’s awareness of both task completion effectiveness and compliance with embodied physical constraints.

**Strengths:**

Enhancing the VLM’s understanding of 3D spatial structures is highly necessary, and enabling it to be aware of the robot’s embodiment feasibility is an important and well-motivated goal.

The introduction of the gating mechanism, task reward, and embodiment feasibility reward is conceptually sound and appears technically reasonable.

The experimental evaluation is comprehensive—covering diverse benchmarks that include 3D spatial reasoning, sub-goal localization in 3D environments, and 2D embodied reasoning tasks.

**Weaknesses:**

The overall writing quality is poor. Although the structure is generally acceptable, many paragraphs lack clear focus, and several sentences are unnecessarily verbose, leading to poor readability and coherence.

Some key sections lack sufficient detail. For instance, several tasks are introduced without intuitive explanations — while I understand that detailed descriptions are provided in the appendix, a concise and intuitive introduction in the main text is still necessary. Moreover, the design and acquisition process of the newly introduced rewards are not clearly explained.

The baselines in Table 1 resemble an ablation study, and the effectiveness of the proposed gating mechanism is not convincingly demonstrated by the presented results.

**Questions:**

Could you elaborate on how the task reward and embodied feasibility rewards are specifically designed and obtained?

In Table 3, did the baseline models also utilize the training datasets from these benchmarks?

In the real-world experiments, how does the model’s pick-and-place reasoning interface with and control the low-level policy?

Since embodiment awareness depends on hardware characteristics, and different hardware platforms have varying physical constraints, how does the proposed method generalize or adapt to these differences?

Can you provide a comparison of the reasoning time across models or settings?

---

> ### Author Response · Authors · 2025-11-27
>
> ## Regarding W1: Overall Writing Quality and Clarity
> We sincerely apologize for the lack of clarity and verbosity in the initial submission. We agree that high readability is crucial for effectively communicating our contributions. In response, we have undertaken a comprehensive revision of the entire manuscript. Key improvements include:
> - ​**Refocused Paragraphs and Explicit Structure**: We have significantly restructured the Methodology (Section 3) to provide a clearer, more logical flow. The three-stage training pipeline is now explicitly defined upfront: **(1) TAGR Pretraining, (2) Supervised Fine-Tuning (SFT) yielding OmniEVA-Base, and (3) Reinforced Finetuning (RFT) yielding OmniEVA-ER**. This structure is consistently maintained throughout the section to enhance focus.
>
> - **Simplifying Language**:We have condensed verbose sentences and replaced technical jargon with more intuitive explanations to improve readability and coherence, particularly in the Introduction and Abstract.
>
> - **Enhanced Implementation Details**: To improve clarity and reproducibility, we have added substantial detail to the Appendix, including:
>   ​**Appendix A.4**: Pseudo-code for the mixed training of 2D and 3D data.
>   **Appendix A.5**: Pseudo-code of the implementation detail of embodiement-aware reasoning.
>   ​**Appendix B.3**: A new analysis on the robustness of OmniEVA under dynamic environments.
>
>   **Appendix C.4**:  Implementation details of the ablation studies of TAGR Module.
>
> We hope that these significant revisions have substantially improved the readability and clarity of the paper.
>
> ## Regarding W2: Sufficient Detail in Key Sections
> Thank you for this critical feedback. We will enhance the main text to provide more intuitive explanations for the introduced tasks (Where2Go, Where2Grasp, etc.) and a clearer description of the reward functions.
>
> - **Task Introductions**:​ In the revised manuscript, Section 4.1 ("Benchmarks for Evaluation") will include concise, intuitive descriptions of each primitive task, explaining their role as fundamental building blocks for composite embodied skills, before referring readers to the appendix for full details.
>
> - **Reward Design Explanation**:
> The design and acquisition of the task reward ($r^{\text{task}}$) and embodiment feasibility reward ($r^{\text{embod}}$) are central to our TE-GRPO algorithm:
>     - **Task Reward ($r^{\text{task}}$):** This reward is derived from the accuracy of the model's output against ground-truth labels (for a pointing task, it could be a region represented using a mask) or success criteria for a given task (e.g., correctly identifying a location). It is directly obtained from the evaluation metrics of the benchmark tasks.
>     - **Embodiment Feasibility Reward ($r^{\text{embod}}$):** This reward is designed to penalize plans that violate physical constraints. It is obtained through a feasibility checker that analyzes the proposed plan (e.g., a predicted grasping point) against a set of predefined constraints, such as:
>     - **Reachability:** Is the target point within the robot's workspace?
>     - **Collision Avoidance:** Does the path to the point result in a collision with other objects?
>
> Additional implementation details, including pseudo-code, are provided in Appendix A.5.

---

> ### Author Response · Authors · 2025-11-27
>
> ## Regarding W3: the Effectiveness of the Gating Mechanism
>
> We appreciate the reviewer's comment that Table 1 resembles an ablation study. To more convincingly demonstrate the gating mechanism's efficacy, we have enhanced our analysis in Section 4.2.
>
> The core advantage of our Task-Adaptive Gated Router (TAGR) is its ability to act as a **dynamic mediator**, selectively injecting 3D features only when they are geometrically essential for the task. This avoids the computational burden and potential noise introduced by static, hard-coded 3D fusion methods used in prior work.
>
> To rigorously validate this design, we conducted **comparative experiments against alternative fusion strategies**, including **cross-attention-based** methods.  However, our results, now detailed in Table 1, show that naive cross-attention approaches (using either separate or interleaved visual and 3D tokens) perform significantly worse. They suffer a substantial performance drop—approximately 6 points on SQA3D and a striking 50-point decrease on Scan2Cap. This degradation occurs because simply treating 3D data as separate tokens doubles the input sequence length and forces the model to learn complex cross-modal interactions from scratch, which is computationally expensive and data-inefficient.
>
> | Methods | SQA3D | ScanQA | Scan2Cap | ScanRefer |
> | :----- | :-----: | :-----: | :-----: | :-----: |
> | Cross-Attention (seperate visual & 3D tokens) | 55.1 |  27.5 |  43.3 | 4.5 |
> | Cross-Attention (interleaved visual & 3D tokens) |  55.8 |  27.5 |  42.0 | 3.6 |
> | Hard-coded 3D Integration | 61.2 |  31.5 |  95.5 | 41.2 |
> | Without 3D Integration |  61.2 | 30.7 | 75.5 | 4.3 |
> | Dynamic 3D Integration (hard-gating) | 62.6 | 30.8 | 97.9 | 43.1 |
>
> Furthermore, we experimented with a **soft-gating alternative** (continuous sigmoid weighting), which consistently underperformed compared to our hard-gating strategy. Soft-gating compromises the numerical stability of the sinusoidal positional encodings that are crucial for precise spatial reasoning.
>
> | Methods | SQA3D | ScanQA | Scan2Cap | ScanRefer |
> | :----- | :-----: | :-----: | :-----: | :-----: |
> | Dynamic 3D Integration (soft-gating) |  60.6 | 30.9 | 85.6 | 26.9 |
> | Dynamic 3D Integration (hard-gating) | 62.6 | 30.8 | 97.9 | 43.1 |
>
> These results confirm that a **dedicated, hard-gating mechanism is more effective than implicit attention-based fusion for 3D feature incorporation**. To further illustrate the router's adaptive behavior, we added an analysis by clustering prompts based on semantic categories. We hope these additional experiments and analyses more convincingly demonstrate the novelty and necessity of the proposed gating mechanism.
>
> ## Regarding Q1: In Table 3, did the baseline models also utilize the training datasets from these benchmarks?
> Yes,  the baseline models in Table 3, recent state-of-the-art 3D models such as Video-3D-LLM and 3DRS, were trained on the training set of these benchmarks. In our ablation study (as described in Table 1), we use the same 3D dataset for a fair a fair comparstion.
>
> ## Regarding Q2: Interface with Low-Level Policy
> OmniEVA acts as a high-level planner, outputting target coordinates (e.g., grasp or place points) or abstract plans. These are passed to a separate, pre-trained low-level policy (a flow-matching model that predicts 6D end-effector poses from RGB-D inputs) for collision-aware path planning and execution.

---

> ### Author Response · Authors · 2025-11-27
>
> ## Regarding Q3: Generalization to Different Hardware
>
> We thank the reviewer for emphasizing real-world generalization.
>
> To address this, we first conducted extensive experiments **varying arm length (72cm–110cm) as a proxy for embodiment diversity**. Models were trained on 75cm, 88cm, and 110cm arms and tested on seen/unseen configurations. As shown in Table 5, OmniEVA-ER achieves an 81.89% average success rate, outperforming baselines by ~15% on seen lengths and generalizing robustly to unseen lengths (80.52% success).
>
> | Models / Embodiments | Average (SR) | Seen Arm Length (cm) | | | | Unseen Arm Length (cm) | | | | | | | |
> |---------------------|-------------|---------------------|-|-|-|-----------------------|-|-|-|-|-|-|-|
> | | | 75 | 88 | 110 | Seen Avg | 72 | 80 | 84 | 92 | 96 | 100 | 105 | Unseen Avg |
> | **RoboBrain2.0-7B** | 18.37 | 7.69 | 12.82 | 38.88 | 19.29 | 0 | 7.69 | 12.82 | 17.94 | 20.51 | 33.33 | 38.88 | 17.97 |
> | **OmniEVA-Base** | 43.30 | 51.28 | 41.02 | 44.44 | 45.61 | 35.89 | 48.71 | 43.58 | 41.02 | 51.28 | 36.11 | 38.88 | 42.32 |
> | **OmniEVA-ER w/o $r_{\text{embod}}$** | 65.35 | 69.23 | 71.79 | 69.44 | 70.17 | 41.02 | 64.10 | 66.66 | 61.53 | 69.23 | 72.22 | 69.44 | 63.29 |
> | **OmniEVA-ER** | 81.89 | 87.17 | 82.05 | 86.11 | 85.08 | 56.41 | 87.17 | 84.61 | 79.48 | 82.05 | 88.88 | 86.11 | 80.52 |
>
> For realworld deployment, we enhanced the **real-world evaluation** scale by conducting experiments on **two distinct wheeled mobile robots** with dual 6-DoF arms of **different arm spans (75cm vs. 70cm)**. This design intentionally introduces physical diversity to test generalization under practical constraints. The results, summarized in Table 6 of our paper, show that OmniEVA-ER delivers consistent performance improvements across both embodiments. For instance, in cluttered manipulation tasks, it achieves near-perfect performance (9/10 success rate in Cluttered Place) and maintains robust navigation capabilities. Importantly, OmniEVA-ER exhibits better adaptation to the more constrained 70cm arm span, highlighting its embodiment-aware robustness.
>
>
> | Models / Tasks | Cluttered Pick | | Cluttered Place | | Constrained Navigation | |
> |---------------|----------------|-|-----------------|-|----------------------|-|
> | | Embodiment 1 | Embodiment 2 | Embodiment 1 | Embodiment 2 | Embodiment 1 | Embodiment 2 |
> | **OmniEVA-Base** | 6 / 10 | 5 / 10 | 6 / 10 | 5 / 10 | 9 / 10 | 8 / 10 |
> | **OmniEVA-ER** | 8 / 10 | 7 / 10 | 9 / 10 | 9 / 10 | 8 / 10 | 8 / 10 |
>
> The cumulative evidence confirms that OmniEVA-ER **not only scales to real-world scenarios** but also **generalizes effectively across diverse physical embodiments**, addressing the reviewer's concerns about limited scale and cross-platform generalization. We believe these additions significantly strengthen the practical relevance of our work and open avenues for future research on embodied generalization.
>
> ## Regarding Q4: Comparison of Reasoning Time
> Inference Time on a single A6000 GPU:
> - Image input (no long-think): ~1.04s
> - Image input (with long-think): ~7.5s
> - Video/RGB-D input (16 frames): ~2.8s

---

### Official Review · Reviewer_5j5y · 2025-10-31

**Soundness:** 2
**Presentation:** 3
**Contribution:** 3
**Rating:** 6
**Confidence:** 3

**Summary:**

This paper identifies two main limitations of current embodied planners (Geometric Adaptability Gap and Embodiment Constraint Gap) and presents a "embodied versatile planner" that uses MLLMs for embodied tasks by introducing two innovations: 1. Task-Adaptive 3D Grounding mechanism using a gated router for explicit, selective 3D feature integration, and 2. embodiment-aware reasoning and training framework that explicitly factors in real-world physical constraints during task planning and learning. Experiments on a range of embodied reasoning benchmarks—including 2D, 3D, and physically grounded settings—demonstrate OmniEVA's performance advantages and improved executability in simulated and, to some degree, physical tasks.

**Strengths:**

1. the key analysis of current embodied planner limitations (Geometric Adaptability Gap and Embodiment Constraint Gap) is insightful and the two corresponding solutions make sense
2.The introduction of the TE-GRPO curriculum addresses the persistent gap between theoretical plans and practical robot feasibility.
3. OmniEVA consistently achieves or surpasses existing state-of-the-art performance on several challenging embodied tasks
4. The work is highly relevant to current efforts to bridge the gap between vision-language models and physically interactive, multimodal embodied AI.

**Weaknesses:**

1. While some results for physical robots are reported and select qualitative examples shown (Figures 7–11), the main quantitative results are from simulation and offline metrics. There is mention of real-world deployment in the conclusion and some visuals, but the scalability and generalization of the framework to a variety of robot morphologies or environments is not systematically confirmed.
2. While OmniEVA delivers strong results, the paper does not sufficiently discuss scenarios or limitations where the dynamic 3D gating may erroneously activate/deactivate or where the model’s embodiment-aware planning might fail or produce brittle outcomes. More failure cases and a nuanced discussion would add value. It would be great make clear the capability boundary of this approach and shed light on areas of improvement.

These two points lead to to question the claimed generalizability and applicability of the proposed method.

**Questions:**

1. Can the authors clarify whether the gate variable $g$ in TAGR is continuous (soft) or strictly discrete (hard) during inference? If it is sometimes fractional, how does this affect feature fusion, and are there cases where blended (i.e., partially injected) 3D cues yield improved or degraded performance?
2. Please provide detailed breakdowns of training data composition (custom vs. public), any pre-processing steps, and how 2D and 3D modalities are mixed during training.

---

> ### Author Response · Authors · 2025-11-27
>
> ## Regarding W1: Real-world experiment scale and cross-embodiment generalization
> We sincerely thank the reviewer for raising this important point regarding the scale of real-world experiments and the need to assess cross-embodiment generalization. We agree that systematic evaluation of scalability and generalization across diverse robot morphologies is crucial for demonstrating practical applicability. In direct response to this feedback, we have significantly expanded our experimental analysis to address these concerns comprehensively.
>
> First, to rigorously evaluate cross-embodiment generalization, we introduced a experimental framework that **varies a key robot parameter—arm length**—as a representative embodiment factor. Arm length directly impacts workspace constraints and serves as a suitable testbed for physical generalization. We generated multiple embodiments by scaling the same base robot arm model, producing lengths from 72cm to 110cm. Models were trained on arm lengths of 75cm, 88cm, and 110cm, and tested on both seen and unseen configurations (72cm, 80cm, 84cm, 92cm, 96cm, 105cm). **The model was dynamically adapted by prompting it with embodiment specifications (e.g., "The robot arm length is 72cm")**. As shown in Table 5 of our paper (reproduced below for convenience), OmniEVA-ER consistently outperforms baselines, achieving an average success rate of 81.89%—a substantial improvement over OmniEVA-Base (43.30%) and even the ablated variant without embodiment rewards (65.35%). Notably, **it generalizes robustly to unseen arm lengths**, attaining an 80.52% success rate, which demonstrates strong cross-embodiment adaptability.
>
> | Models / Embodiments | Average (SR) | Seen Arm Length (cm) | | | | Unseen Arm Length (cm) | | | | | | | |
> |---------------------|-------------|---------------------|-|-|-|-----------------------|-|-|-|-|-|-|-|
> | | | 75 | 88 | 110 | Seen Avg | 72 | 80 | 84 | 92 | 96 | 100 | 105 | Unseen Avg |
> | **RoboBrain2.0-7B** | 18.37 | 7.69 | 12.82 | 38.88 | 19.29 | 0 | 7.69 | 12.82 | 17.94 | 20.51 | 33.33 | 38.88 | 17.97 |
> | **OmniEVA-Base** | 43.30 | 51.28 | 41.02 | 44.44 | 45.61 | 35.89 | 48.71 | 43.58 | 41.02 | 51.28 | 36.11 | 38.88 | 42.32 |
> | **OmniEVA-ER w/o $r_{\text{embod}}$** | 65.35 | 69.23 | 71.79 | 69.44 | 70.17 | 41.02 | 64.10 | 66.66 | 61.53 | 69.23 | 72.22 | 69.44 | 63.29 |
> | **OmniEVA-ER** | 81.89 | 87.17 | 82.05 | 86.11 | 85.08 | 56.41 | 87.17 | 84.61 | 79.48 | 82.05 | 88.88 | 86.11 | 80.52 |
>
> Second, we enhanced the **real-world evaluation** scale by conducting experiments on **two distinct wheeled mobile robots with dual 6-DoF arms of different arm spans (75cm vs. 70cm)**. This design intentionally introduces physical diversity to test generalization under practical constraints. The results, summarized in Table 6 of our paper (shown below), demonstrate that OmniEVA-ER delivers consistent performance improvements across both embodiments. For instance, in cluttered manipulation tasks, **it achieves near-perfect performance (9/10 success rate in Cluttered Place)** and maintains robust navigation capabilities. Importantly, OmniEVA-ER exhibits better adaptation to the more constrained 70cm arm span, highlighting its embodiment-aware robustness.
>
> | Models / Tasks | Cluttered Pick | | Cluttered Place | | Constrained Navigation | |
> |---------------|----------------|-|-----------------|-|----------------------|-|
> | | Embodiment 1 | Embodiment 2 | Embodiment 1 | Embodiment 2 | Embodiment 1 | Embodiment 2 |
> | **OmniEVA-Base** | 6 / 10 | 5 / 10 | 6 / 10 | 5 / 10 | 9 / 10 | 8 / 10 |
> | **OmniEVA-ER** | 8 / 10 | 7 / 10 | 9 / 10 | 9 / 10 | 8 / 10 | 8 / 10 |
>
> The cumulative evidence confirms that OmniEVA-ER not only **scales to real-world scenarios** but also **generalizes effectively across diverse physical embodiments**, addressing the reviewer's concerns about limited scale and cross-platform generalization. We believe these additions significantly strengthen the practical relevance of our work and open avenues for future research on embodied generalization.

---

> ### Author Response · Authors · 2025-11-27
>
> ## Regarding W2: Limitations and Failure Cases of Dynamic 3D Gating and Embodiment-Aware Planning
>
> We sincerely thank the reviewer for highlighting the importance of discussing limitations and failure cases. In response, we have added a dedicated limitations section in the revised manuscript to provide a more nuanced analysis. Below, we address the key points raised, focusing on dynamic 3D gating and embodiment-aware planning.
>
> 1. **Dynamic 3D Gating Activation Analysis**: The task-adaptive gated router (TAGR) is designed to dynamically inject 3D features based on task context, but we acknowledge that erroneous activation or deactivation may occur in edge cases. To systematically evaluate this, we analyzed gate activation probabilities across semantic categories of prompts. As shown in Figure 4, the activation generally aligns with intuitive expectations—for instance, shape-related tasks exhibit high activation (76.9%), while state condition tasks show low activation (6.3%). **However, moderate activation in categories like "Spatial Relationship" (19.7%) and "Arrangement Pattern" (15.6%) may indicate suboptimal decisions**, as these tasks could benefit more substantially from 3D reasoning. This suggests that while TAGR is effective overall, its **scene-level gating approach might not fully capture fine-grained spatial demands**.
>
>     Future work could explore patch-level or object-wise gating to enable more precise activation, reducing potential errors in heterogeneous environments.
>
> 2. **Failure Scenarios in Embodiment-Aware Planning**: For embodiment-aware planning, our experiments demonstrated robustness to arm length variations (as detailed in Table 5), but we recognize that **unmodeled dynamics factors** could lead to brittle outcomes. For example, factors such as **joint limits, payload constraints, or installation height** were not explicitly incorporated and might cause plans to fail in complex scenarios like cluttered manipulation or dynamic environments. Although our ablation studies (Figure 5) show that the embodiment reward significantly improves performance, the planning may still struggle with unseen kinematics or real-world noise. **These limitations highlight the need for future work to integrate broader physical constraints** and simulation-based noise injection.
>
> These limitations highlight the need for future work to integrate broader physical constraints and simulation-based noise injection.
>
>
> ## Regarding Q1: About the gate variable in TAGR Module
> Thank you for your insightful question regarding the gating mechanism in our TAGR module. To clarify, **the gate variable in TAGR is strictly discrete (hard)** during both training and inference, taking values in {0, 1} as sampled via the **Gumbel-Softmax** trick. This design ensures a clear binary decision - either fully injecting 3D positional features or using purely 2D visual features - and avoids any fractional, blended injection of 3D cues.
>
> The primary reason for adopting hard gating **rather than soft gating** is to preserve the integrity of the 3D positional encodings. **Employing soft gating (which would apply a continuous weight to the 3D features) would distort the amplitudes of the 3D features, making the 3D features unstable** during training and significantly degrading model performance.
>
> Our ablation studies provide empirical evidence supporting this design choice. As shown in our comparative experiments:
>
> | Methods | SQA3D | ScanQA | Scan2Cap | ScanRefer |
> | :----- | :-----: | :-----: | :-----: | :-----: |
> | Dynamic 3D Integration (soft-gating) |  60.6 | 30.9 | 85.6 | 26.9 |
> | Dynamic 3D Integration (hard-gating) | 62.6 | 30.8 | 97.9 | 43.1 |
>
> The results clearly demonstrate that hard gating outperforms soft gating across all benchmarks. The performance gap is particularly pronounced on Scan2Cap (+12.3 points) and ScanRefer (+16.2 points), underscoring the detrimental effect of amplitude distortion on tasks requiring precise 3D understanding and descriptive generation.

---

> ### Author Response · Authors · 2025-11-27
>
> ## Regarding Q2: Detail of training data, and how 2D and 3D modalities are mixed during training.
>
> We thank the reviewer for the question regarding the training data composition and modality mixing in OmniEVA. We appreciate the opportunity to provide more detailed explanations.
>
> - **How 3d and 3D modalities are mixed**: We enable mixed-training across modalities by incorporating text-only, image-text, and RGB-D samples within the same mini-batches. To support this, we apply preprocessing to standardize tensor dimensions across all data types. For samples lacking 3D data (e.g., image-only or text-only inputs), we use a **3D data placeholder**. Each sample is tagged with a **binary depth flag** indicating whether valid 3D information is present. During training, this flag acts as a **zero mask**, effectively **disabling the 3D fusion pathway for samples without depth data**. This ensures that invalid 3D inputs do not interfere with training on non-3D tasks, while preserving batch consistency. Pseudocode illustrating this implementation is provided in **Appendix A.4.**
>
> - **Detail of Training data composition**: The detailed breakdown of all datasets used is provided in Appendix D.2. Our custom datasets are specifically designed to target core embodied skills. These include several 2D pointing (for manipulation) tasks within the ManiSkill simulator to develop precise spatial referencing, 3D scene imagination tasks to foster reasoning about occluded structures, and 3D subgoal prediction tasks using the Habitat simulator with MP3D and HM3D datasets to enable long-horizon navigation planning. This hybrid strategy ensures the model learns both broad spatial cognition and task-specific physical interactions.
>
> We hope this clarification adequately addresses your query. Please refer to the appendices for further technical details.

---

### Official Review · Reviewer_WCjS · 2025-11-01

**Soundness:** 3
**Presentation:** 3
**Contribution:** 3
**Rating:** 6
**Confidence:** 4

**Summary:**

This paper aims to address two limitations of current multi-modal large language models (MLLMs): insufficient 3D geometric understanding and the embodiment constraint gap. The authors propose OmniEVA, featuring task-adaptive 3D grounding and embodiment-aware reasoning. OmniEVA achieves state-of-the-art performances across 7 out of 8 public benchmarks, including VQA and navigation tasks. Furthermore, this paper introduces 4 primitive benchmarks to demonstrate the strong embodied capabilities of OmniEVA.

**Strengths:**

- Good motivation. Insufficient geometric understanding is a long-standing problem for MLLMs. And this paper takes one step further to address the gap between MLLMs and embodied AI applications by proposing embodiment-aware learning.
- Thoughtful model design for the utilization of geometric features. Task-adaptive gated router (TAGR) effectively addresses the information bias between 2D semantics and 3D geometry. And the ablation study demonstrates the advantages of TAGR.
- Embodiment-aware reasoning (TE-GRPO) is a novel solution to address the embodiment constraint gap, which properly integrates embodiment-specific rewards and reinforcement learning.
- The experimental results are comprehensive and the authors propose four extra embodiment-oriented benchmarks that can take considerable efforts.

**Weaknesses:**

- The whole training pipeline is not delivered clearly, especially the data front. I cannot find an introduction to how the OmniEVA is trained. I only know that it goes through three stages: TAGR pretraining, SFT, and RFT. But for each stage, what datasets are involved? The terminologies are also confusing, such as general embodied reasoning, omnibodied reasoning, and embodiment-aware reasoning. Moreover, this results in my confusion about which model (stage) is reported in various experiment tables.
- Limited insights beyond the overall good performance. The underlying factors include base model capability and data scale, which can make the comparisons unfair. The results turn out to be a pretty good performance with extensive training data and three training stages. However, we have no idea what data is significant and how much each stage contributes to the final performance.
- Despite the efforts regarding embodiment-aware reasoning, I think the claims (such as Line 425) about real-world robotics tasks are not appropriate since there are no real-robot tasks and results.

**Questions:**

- Figure 4 shows the difference in gate activation across some object categories that involve only semantics, e.g., armchair, towel, copier, box, etc. How to explain this? Does this indicate a bias stemming from data distribution?

---

> ### Author Response · Authors · 2025-11-27
>
> We thank the reviewer for the thorough review and valuable feedback. We sincerely appreciate the time and effort you have invested in evaluating our work. Your comments have been instrumental in helping us improve the clarity, depth, and overall quality of our manuscript. We have carefully revised the paper to address all the points you raised, with major additions highlighted in dark green for your convenience. Our point-by-point responses are detailed below.
>
> ## Regarding W1: Clarification of the Training Pipeline, Datasets, Terminology, and Model Reporting
>
> We sincerely apologize for the lack of detail which caused significant confusion. We have thoroughly revised the manuscript to provide a clear and comprehensive explanation.
>
> - ​**Explicit Three-Stage Pipeline**: We have significantly expanded Section 3.3, "Embodiment-Aware Training Strategy", to provide a detailed, sequential explanation of the entire pipeline. The revision now explicitly delineates the three stages: (1) **TAGR Pretraining**, (2) **Supervised Fine-Tuning (SFT)**, and (3) **Task- and Embodiment-aware Reinforced Finetuning (RFT)**. For each stage, we have added descriptions of the core objectives, the datasets involved (all 3D-aware for Stage 1; a hybrid of general multimodal and custom embodied data for Stage 2), and the training methodology.
> - ​**Model Evaluation Clarity**: To eliminate ambiguity, we have added explicit annotations throughout Section 3 & 4 (Methods and Experiments). The results in **Table 2, Table 3, and Table 4** are clearly marked as belonging to the **OmniEVA-Base model (from Stage 2)**, demonstrating its strong general reasoning capabilities. The results shown in **Figure 5**, which specifically evaluate the contribution of embodiment-aware reasoning, are performed on the **OmniEVA-ER model (from Stage 3, where ER means Embodiement-aware Reasoning)**.
> - ​**Terminology Clarification**: We have carefully refined the terminology throughout the paper. Potentially confusing terms have been replaced with clearer descriptions. The distinctions between "general embodied reasoning" (the focus of Stage 2, building **broad spatial cognition**) and "embodiment-aware reasoning" (the focus of Stage 3, **incorporating physical constraints**) are now explicitly defined in their respective sections.
>
>
> ## Regarding W2: Providing Deeper Insights into Factors Behind Performance
> This is a very insightful point. We thank the reviewer for pushing us to provide a more profound analysis beyond aggregate performance metrics. We agree that understanding the contribution of each component is essential.
>
> - **About Data Scale**: To directly address the concern about data scale and ensure a fair comparison for our core architectural innovation (the TAGR module), we would like to emphasize that our ablation study was designed from the outset under strictly controlled conditions. As presented in Table 1, **the comparison of different 3D integration methods was conducted using the  exact same 220K samples** from the standard datasets (SQA3D, ScanQA, etc.). This dataset choice was intentional and aligns with the established protocol used by recent SOTA baselines like **Video-3D-LLM** and **3DRS**, **ensuring a fair comparison**. The results robustly demonstrate that our dynamic hard-gating mechanism outperforms other variants, confirming that its advantage is attributable to the methodological innovation rather than data scale.
>
>     | Methods | Data | SQA3D | ScanQA | Scan2Cap | ScanRefer |
>     | :----- | :-----: | :-----: | :-----: | :-----: | :-----: |
>     | LLaVA-3D | 859 k | 55.6 | 27.0 | 79.2 | - |
>     | Video-3D-LLM | 220 k | 58.6 |  30.1 |   83.8 | - |
>     | 3DRS | 220 k | 60.6 |  30.3 | 86.1 | - |
>     | Hard-coded 3D Integration | 220 k | 61.2 |  31.5 |  95.5 | 41.2 |
>     | Without 3D Integration | 220 k | 61.2 | 30.7 | 75.5 | 4.3 |
>     | Dynamic 3D Integration (soft-gating) | 220 k |  60.6 | 30.9 | 85.6 | 26.9 |
>     | Dynamic 3D Integration (hard-gating) | 220 k |  62.6 | 30.8 | 97.9 | 43.1 |

---

> ### Author Response · Authors · 2025-11-27
>
> - ​**About Base Model**:  We fully agree with your observation regarding the difficulty of fair comparison due to differing base models. As you rightly pointed out, prior methods are built upon diverse and often incompatible foundations—for instance, LLaVA-video for Video-3D-LLM, LLaVA-next-video for 3DRS, and Qwen2.5VL for RoboBrain, this variability makes it difficult to conduct direct comparisons on a same base model. However, a core strength of our experimental design is that **all ablation variants** ('Hard-coded', 'Without 3D', 'Soft Gating', 'Hard Gating') and the Embodiement-aware RFT are **built on the exact same base model**. This controlled setup is crucial as it enables us to isolate and cleanly attribute any performance differences directly to our proposed innovations (namely, the task-adaptive gated router and the embodiment-aware reasoning framework), thereby providing unambiguous evidence for their effectiveness.
> - **Quantifying Stage-wise Contribution**: To provide a clear decomposition of each training stage's impact, we have conducted comprehensive evaluations that distinctly highlight the incremental value of our pipeline. **The effectiveness of the first two stages—TAGR Pretraining (Stage 1) and Supervised Fine-Tuning (Stage 2)—is jointly demonstrated in Table 1**. This table showcases how these foundational stages enable the model to acquire **flexible reasoning ability across diverse dimensional inputs**, with our method outperforming various 3D integration strategies under controlled conditions. The specific **contribution of Stage 3** (Task- and Embodiment-aware Reinforced Finetuning) is **quantitatively analyzed in Figure 5**. This ablation study directly measures the performance gains achieved by incorporating embodiment-aware reasoning, showing significant improvements in both primitive skills and downstream tasks. As illustrated in the figure, the OmniEVA-ER model (after Stage 3) demonstrates substantial performance enhancements over the OmniEVA-Base model (after Stage 2), particularly in tasks requiring physical constraint awareness. For instance, we observe improvements of +28.95% on Where2Approach and +34.28% on Where2Fit, which directly translate to higher success rates in complex mobile manipulation scenarios. This staged analysis provides clear evidence of how each phase of our training pipeline contributes uniquely to the final model's capabilities.
>
> ## Regarding W3: Despite the efforts regarding embodiment-aware reasoning, I think the claims (such as Line 425) about real-world robotics tasks are not appropriate since there are no real-robot tasks and results.
>
> We sincerely thank the reviewer for this critical feedback, which highlights the importance of validating our approach in real-world settings. We agree that claims about real-world applicability must be substantiated with concrete experimental evidence. In response, we have significantly expanded our evaluation to include rigorous real-robot experiments, ensuring that our assertions are well-supported.
>
> **Major Additions: Comprehensive Robotics Validation​**
>
> To address this concern directly, we have incorporated two key sets of experiments into the manuscript, each designed to rigorously test different aspects of embodiment-aware reasoning:
>
> 1. **Simulation-Based Embodiment Variation Study**: We conducted a systematic analysis in simulation to evaluate how our model generalizes across different physical embodiments by **varying a critical robot parameter—arm length**. This study, presented in Table 5​ (Section 4.3), assesses performance on both **seen** and **unseen** arm configurations (e.g., lengths from 72cm to 110cm), demonstrating the model's **adaptability to diverse constraints**. The results show that OmniEVA-ER achieves high success rates (e.g., 85.08% on seen lengths and 80.52% on unseen lengths), significantly outperforming baselines. This simulation-based approach provides controlled, reproducible evidence of generalization capabilities before real-world deployment.
>
>     | Models / Embodiments | Average (SR) | Seen Arm Length (cm) | | | | Unseen Arm Length (cm) | | | | | | | |
>     |---------------------|-------------|---------------------|-|-|-|-----------------------|-|-|-|-|-|-|-|
>     | | | 75 | 88 | 110 | Seen Avg | 72 | 80 | 84 | 92 | 96 | 100 | 105 | Unseen Avg |
>     | **RoboBrain2.0-7B** | 18.37 | 7.69 | 12.82 | 38.88 | 19.29 | 0 | 7.69 | 12.82 | 17.94 | 20.51 | 33.33 | 38.88 | 17.97 |
>     | **OmniEVA-Base** | 43.30 | 51.28 | 41.02 | 44.44 | 45.61 | 35.89 | 48.71 | 43.58 | 41.02 | 51.28 | 36.11 | 38.88 | 42.32 |
>     | **OmniEVA-ER w/o $r_{\text{embod}}$** | 65.35 | 69.23 | 71.79 | 69.44 | 70.17 | 41.02 | 64.10 | 66.66 | 61.53 | 69.23 | 72.22 | 69.44 | 63.29 |
>     | **OmniEVA-ER** | 81.89 | 87.17 | 82.05 | 86.11 | 85.08 | 56.41 | 87.17 | 84.61 | 79.48 | 82.05 | 88.88 | 86.11 | 80.52 |

---

> ### Author Response · Authors · 2025-11-27
>
> 2. ​**Real-World Deployment on Physical Robots**: To directly validate real-world applicability, we performed experiments on **two distinct wheeled mobile robots with dual 6-DoF arms of different arm spans (75cm vs. 70cm)**. These tests cover challenging tasks such as Cluttered Pick, Cluttered Place, and Constrained Navigation, which require precise spatial reasoning and physical compliance. The quantitative results in Table 6 (Section 4.3) show that OmniEVA-ER consistently improves performance over the base model, achieving near-perfect scores in manipulation tasks (e.g., 9/10 success in Cluttered Place) and maintaining robust navigation capabilities. This confirms the executability of our plans in actual physical environments.
>
>     | Models / Tasks | Cluttered Pick | | Cluttered Place | | Constrained Navigation | |
>     |---------------|----------------|-|-----------------|-|----------------------|-|
>     | | Embodiment 1 | Embodiment 2 | Embodiment 1 | Embodiment 2 | Embodiment 1 | Embodiment 2 |
>     | **OmniEVA-Base** | 6 / 10 | 5 / 10 | 6 / 10 | 5 / 10 | 9 / 10 | 8 / 10 |
>     | **OmniEVA-ER** | 8 / 10 | 7 / 10 | 9 / 10 | 9 / 10 | 8 / 10 | 8 / 10 |
>
> These comprehensive evaluations—spanning **simulation-based generalization** tests and **real-world physical deployments**—directly support our claims about real-world robotics applicability. The embodiment-aware reasoning framework, optimized through TE-GRPO, ensures that generated plans are not only semantically correct but also physically feasible, as evidenced by the high success rates in both controlled simulations and complex real-world scenarios. **We have revised the manuscript to clearly present these results and link them to our claims** .
>
> ## Regarding Q1: Gate Activation Analysis
> Thank you for this insightful question. We have significantly improved our analysis methodology to address precisely this concern.
>
> **Revised Analysis Method**
> ​We transitioned from **word-level statistics** to **semantic-level clustering of entire prompts using a sentence transformer**. This groups prompts by contextual meaning, providing a more accurate reflection of task requirements rather than isolated word frequencies. (See Figure 4 for detail. Full implementation details are provided in Appendix)
>
> **Key Findings from Semantic Clustering**
> - **High activation tasks**: Geometric queries (76.9%), action/activity prompts (50.9%), and occlusion reasoning (33.0%)
> - **Low activation tasks**: State descriptions (6.3%), counting (9.0%), and material queries (19.4%)
>
> This pattern **confirms the TAGR module's task-adaptive nature**, not data distribution bias. The gate activates specifically when 3D reasoning is essential, regardless of which objects are mentioned. We have updated Section 4.2 with this more rigorous analysis and added implementation detail of the analysis in Appendix C.2. Thanks for prompting this methodological improvement.

---

> ### Comment · Reviewer_WCjS · 2025-11-28
> **Reply to Rebuttal**
>
> Thanks for the authors' response. My concerns are well addressed. Despite a few flaws such as clarity issues, this is a technically solid paper and I lean towards acceptance.

---

> > ### Author Response · Authors · 2025-11-28
> >
> > We thank the reviewer for the final positive comment. We are truly delighted to know that you are satisfied with our revisions and that you now lean towards acceptance. We greatly appreciate the time and effort you have invested in reviewing our work.

---

### Official Review · Reviewer_BSqi · 2025-11-01

**Soundness:** 3
**Presentation:** 3
**Contribution:** 3
**Rating:** 4
**Confidence:** 2

**Summary:**

This paper introduces OmniEVA, an embodied planning framework that dynamically fuses 2D and 3D information via a task-adaptive gating mechanism (TAGR) and trains with embodiment-aware rewards (TE-GRPO). The model generates physically feasible plans for robotic manipulation and navigation. Extensive experiments on 8 benchmarks (2D, 3D, video, navigation, simulation, and real robot) show SOTA results with only 8B parameters.

**Strengths:**

1. The authors identify two critical yet often-ignored issues in embodied reasoning (geometric adaptability and embodiment constraints) and propose well-defined mechanisms to address both.

2. Explicitly integrating robot kinematic limits and workspace awareness into the planner is both novel and useful for real-world applications.

3. The model outperforms state-of-the-art baselines on several benchmarks and demonstrates credible real-robot deployment results.

**Weaknesses:**

1. The overall pipeline still largely follows prior MLLM-based planners, with incremental contributions (the gating and embodiment modules) rather than a fundamentally new paradigm. In addition, the claimed innovation of the gated router seems somewhat redundant, since modern attention layers already allow for selective fusion and weighting of modality-specific features. The paper does not convincingly justify why a separate gating mechanism is needed beyond what cross-attention or learned query routing could provide.

2. While results cover 2D and 3D reasoning, most experiments focus on tabletop or indoor manipulation. It remains unclear how well the approach scales to more complex, dynamic environments.

3. The real-world experiments are promising but relatively limited in scale. Only a few scenarios and robots are tested; cross-platform generalization is not discussed.

**Questions:**

No

---

> ### Author Response · Authors · 2025-11-27
>
> We thank the reviewer for the insightful comments and the positive rating. We appreciate the reviewer's perspective on the potential of attention mechanisms for feature fusion, which indeed provides an important baseline for understanding our design choices. Below we provide a point-by-point response to address the raised concerns.
>
> ## Regarding W1: Novelty and necessity of the gating mechanism
>
> We appreciate the reviewer's perspective that modern attention mechanisms can  perform feature weighting. This is a valuable point, and using cross-attention as a fusion baseline is indeed a logical comparison to better understand our contribution.
>
> However, **directly inputting 3D tokens as separate tokens** (instead of fusing them into visual tokens) doubles the token length and requires the model to learn cross-modal interactions from scratch, which is computationally expensive and data-inefficient. To validate this, we conducted two additional experiments using cross-attention between separate 3D and visual tokens (in both **interleaved** and **separated** arrangements). As shown in the table below, these approaches perform significantly worse than our gated fusion (e.g., ~6 points drop on SQA3D and ~50 points on Scan2Cap), confirming that naive cross-attention is suboptimal for 3D grounding.
>
> | Methods | SQA3D | ScanQA | Scan2Cap | ScanRefer |
> | :----- | :-----: | :-----: | :-----: | :-----: |
> | Cross-Attention (seperate visual & 3D tokens) | 55.1 |  27.5 |  43.3 | 4.5 |
> | Cross-Attention (interleaved visual & 3D tokens) |  55.8 |  27.5 |  42.0 | 3.6 |
> | Hard-coded 3D Integration | 61.2 |  31.5 |  95.5 | 41.2 |
> | Without 3D Integration |  61.2 | 30.7 | 75.5 | 4.3 |
> | Dynamic 3D Integration (hard-gating) | 62.6 | 30.8 | 97.9 | 43.1 |
>
> Moreover, we experimented with **soft-gating (continuous weighting of 3D features)**, which degrades performance because it unstable the 3D positional encodings, which are critical for spatial reasoning. In contrast, hard-gating preserves the numerical stability of 3D geometry representations while still enabling dynamic feature selection based on task context.  This demonstrates that a dedicated gating mechanism is more effective than implicit attention-based fusion for 3D feature incorporation.
>
> | Methods | SQA3D | ScanQA | Scan2Cap | ScanRefer |
> | :----- | :-----: | :-----: | :-----: | :-----: |
> | Dynamic 3D Integration (soft-gating) |  60.6 | 30.9 | 85.6 | 26.9 |
> | Dynamic 3D Integration (hard-gating) | 62.6 | 30.8 | 97.9 | 43.1 |
>
>
> ## Regarding W2: Scalability to complex, dynamic environments
>
> We thank the reviewer for raising this important point regarding dynamic environments. A key advantage of our gating mechanism is its orthogonality to temporal modeling techniques (e.g., MRoPE), making it readily adaptable to sequential, dynamic inputs.
>
> To empirically assess this scalability, we conducted new experiments in direct response to your feedback. While a **standardized 3D dynamic planning benchmark is currently lacking**, we designed scenarios with dynamic pedestrian obstructions (**shown in Appendix B.3**). In one task, the model had to locate a table that was initially **occluded by a pedestrian and became visible only after a change in the agent's viewpoint**. In a more challenging task, the model needed to detect a target object despite **a pedestrian continuously moving** within its field of view.
>
> OmniEVA succeeded in both cases, demonstrating an emergent capability to **handle visual uncertainty** from **dynamic obstructions**. These qualitative results confirm our method's inherent robustness and its strong potential for complex, dynamic settings.

---

> ### Author Response · Authors · 2025-11-27
>
> ## Regarding W3: Real-world experiment scale and cross-embodiment generalization
>
> We sincerely thank the reviewer for raising this important point regarding the scale of real-world experiments and the need to assess cross-embodiment generalization. We agree that evaluating generalization across diverse physical configurations is crucial for real-world applicability. In direct response to this feedback, we have significantly expanded our experimental analysis to address these concerns comprehensively.
>
> 1. **About Cross-Embodiment Generalization**: we introduced a novel experimental framework that **varies a key robot parameter—arm length**—as a representative embodiment factor. Arm length directly impacts workspace constraints and serves as a suitable testbed for physical generalization. We generated multiple embodiments by scaling the same base robot arm model, producing lengths from 72cm to 110cm. Models were trained on arm lengths of 75cm, 88cm, and 110cm, and **tested on both seen and unseen configurations** (72cm, 80cm, 84cm, 92cm, 96cm, 105cm). The model was dynamically adapted by prompting it with embodiment specifications (e.g., "The robot arm length is 72cm"). As shown in Table 5 of our paper, **OmniEVA-ER consistently outperforms baselines**, achieving an average success rate of 81.89%—a substantial improvement over OmniEVA-Base (43.30%) and even the ablated variant without embodiment rewards (65.35%). Notably, **it generalizes robustly to unseen arm lengths**, attaining an 80.52% success rate, which demonstrates strong cross-embodiment adaptability.
>
>     | Models / Embodiments | Average (SR) | Seen Arm Length (cm) | | | | Unseen Arm Length (cm) | | | | | | | |
>     |---------------------|-------------|---------------------|-|-|-|-----------------------|-|-|-|-|-|-|-|
>     | | | 75 | 88 | 110 | Seen Avg | 72 | 80 | 84 | 92 | 96 | 100 | 105 | Unseen Avg |
>     | **RoboBrain2.0-7B** | 18.37 | 7.69 | 12.82 | 38.88 | 19.29 | 0 | 7.69 | 12.82 | 17.94 | 20.51 | 33.33 | 38.88 | 17.97 |
>     | **OmniEVA-Base** | 43.30 | 51.28 | 41.02 | 44.44 | 45.61 | 35.89 | 48.71 | 43.58 | 41.02 | 51.28 | 36.11 | 38.88 | 42.32 |
>     | **OmniEVA-ER w/o $r_{\text{embod}}$** | 65.35 | 69.23 | 71.79 | 69.44 | 70.17 | 41.02 | 64.10 | 66.66 | 61.53 | 69.23 | 72.22 | 69.44 | 63.29 |
>     | **OmniEVA-ER** | 81.89 | 87.17 | 82.05 | 86.11 | 85.08 | 56.41 | 87.17 | 84.61 | 79.48 | 82.05 | 88.88 | 86.11 | 80.52 |
>
> 2. **Real World Evaluation**: we enhanced the **real-world evaluation** scale by conducting experiments on **two distinct wheeled mobile robots** with dual 6-DoF arms of **different arm spans (75cm vs. 70cm)**. This design intentionally introduces physical diversity to test generalization under practical constraints. The results, summarized in Table 6 of our paper, show that OmniEVA-ER delivers consistent performance improvements across both embodiments. For instance, **in cluttered manipulation tasks, it achieves near-perfect performance (9/10 success rate in Cluttered Place)** and maintains robust navigation capabilities. Importantly, OmniEVA-ER exhibits better adaptation to the more constrained 70cm arm span, highlighting its embodiment-aware robustness.
>
>
> | Models / Tasks | Cluttered Pick | | Cluttered Place | | Constrained Navigation | |
> |---------------|----------------|-|-----------------|-|----------------------|-|
> | | Embodiment 1 | Embodiment 2 | Embodiment 1 | Embodiment 2 | Embodiment 1 | Embodiment 2 |
> | **OmniEVA-Base** | 6 / 10 | 5 / 10 | 6 / 10 | 5 / 10 | 9 / 10 | 8 / 10 |
> | **OmniEVA-ER** | 8 / 10 | 7 / 10 | 9 / 10 | 9 / 10 | 8 / 10 | 8 / 10 |
>
> These enhancements are further supported by the ablation results in Figure 5, which illustrate the effectiveness of our embodiment-aware reasoning framework across tasks like Where2Approach and Where2Fit. The cumulative evidence confirms that OmniEVA-ER **not only scales to real-world scenarios** but also **generalizes effectively across diverse physical embodiments**, addressing the reviewer's concerns about limited scale and cross-platform generalization. We believe these additions significantly strengthen the practical relevance of our work and open avenues for future research on embodied generalization.

---

### Author Response · Authors · 2025-11-29
**Response to the Reviewers, ACs and PCs**

We sincerely thank you for overseeing the review process and all the Reviewers for their insightful and constructive feedback. The reviewers' comments have been invaluable in helping us significantly improve the paper. We have carefully addressed all the points raised in the reviews in our revised manuscript.

The revised manuscript has been updated accordingly. All changes and additions are highlighted in color for easy reference: **dark blue​ for Reviewer 1 (BSqi), dark green​ for Reviewer 2 (WCjS), brown​ for Reviewer 3 (5j5y), and dark purple​ for Reviewer 4 (BYjf).**

Below, we summarize the key revisions made in response to the primary concerns raised by the reviewers.

## **Summary of Revisions**


### **1. Enhanced Analysis of Generalization Across Embodiments and Sim-to-Real Transfer (Raised by R1, R2, R3)**

We have strengthened the evidence for our method's generalization capability from two perspectives:

- **Simulation with Embodiment Diversity**:​ We conducted extensive experiments **varying arm length (72cm–110cm) as a proxy for embodiment diversity**. The results (Table 5) show our method's significant improvement and **strong generalization to unseen arm lengths**.
- **Extended Real-World Evaluation**:​ We supplemented the **real-world** experiments with more quantitative results (in Table 6). This includes tests on pick/place/navigation tasks performed on **two different mobile manipulator platforms(with 75cm and 70cm arm spans)**, demonstrating compelling success rates and robust real world performance.

### **2. Extended Comparison with Alternative 3D Fusion Strategies (Raised by R1)**

As suggested, we have explicitly added comparisons with **alternative 3D fusion methodologies**. The results (in Table 1) now include:

- A detailed comparison with **Cross-Attention-based fusion**​ (with both separate and interleaved token arrangements).
- Further comparison with the **Soft-gating**​ mechanism.

The results confirm that our proposed approach outperforms these existing fusion strategies.


### **3. Clarifications on Training and Experimental Details (Raised by R2, R3, R4)**


We have thoroughly clarified the experimental setup and provided additional  details:

- **Training Stages**:​ We have explicitly rephrased the description of the **3-stage training** process to eliminate ambiguity. The model versions in all tables are now clearly labeled (e.g., **OmniEVA-Base** from Stage 2, **OmniEVA-ER** from Stage 3).
- **Extended Appendix**:​ We have added comprehensive details in the appendix, including:
    - PyTorch-style pseudocode for **mixed-training of 2/3D data** (Appendix A.4).
    Implementation details for **baselines and the gating activation ablation** (Appendix C).
    - Detailed description of the **reward function** used in the RFT stage (Appendix A.5).
    - Additional **generalization tests** in more complex scenarios with **dynamic pedestrian interference** (Appendix B.3).


We believe these revisions have directly addressed the reviewers' concerns and greatly enhanced the clarity, rigor, and impact of our work. We are grateful to the reviewers for their thoughtful comments and to the ACs and PCs for the time and effort invested in this process.

---

### Meta-Review · Area_Chair_7Yjr · 2026-01-12

**Summary:**

The paper targets the problem that models come up with plans that are technically correct but impossible for a real robot to actually do because of physical limits. The major concerns are redundancy of gating that the proposed method might be an incremental contribution, as modern attention layers already allow for selective modality fusion, and questioned whether the gating was continuous or discrete and how fractional injection of 3D features might affect performance. Multiple reviewers  found the three-stage training process and the specific datasets used in each stage to be poorly explained in the original manuscript, but has been improved during the rebuttal. The remaining concerns are primarily centered on the novelty of its architectural components and the depth of its experimental validation. As a result, I am leaning towards to acceptance, but would not mind if the decision being bumped down.

**Reviewer Concerns:**

Most concerns have been addressed. For reviewer BSqi, redundancy of gating should be addressed with the experiments showing that naive cross-attention and soft-gating perform significantly worse than their hard-gating mechanism. About dynamic environments, the authors added new qualitative results in the appendix involving dynamic pedestrian interference to demonstrate scalability beyond static scenes. Reviewer WCjS joined discussion mentioning the concerns have been addressed. About reviewer 5j5y, the added Table 5 shows experiments varying robot arm lengths to prove the model generalizes to different physical configurations. The authors also clarified that the gating is strictly discrete/hardduring inference to maintain 3D feature stability. for reviewer BYjf, the authors provided a detailed breakdown of how task and embodiment rewards are calculated using a feasibility checker, and clarified that the model outputs target coordinates to a separate, pre-trained flow-matching policy for final execution. There are still outstanding concerns. The major concern is still on the novelty side, characterizing the architecture as an incremental contribution rather than a fundamentally new paradigm. The authors also admitted that the current scene-level gating may still struggle with very fine-grained spatial demands and included it in their limitations section.

**Reviewer Scores:**

The scores are 4 6 6 4, and only the third reviewer joined the discussion, but there is no score changes. For the two scores of 4, the major concern is still on the novelty side, where reviewers may remain skeptical of the overall contribution level relative to existing VLMs, despite acknowledging the technical soundness of the rebuttal additions, and may not change the score.

---

### Decision · Program_Chairs · 2026-01-26

Accept (Poster)